# Coherence-free Entrywise Estimation of Eigenvectors in Low-rank Signal-plus-noise Matrix Models

**Hao Yan**
Department of Statistics
University of Wisconsin–Madison
Madison, WI 53706
United States of America
hyan84@wisc.edu

**Keith Levin**
Department of Statistics
University of Wisconsin–Madison
Madison, WI 53706
United States of America
kdlevin@wisc.edu

## Abstract

Spectral methods are widely used to estimate eigenvectors of a low-rank signal matrix subject to noise. These methods use the leading eigenspace of an observed matrix to estimate this low-rank signal. Typically, the entrywise estimation error of these methods depends on the coherence of the low-rank signal matrix with respect to the standard basis. In this work, we present a novel method for eigenvector estimation that avoids this dependence on coherence. Assuming a rank-one signal matrix, under mild technical conditions, the entrywise estimation error of our method provably has no dependence on the coherence under Gaussian noise (i.e., in the spiked Wigner model), and achieves the optimal estimation rate up to logarithmic factors. Simulations demonstrate that our method performs well under non-Gaussian noise and that an extension of our method to the case of a rank-$r$ signal matrix has little to no dependence on the coherence. In addition, we derive new metric entropy bounds for rank-$r$ singular subspaces under $\ell_{2,\infty}$ distance, which may be of independent interest. We use these new bounds to improve the best known lower bound for rank-$r$ eigenspace estimation under $\ell_{2,\infty}$ distance.

## 1 Introduction

Spectral methods are extensively used in contemporary data science and engineering [27]. The fundamental idea underlying these methods is that the eigenspace or singular subspace of an observed matrix reflects important structure present in the data from which it is derived. Spectral methods have been deployed successfully in a variety of tasks, including low-rank matrix denoising [30, 10], factor analysis [21, 32, 3, 61, 11, 62], community detection [49, 52, 1, 40, 50], pairwise ranking [43, 25] and matrix completion [48, 24, 7]. The widespread use of spectral methods has driven extensive research into the theoretical properties of eigenspaces and singular subspaces, yielding normal approximation results [10, 11, 33, 58, 5] as well as perturbation bounds [60, 20, 22, 45]. For a more comprehensive recent review of spectral methods, see [27].

### 1.1 Eigenspace estimation in low-rank matrix models

Consider an unknown symmetric matrix $\boldsymbol{M}^\star = \boldsymbol{U}^\star \boldsymbol{\Lambda}^\star \boldsymbol{U}^{\star\top} \in \mathbb{R}^{n \times n}$, where $\boldsymbol{U}^\star \in \mathbb{R}^{n \times r}$ has orthonormal columns and $\boldsymbol{\Lambda}^\star \in \mathbb{R}^{r \times r}$ is diagonal, containing the nonzero eigenvalues of $\boldsymbol{M}^\star$ ordered so that $|\lambda_1^\star| \geq |\lambda_2^\star| \geq \cdots \geq |\lambda_r^\star| > 0$. Our goal is to estimate $\boldsymbol{U}^\star$ from a noisy observation

$$\boldsymbol{Y} = \boldsymbol{M}^\star + \boldsymbol{W} \in \mathbb{R}^{n \times n}, \tag{1}$$

38th Conference on Neural Information Processing Systems (NeurIPS 2024).

where $\boldsymbol{W} = [W_{ij}]_{1 \leq i,j \leq n}$ is a symmetric random noise matrix with mean zero. We restrict our attention here to the symmetric case for the sake of simplicity, but we expect that our results can be extended to the asymmetric case using standard dilation arguments [53].

Throughout this paper, we assume that the entries of $\boldsymbol{W}$ are subgaussian.

**Assumption 1.** *The entries of $\boldsymbol{W}$ on and above the diagonal are independent and symmetric about zero with common variance $\sigma^2$ and common subgaussian parameter $\nu_W$.*

We remind the reader that the subgaussian parameter $\nu_W$ serves as a "proxy" for the variance. Indeed, in the Gaussian case, we have $\sigma^2 = \nu_W$, while $\sigma^2 \leq \nu_W$ more generally [54, 56].

Spectral methods often estimate $\boldsymbol{U}^\star$ directly using the $r$ leading eigenvectors $\boldsymbol{U}$ of $\boldsymbol{Y}$. As a result, the entrywise and row-wise behavior of $\boldsymbol{U}$ has attracted considerable attention [31, 22, 41, 2, 4, 14]. Given an estimator $\widehat{\boldsymbol{U}} \in \mathbb{R}^{n \times r}$, the estimation error is measured in terms of the $\ell_{2,\infty}$ distance

$$d_{2,\infty}(\widehat{\boldsymbol{U}}, \boldsymbol{U}^\star) = \min_{\boldsymbol{\Gamma} \in \mathbb{O}_r} \|\boldsymbol{U}^\star - \widehat{\boldsymbol{U}}\boldsymbol{\Gamma}\|_{2,\infty}, \tag{2}$$

where the presence of $\boldsymbol{\Gamma}$ is to resolve rotational non-identifiability. In the rank-one case, this reduces to the $\ell_\infty$ distance,

$$d_\infty(\widehat{\boldsymbol{u}}, \boldsymbol{u}^\star) = \min\left\{\|\boldsymbol{u}^\star - \widehat{\boldsymbol{u}}\|_\infty, \|\boldsymbol{u}^\star + \widehat{\boldsymbol{u}}\|_\infty\right\} \tag{3}$$

for $\boldsymbol{u}^\star \in \mathbb{R}^n$ and a given estimator $\widehat{\boldsymbol{u}} \in \mathbb{R}^n$. Estimation error bounds in $\ell_\infty$ or $\ell_{2,\infty}$ distance typically rely on the incoherence parameter $\mu$ of $\boldsymbol{U}^\star$, defined as

$$\mu = \frac{n}{r}\|\boldsymbol{U}^\star\|_{2,\infty}^2 \in [1, n/r].$$

In the rank-one case with Gaussian noise, if the leading eigenvalue $\lambda^\star$ of $\boldsymbol{M}^\star$ satisfies $|\lambda^\star| = \Omega(\sigma\sqrt{n})$, Theorem 4.1 of [27] shows that with probability at least $1 - O(n^{-8})$, the leading eigenvector $\boldsymbol{u}$ of $\boldsymbol{Y}$ satisfies

$$d_\infty(\boldsymbol{u}, \boldsymbol{u}^\star) \lesssim \frac{\sigma\left(\sqrt{\log n} + \sqrt{n}\|\boldsymbol{u}^\star\|_\infty\right)}{|\lambda^\star|} = \frac{\sigma\sqrt{\log n} + \sigma\sqrt{\mu}}{|\lambda^\star|}. \tag{4}$$

When $\sigma\sqrt{n} \lesssim |\lambda^\star| \lesssim \sigma\sqrt{n \log n}$, a regime of most interest (no ploynomial-time algorithm is known when $|\lambda^\star| \ll \sigma\sqrt{n}$ [6], and estimation is easy when $|\lambda^\star| \gg \sigma\sqrt{n \log n}$), we show in Lemma 1 that Equation (4) is not improvable up to log-factors , as a result of the Baik-Ben Arous-Péché (BBP) phase transition [8] (see [12, 38, 34] for BBP-style phase transitions in the setting of this paper).

**Lemma 1.** *Under Equation* (1) *with Gaussian noise, let $\boldsymbol{M}^\star = \lambda^\star \boldsymbol{u}^\star \boldsymbol{u}^{\star\top}$. If both limits $\lim_{n\to\infty} \lambda^\star/(\sigma\sqrt{n}) > 1$ and $\lim_{n\to\infty} \mu/n$ exist, then for any $\mu \in [1, n]$, there exists $\boldsymbol{u}^\star \in \mathbb{S}^{n-1}$ such that almost surely,*

$$\liminf_{n\to\infty} d_\infty(\boldsymbol{u}, \boldsymbol{u}^\star) \geq \lim_{n\to\infty} \frac{\sigma^2\sqrt{n\mu}}{2\sqrt{2}|\lambda^\star|^2}. \tag{5}$$

Equation (5) shows that the spectral estimate $\boldsymbol{u}$ has an intrinsic dependence on $\mu$, with especially bad performance when $\mu$ is large and $|\lambda^\star| = \Theta(\sigma\sqrt{n \log n})$. In this large-$\mu$ regime, beyond low-rankedness, $\boldsymbol{M}^\star$ exhibits additional structure (e.g., sparsity) that is not fully utilized by the spectral estimator. This suggests that the dependence on $\mu$ in Equations (4) and (5) is a shortcoming of the spectral estimator. In Algorithm 1, we present a new estimator designed to remove this dependence on $\mu$. Theorem 1 shows that up to log-factors, it matches the minimax lower bound discussed below (see Equation (9) in Section 1.2). Experiments in Section 5 further support our theoretical results.

In the rank-$r$ case with Gaussian noise, Theorem 4.2 in [27] shows that when $|\lambda_r^\star| \gtrsim \sigma\sqrt{n \log n}$,

$$d_{2,\infty}(\boldsymbol{U}, \boldsymbol{U}^\star) \lesssim \frac{\sigma\left(\kappa\sqrt{\mu r} + \sqrt{r \log n}\right)}{|\lambda_r^\star|} \tag{6}$$

with probability at least $1 - O(n^{-8})$, where $\kappa = |\lambda_1^\star|/|\lambda_r^\star|$ is the condition number of $\boldsymbol{M}^\star$. Here again, the estimation error depends on $\mu$, and we conjecture that this dependence is also sub-optimal. Algorithm 2 extends our rank-1 estimation algorithm to this more general rank-$r$ case. Experiments in Section 5 show that Algorithm 2 outperforms the naïve spectral method in the general rank-$r$ case, with little to no sensitivity to the coherence $\mu$. We note in passing that the dependence on $\kappa$ in Equation (6) can likely be removed [3, 62, 57], though we do not pursue this here.

## 1.2 Minimax lower bounds for subspace estimation

Minimax lower bounds have been established for a variety of subspace estimation problems, including sparse PCA [21, 55], matrix denoising [20], structural matrix estimation [19], network estimation [35, 63] and estimating linear functions of eigenvectors [42, 28]. Most of these studies focus on minimax lower bounds under the Frobenius or operator norm, derived using the packing numbers of Grassmann manifolds [46, 13]. In the matrix denoising literature, it is well-known that for $r \geq 1$,

$$\min_{\boldsymbol{\Gamma} \in \mathbb{O}_r} \left\| \widehat{\boldsymbol{U}} \boldsymbol{\Gamma} - \boldsymbol{U}^\star \right\|_{\mathrm{F}} \gtrsim \min \left\{ \frac{\sigma \sqrt{nr}}{|\lambda_r^\star|}, \sqrt{r} \right\} \tag{7}$$

holds in a minimax sense (see Theorem 3 in [28] or Theorem 4 in [63]). Far fewer papers have considered lower bounds under $d_{2,\infty}$ [18, 4], and these results are derived via the trivial lower bound

$$d_{2,\infty}(\widehat{\boldsymbol{U}}, \boldsymbol{U}^\star) \geq \frac{1}{\sqrt{n}} \min_{\boldsymbol{\Gamma} \in \mathbb{O}_r} \left\| \widehat{\boldsymbol{U}} \boldsymbol{\Gamma} - \boldsymbol{U}^\star \right\|_{\mathrm{F}}, \tag{8}$$

which holds for any $\widehat{\boldsymbol{U}}, \boldsymbol{U}^\star \in \mathbb{R}^{n \times r}$. Applying the lower bounds in Equations (7) and (8), we have

$$\sup_{\boldsymbol{U}^\star \in \mathbb{R}^{n \times r} : \boldsymbol{U}^{\star\top} \boldsymbol{U}^\star = \boldsymbol{I}_r} d_{2,\infty}(\widehat{\boldsymbol{U}}, \boldsymbol{U}^\star) \gtrsim \min \left\{ \frac{\sigma \sqrt{r}}{|\lambda_r^\star|}, \sqrt{\frac{r}{n}} \right\} \tag{9}$$

holds for any estimator $\widehat{\boldsymbol{U}} \in \mathbb{R}^{n \times r}$. When $\boldsymbol{U}^\star$ is incoherent, meaning that $\mu = O(1)$, this lower bound is achieved by the spectral estimation rate in Equation (6) when $|\lambda_r^\star| = \Omega(\sigma \sqrt{n \log n})$, and is achieved trivially by $\widehat{\boldsymbol{U}} = \boldsymbol{0}_{n,r}$ when $|\lambda_r^\star| = o(\sigma \sqrt{n})$. On the other hand, when $\boldsymbol{U}^\star$ is coherent, in the sense that $\mu = \omega(1)$, and $|\lambda_r^\star| = \Omega(\sigma \sqrt{n \log n})$, our discussion above in Section (1.1) (including our new results in Theorem 1) suggests that the rate in Equation (9) can be achieved up to log-factors.

This leaves open the question of the minimax rate when $\mu = \omega(1)$ and $|\lambda_r^\star| = o(\sigma \sqrt{n})$. In this case, the lower bound in Equation (9) cannot exceed $\sqrt{r/n}$. This seems suboptimal, as we expect some dependence on $\|\boldsymbol{U}^\star\|_{2,\infty}$ (for example, consider the extreme case when $\lambda^\star$ is very near zero). This suboptimality arises from the naïve lower bound in Equation (8). In Theorem 2, we improve this lower bound, removing the $\sqrt{r/n}$ dependence in Equation (9). This improved lower bound makes use of novel metric entropy bounds for singular subspaces, which may be of independent interest.

## 1.3 Notation and roadmap

We use $C$ to denote a constant whose precise values may change from line to line. For a positive integer $n$, we write $[n] = \{1, 2, \ldots, n\}$. $|\mathcal{A}|$ denotes the cardinality of a set $\mathcal{A}$. For real numbers $a$ and $b$, we write $a \vee b = \max\{a, b\}$ and $a \wedge b = \min\{a, b\}$. For a vector $\boldsymbol{v} = (v_1, v_2, \ldots, v_n)^\top \in \mathbb{R}^n$, we use the norms $\|\boldsymbol{v}\|_2 = \sqrt{\sum_{i=1}^n v_i^2}$ and $\|\boldsymbol{v}\|_\infty = \max_i |v_i|$. We let $\boldsymbol{e}_i \in \mathbb{R}^n, i \in [n]$ denote the standard basis vectors of $\mathbb{R}^n$. $\mathbb{S}^{n-1} = \{\boldsymbol{u} \in \mathbb{R}^n : \|\boldsymbol{u}\|_2 = 1\}$ denotes the unit sphere. For a matrix $\boldsymbol{M} \in \mathbb{R}^{n \times n}$, $\boldsymbol{M}_{i,\cdot}$ denotes its $i$-th row as a row vector, $\|\boldsymbol{M}\|$ denotes its operator norm and $\|\boldsymbol{M}\|_{2,\infty} = \max_{i \in [n]} \|\boldsymbol{M}_{i,\cdot}\|_2$ indicates the maximum row-wise $\ell_2$ norm. $\boldsymbol{I}_n \in \mathbb{R}^n$ denotes the $n$-by-$n$ identity matrix. $\mathbb{O}_r$ denotes the $r$-dimensional orthogonal group. We use both standard Landau notation and asymptotic notation: for positive functions $f(n)$ and $g(n)$, we write $f(n) \gg g(n)$, $f(n) = \omega(g(n))$ or $g(n) = o(f(n))$ if $f(n)/g(n) \to \infty$ as $n \to \infty$. We write $f(n) \gtrsim g(n)$, $f(n) = \Omega(g(n))$ or $g(n) = O(f(n))$ if for some constant $C > 0$, we have $f(n)/g(n) \geq C$ for all sufficiently large $n$. We write $f(n) = \Theta(g(n))$ if both $f(n) = O(g(n))$ and $g(n) = O(f(n))$.

The remainder of the paper is organized as follows. In Section 2, we study the eigenspace estimation problem for rank-one matrices and propose a new algorithm that achieves the minimax optimal error rate up to logarithmic factors in the growth regime where $|\lambda^\star| = \Omega(\sigma \sqrt{n \log n})$ (Theorem 1). In Section 3, we extend this algorithm to rank-$r$ eigenspace estimation. In Section 4, we present theoretical results for the metric entropy of subspaces under $d_\infty$ and $d_{2,\infty}$ and improve Equation (9) under the growth regime where $|\lambda^\star| = O(\sigma \sqrt{n})$ (Theorem 2). Numerical results are provided in Section 5. We conclude in Section 6 with a discussion of the limitations of our study and directions for future work. Detailed proofs of all lemmas and theorems can be found in the appendix.

## 2 Rank-one matrix eigenspace estimation

In this section, we study the model in Equation (1) when $M^\star$ is rank-one with eigendecomposition $M^\star = \lambda^\star u^\star u^{\star\top}$, where $\lambda^\star \in \mathbb{R}$ and $u^\star \in \mathbb{S}^{n-1}$. As discussed in Section 1, spectral methods may have sub-optimal dependence on $\mu$ compared to the lower bound in Equation (9). We show that a better estimator is possible by working with a carefully selected subset of entries of $Y$. We start with a key observation in Lemma 2, which states that any unit vector contains a subset of large entries, and this subset has a sufficiently large cardinality. This subset is the key to our new estimator.

**Lemma 2.** *Let $\mathcal{A} \subset [0,1]$ be the set*

$$\mathcal{A} = \left\{ \log^{-\frac{1}{2}} n, \ldots, \log^{-\frac{\lceil L \rceil - 1}{2}} n, \log^{-\frac{L}{2}} n \right\}, \tag{10}$$

*where $L$ is given by*

$$L = \frac{\log(2n)}{\log \log n}. \tag{11}$$

*For $n$ sufficiently large, for every $v \in \mathbb{S}^{n-1}$, there exists $\alpha_0 \in \mathcal{A}$ such that*

$$\frac{1}{\alpha_0^2} \geq |\{i : |v_i| \geq \alpha_0\}| > \frac{1}{\alpha_0^2 \log^2 n}. \tag{12}$$

To motivate Algorithm 1, suppose $u^\star$ is entrywise positive. For $\alpha_0 \in \mathcal{A}$, denote the set in Equation (12) by $I_{\alpha_0}$. By Equation (12), the sum of entries $M_{ij}^\star$ with $i, j \in I_{\alpha_0}$ grows as $\Omega(|\lambda^\star||I_{\alpha_0}|\log^{-2} n)$, while the sum of the corresponding entries of $W$ grows as $O(\sigma|I_{\alpha_0}|\sqrt{\log n})$. That is, when $|\lambda^\star|$ is sufficiently large, the signal contained in the entries $i, j \in I_{\alpha_0}$ dominates the noise. If we knew $I_{\alpha_0}$, utilizing the entries in $I_{\alpha_0}$ would reduce the estimation error incurred by small entries of $u^\star$. In practice, we do not know $I_{\alpha_0}$ and must estimate such a subset. To ensure that this is possible, we impose a technical assumption on $u^\star$. We discuss this assumption below in Remark 2.

**Assumption 2.** *There exists an $\alpha_0 \in \mathcal{A}$ satisfying Equation (12) and a constant $0 < \epsilon_0 \leq 1$ such that for all sufficiently large $n$,*

$$\{i : |u_i^\star| \in [(1 - \epsilon_0)\alpha_0, (1 + \epsilon_0)\alpha_0]\} = \emptyset.$$

We pause to give a few examples to illustrate Assumption 2. First, consider $u^\star = c_1 e_1 + c_2 n^{-1/2} \mathbf{1}_n$, with $c_1, c_2 = \Theta(1)$ chosen so that $\|u^\star\|_2 = 1$. We note that $c_1, c_2$ both depend on $n$, but are bounded away from zero as $n$ grows, and one can verify that Assumption 2 holds with $\alpha_0 = \log^{-1/2} n$ and $\epsilon_0 = 1/2$. As another example, consider $u^\star = n^{-1/2} \mathbf{1}_n \in \mathbb{R}^n$. One may verify that taking $\alpha_0 = (\log n)^{-L/2} = 1/\sqrt{2n}$ and $\epsilon_0 = 0.4$ satisfies the conditions in Assumption 2.

As an example of a setting that violates Assumption 2, consider $u^\star$ obtained by renormalizing a vector of i.i.d. Gaussians. This results in $u^\star$ being Haar-distributed on $\mathbb{S}^{n-1}$ and Assumption 2 is violated with high probability. To see this, note that $u^\star \approx g/\sqrt{n}$ where $g \sim N(0, I_n)$ (see Theorem 3.4.6 in [54]). Since with high probability $\|g\|_\infty = O(\sqrt{\log n})$, Equation (12) holds only when $\alpha_0 \approx 1/\sqrt{n}$. For Assumption 2 to hold, there must be a gap of $\Theta(1)$ between the entries of $g$, which fails with high probability.

It is tempting to conclude from the counter-example just given that renormalizing a vector of i.i.d. entries must necessarily result in a $u^\star$ that violates Assumption 2, but this is not always the case. If the entrywise distribution has suitable structure, $u^\star$ may still obey Assumption 2. As an illustration, suppose that $u^\star$ is obtained by renormalizing a vector $g = (g_1, g_2, \ldots, g_n)^\top \in \mathbb{R}^n$ with i.i.d. entries from a distribution with variance 1, so that $u^\star \approx g/\sqrt{n}$. If the $g_i$ are drawn by taking $g_i = a$ with probability $p$ and $g_i = b$ with probability $1 - p$, then each entry of $u^\star$ is either approximately $ap/\sqrt{n}$ or approximately $b(1-p)/\sqrt{n}$. Choosing $a, b$ and $p$ appropriately, we can ensure a gap between the entries of $u^\star$ of size $O(n^{-1/2})$, and we can take $\alpha_0 = (\log n)^{-L} \approx n^{-1/2}$.

Our new estimator of $u^\star$ is a refinement based on the leading eigenvector and eigenvalue of $Y$. For the spectral estimator to provide a useful initialization, we make Assumption 3 on $\lambda^\star$.

**Assumption 3.** *There exists a constant $C_1 > 2400/\epsilon_0$, where $\epsilon_0$ is as in Assumption 2, such that the leading eigenvalue $\lambda^\star$ of $M^\star$ satisfies*

$$|\lambda^\star| \geq C_1 \sqrt{\nu_W n \log n}. \tag{13}$$

**Remark 1.** *The dependence on $\epsilon_0$ in Assumption 3 is for technical reasons discussed in Remark 2. In our proofs, we do not optimize the dependence on $C_1$ and assume that $C_1 > 2400/\epsilon_0$. As demonstrated in our experiments in Section 5, $|\lambda^\star| \geq \sqrt{\nu_W n \log n}$ appears sufficient in practice. When $|\lambda^\star| \leq \sqrt{\nu_W n}$, the spectral estimator fails to provide any useful initial estimate and it is believed that no polynomial-time algorithm can succeed. We provide more discussion on this matter in Remark 6.*

The final ingredient required for Algorithm 1 is a leading eigenvalue estimate $\widehat{\lambda}$ that recovers the true signal eigenvalue $\lambda^\star$ suitably well.

**Assumption 4.** *Under Assumption 3, $\widehat{\lambda}$ is such that with probability at least $1 - O(n^{-8} \log n)$,*

$$\left| \widehat{\lambda} - \lambda^\star \right| \leq C_2 \sqrt{\nu_W} \log^{5/2} n.$$

Assumption 4 seems stringent at first. The top eigenvalue $\lambda$ of $\boldsymbol{Y}$ achieves only a $O(\sqrt{\nu_W n})$ error rate (see Lemma 2.2 and Equation (3.12) in [27]). This is because $\lambda = \lambda^\star + n\sigma^2/\lambda^\star + O(\sqrt{\nu_W \log n})$ (see [47] or Theorem 2.3 in [23]; see also [26, 51, 17]). Luckily, in our setting, the bias-corrected estimate

$$\widehat{\lambda}_c = \frac{1}{2} \left( \lambda + \sqrt{\lambda^2 - 4n\sigma^2} \right), \tag{14}$$

*does* satisfy Assumption 4 [26]. Another estimator, which falls naturally out of Algorithm 1, also satisfies Assumption 4. We find that it performs similarly to the debiased estimator $\widehat{\lambda}_c$ empirically, and so we do not explore it here. Theoretical results for this estimator are in the appendix.

With the above assumptions in hand, we propose a new method given in Algorithm 1. We note that the main computational bottleneck of Algorithm 1 is to find the leading eigenvector $\boldsymbol{u}$ of $\boldsymbol{Y}$, and thus the runtime is essentially the same as for standard spectral methods (see, e.g., [37]).

---

**Algorithm 1** Coherence-free eigenvector estimation algorithm

---

**Input:** Observed matrix $\boldsymbol{Y} \in \mathbb{R}^{n \times n}$; leading eigenvalue estimate $\widehat{\lambda}$; parameter $\beta > 0$.
**Output:** $\widehat{\boldsymbol{u}} \in \mathbb{R}^n$
1: If $\widehat{\lambda} < 0$, set $\boldsymbol{Y} = -\boldsymbol{Y}$. Obtain the top eigenvector $\boldsymbol{u} \in \mathbb{S}^{n-1}$ of $\boldsymbol{Y}$.
2: Pick any $\widehat{\alpha} \in \mathcal{A}$ such that the set $\hat{I} = \{i : |u_i| \geq \widehat{\alpha}\}$ satisfies

$$|\hat{I}| \geq \frac{1}{\widehat{\alpha}^2 \log^2 n}, \quad \text{and} \tag{15}$$

$$\{i : (1 - \beta)\widehat{\alpha} < |u_i| < (1 + \beta)\widehat{\alpha}\} = \emptyset, \tag{16}$$

3: Let $\boldsymbol{Q} \in \mathbb{R}^{n \times n}$ be diagonal with $Q_{kk} = \begin{cases} \text{sgn}\,(u_k) & \text{if } k \in \hat{I} \\ 1 & \text{if } k \in \hat{I}^c \end{cases}$ and let $\widetilde{\boldsymbol{Y}} = \boldsymbol{Q}\boldsymbol{Y}\boldsymbol{Q}$.
4: Set $\widehat{S} = \sqrt{\sum_{j,k \in \hat{I}} \widetilde{Y}_{jk}}$ and let $\widehat{v}_j = \left( \sum_{k \in \hat{I}} \widetilde{Y}_{jk} \right) \Big/ \left( \widehat{S}\sqrt{\widehat{\lambda}} \right)$ for $j \in [n]$.
5: For each $j \in [n]$, set $\widehat{u}_j = u_j$ if $|u_j| \leq \left( \sigma/\widehat{\lambda} \right) \log n$, and $\widehat{u}_j = Q_{jj}\widehat{v}_j$ otherwise.

---

**Remark 2.** *In our proofs, we set $\beta = \epsilon_0/2$. Equation (16), Assumption 2 and the $\epsilon_0$-dependence in Assumption 3 are technical requirements to ensure that with high probability, $\hat{I}$ is one of a few deterministic sets, avoiding the complicated dependence between $\hat{I}$ and $\boldsymbol{W}$. Empirically, Algorithm 1 works well even without these technical conditions. We conjecture that Assumption 2 as well as the $\epsilon_0$-dependence in Assumption 3 can be removed. See Section 5 for further discussion.*

As alluded to above, the intuition behind Algorithm 1 is that we aim to concentrate our efforts on estimating the large entries of $\boldsymbol{u}^\star$. Consider an entry of $\boldsymbol{Y}$ given by $\lambda^\star u_i^\star u_j^\star + W_{ij}$. Intuitively, locations corresponding to small entries of $\boldsymbol{u}^\star$ produce small $u_i^\star u_j^\star$. These entries of $\boldsymbol{Y}$ have a small signal to noise ratio compared to those arising from products of large entries of $\boldsymbol{u}^\star$. If we knew the locations of the large entries of $\boldsymbol{u}^\star$, we could use them to obtain more accurate estimates of $\boldsymbol{u}^\star$. Essentially, both Algorithm 1 above and Algorithm 2 presented below consist of two parts: finding the large locations, and using those locations to improve our initial spectral estimate of $\boldsymbol{u}^\star$.

Algorithm 1 assumes that the entrywise variance $\sigma^2$ of $\boldsymbol{W}$ is known. Of course, in practice, this is not the case, and we must estimate $\sigma^2$. There are several well-established methods for this estimation task. For example, when $\boldsymbol{W}$ is asymmetric, [36] introduces an estimator based on the median singular value of $\boldsymbol{Y}$. In our case, it suffices to estimate $\sigma^2$ using a simple plug-in estimator

$$\widehat{\sigma}^2 = \frac{2}{n(n+1)} \sum_{1 \leq i \leq j \leq n} \left( Y_{ij} - \widehat{M}_{ij} \right)^2, \tag{17}$$

where $\widehat{\boldsymbol{M}} = \lambda \boldsymbol{u} \boldsymbol{u}^\top \in \mathbb{R}^{n \times n}$. In general, if $\boldsymbol{M}^\star$ has rank $r$, then we set $\widehat{\boldsymbol{M}} = \boldsymbol{U} \boldsymbol{\Lambda} \boldsymbol{U}^\top$, where $\boldsymbol{\Lambda}$ is the leading $r$ eigenvalues of $\boldsymbol{Y}$ (sorted by non-increasing magnitude) and $\boldsymbol{U} \in \mathbb{R}^{n \times r}$ contains the corresponding $r$ leading orthonormal eigenvectors as its columns. Lemma 3 controls the estimation error of the plug-in estimator $\widehat{\sigma}^2$ for a general rank-$r$ signal matrix.

**Lemma 3.** *Under the model given in Equation* (1)*, let $\boldsymbol{M}^\star = \boldsymbol{U}^\star \boldsymbol{\Lambda}^\star \boldsymbol{U}^{\star\top}$ be a rank-$r$ matrix with $r \geq 1$, where $\boldsymbol{\Lambda}^\star = \mathrm{diag}\left(\lambda_1^\star, \lambda_2^\star, \ldots, \lambda_r^\star\right)$ such that $|\lambda_1^\star| \geq \cdots \geq |\lambda_r^\star|$. Suppose that Assumption 1 holds and that $|\lambda_r^\star| \geq 20\sqrt{\nu_W n}$, then the estimator $\widehat{\sigma}^2$ given in Equation* (17) *is such that with probability at least $1 - O(n^{-8})$,*

$$\left| \widehat{\sigma}^2 - \sigma^2 \right| \leq \frac{400 \nu_W r}{n} + \frac{4 \nu_W \sqrt{\log n}}{cn} + \frac{200 \nu_W r \sqrt{\log n}}{n^{3/2}} \tag{18}$$

*where $c > 0$ is a universal constant.*

**Remark 3.** *We use the plug-in estimator $\widehat{\sigma}$ in the debiased estimator $\widehat{\lambda}_c$ in Equation* (14) *and to construct $\widehat{\boldsymbol{u}}$ in Step 5 of Algorithm 1. Lemma 3 shows that this only introduces an extra log-factor to the error bound, which does not affect the estimation error rate of either $\lambda^\star$ or $\boldsymbol{u}^\star$.*

Our main result, Theorem 1, controls the estimation error of Algorithm 1, as measured under $\ell_\infty$.

**Theorem 1.** *Under the model in Equation* (1)*, suppose that Assumptions 1, 2, 3 and 4 hold. Then for $n$ sufficiently large, the estimate $\widehat{\boldsymbol{u}} \in \mathbb{R}^n$ produced by Algorithm 1 satisfies*

$$d_\infty \left( \widehat{\boldsymbol{u}}, \boldsymbol{u}^\star \right) \leq \frac{C \sqrt{\nu_W} (\log n)^{5/2}}{|\lambda^\star|}$$

*with probability at least $1 - O(n^{-8} \log n)$, where $C > 0$ is a universal constant.*

**Remark 4.** *Under Gaussian noise, in the regime $|\lambda^\star| = \Omega(\sigma \sqrt{n \log n})$, our upper bound is minimax rate-optimal up to log-factors compared to the lower bound in Equation* (9)*. In particular, the rate obtained in Theorem 1 does not depend on the coherence parameter $\mu$. A more careful analysis might be able to remove some of the log-factors in Theorem 1, but we leave this matter for future work.*

## 3 Rank-$r$ matrix eigenspace estimation

To handle the more general case in which the signal matrix $\boldsymbol{M}^\star$ is rank $r$, we propose Algorithm 2, which yields an estimate of $\boldsymbol{U}^\star$. This is achieved by estimating the $r$ leading eigenvectors separately, then combining them into an estimate $\widehat{\boldsymbol{U}} \in \mathbb{R}^{n \times r}$. We explore the empirical performance of Algorithm 2 via simulation in Section 5.2 and leave its theoretical analysis to future work. Algorithm 2 requires the observed matrix $\boldsymbol{Y}$ and an estimate of the $k$-th leading eigenvalue $\lambda_k^\star$ of $\boldsymbol{M}^\star$ as input. Similar to the rank-one case, the top-$r$ leading eigenvalues $\lambda_1, \lambda_2, \ldots, \lambda_r$ of $\boldsymbol{Y}$ are biased [47]. We again use a debiased estimator $\widehat{\lambda}_{k,c}$ for $k \in [r]$ as input to Algorithm 2, given by

$$\widehat{\lambda}_{k,c} = \frac{1}{2} \left( \lambda_k + \sqrt{\lambda_k^2 - 4n\sigma^2} \right). \tag{19}$$

Algorithm 2 is a natural extension of Algorithm 1. Essentially, it converts the problem of estimating $\boldsymbol{u}_k^\star$ into an eigenvector estimation problem under a rank-one signal-plus-noise model given by

$$\boldsymbol{Y} = \lambda_k^\star \boldsymbol{u}_k^\star \boldsymbol{u}_k^{\star\top} + (\boldsymbol{M}_{-k}^\star + \boldsymbol{W}) = \lambda_k^\star \boldsymbol{u}_k^\star \boldsymbol{u}_k^{\star\top} + \left( \boldsymbol{M}^\star - \lambda_k^\star \boldsymbol{u}_k^\star \boldsymbol{u}_k^{\star\top} + \boldsymbol{W} \right),$$

where $\boldsymbol{M}_{-k}^\star = \boldsymbol{M}^\star - \lambda_k^\star \boldsymbol{u}_k^\star \boldsymbol{u}_k^{\star\top}$. There are two differences between Algorithms 1 and 2. First, we remove Equation (16) from Algorithm 1, as this is mainly a technical requirement (see Remark 2).

More importantly, in Algorithm 2, we conjugate $\boldsymbol{Y}$ by a random orthogonal matrix $\boldsymbol{H} \in \mathbb{O}_n$. The $(r-1)$ leading eigenvectors of $\boldsymbol{H}\boldsymbol{M}^\star_{-k}\boldsymbol{H}^\top$ form a random subspace of $\mathbb{R}^{n \times (r-1)}$, which allows us to treat $\boldsymbol{H}\boldsymbol{M}^\star_{-k}\boldsymbol{H}^\top$ as a noise matrix. By way of illustration, consider the rank-2 case with $\boldsymbol{M}^\star_{-1} = \lambda^\star_2 \boldsymbol{u}^\star_2 \boldsymbol{u}^{\star\top}_2$. $\boldsymbol{H}\boldsymbol{u}^\star_2$ behaves similarly to a random vector drawn uniformly from $\mathbb{S}^{n-1}$. Therefore, one would expect $\boldsymbol{H}\boldsymbol{u}^\star_2 \boldsymbol{u}^{\star\top}_2 \boldsymbol{H}$ to behave similarly to a noise matrix $n^{-1}\boldsymbol{g}\boldsymbol{g}^\top$, where $\boldsymbol{g} \sim N(0, \boldsymbol{I}_n)$ (see Chapter 3 of [54]). As in the discussion after Lemma 2, we can find a set of indices $I_{\alpha_0}$ such that the signal in the corresponding entries of $\boldsymbol{H}\boldsymbol{u}^\star_1$ dominates the noise.

---

**Algorithm 2** Coherent optimal eigenvector estimation algorithm

---

**Input:** Observed matrix $\boldsymbol{Y} \in \mathbb{R}^{n \times n}$; $k$-th leading eigenvalue estimate $\widehat{\lambda}_k$. If $\widehat{\lambda}_k < 0$, set $\boldsymbol{Y} = -\boldsymbol{Y}$.
**Output:** $\widehat{\boldsymbol{u}}_k \in \mathbb{R}^n$
  1: Obtain the top-$r$ eigenvectors $\widetilde{\boldsymbol{U}}$ of $\boldsymbol{H}\boldsymbol{Y}\boldsymbol{H}^\top$, where $\boldsymbol{H} \in \mathbb{O}_n$ is Haar-distributed.
  2: Set $\boldsymbol{Q} = \operatorname{diag}\left(\operatorname{sgn}\left(\widetilde{\boldsymbol{U}}_{\cdot,k}\right)\right)$ and set $\widetilde{\boldsymbol{Y}} = \boldsymbol{Q}\boldsymbol{H}\boldsymbol{Y}\boldsymbol{H}^\top\boldsymbol{Q}$.
  3: Pick an $\alpha_0 \in \mathcal{A}$ such that for $\hat{I} := \left\{i : |\widetilde{U}_{i,k}| \geq \alpha_0\right\}$, $|\hat{I}| \geq 1/\alpha_0^2 \log^2 n$.
  4: Set $\widehat{S} = \left(\sum_{j,\ell \in \hat{I}} \widetilde{Y}_{j\ell}\right)^{1/2}$ and set $\widehat{v}_j = \sum_{\ell \in \hat{I}} \widetilde{Y}_{j\ell} \Big/ \left(\widehat{S}\sqrt{\widehat{\lambda}_k}\right)$ for $j \in [n]$.
  5: Let $\boldsymbol{U} = \boldsymbol{H}^\top\widetilde{\boldsymbol{U}}$. For $j \in [n]$, set $\widehat{u}_{k,j} = \begin{cases} U_{k,j} & \text{if } |U_{k,j}| \leq \left(\sigma/|\widehat{\lambda}_k|\right)\log n \\ \left(\boldsymbol{H}^\top\boldsymbol{Q}\widehat{\boldsymbol{v}}\right)_j & \text{otherwise.} \end{cases}$

---

As mentioned above, our experiments in Section 5.2 indicate that Algorithm 2 performs well. A proof of its performance, however, is more complicated than Theorem 1. The main difficulty arises from the fact that in Algorithm 2, we conjugate by a random orthogonal transformation. This ensures that when considering the large entries of one signal eigenvector, the other signal eigenvectors are ignorable. Unfortunately, this random orthogonal transformation breaks Assumption 2 and introduces complicated dependency structure, requiring a more careful analysis that we leave for future work.

## 4 Estimation lower bounds under $\ell_{2,\infty}$ distance

Theorem 1 demonstrates that for a rank-one signal, dependence of the estimation rate on $\mu$ can be removed when $|\lambda^\star| = \Omega(\sigma\sqrt{n \log n})$. In this regime, our result matches the lower bound in Equation (9) up to log-factors, making this the minimax lower bound under Assumption 2. As discussed in Remark 2, we expect Assumption 2 can be removed via a more careful analysis, rendering the lower bound in Equation (9) optimal under $|\lambda^\star| = \Omega(\sigma\sqrt{n \log n})$. On the other hand, as discussed in Section 1.2, the lower bound in Equation (9) is sub-optimal in the regime where $|\lambda^\star| = O(\sigma\sqrt{n})$. In what follows, we aim to improve upon Equation (9) by deriving metric entropy bounds [54] for rank-$r$ singular subspaces under the $\ell_{2,\infty}$ distance when $|\lambda^\star| = O(\sigma\sqrt{n})$.

Recall that for a semi-metric $\rho$ defined on a set $\mathbb{K}$, we may define the $\delta$-packing number $\mathcal{M}(\mathbb{K}, \rho, \delta)$ of $\mathbb{K}$ under $\rho$ (see Chapter 15 in [56]). The packing $\delta$-entropy, $\log \mathcal{M}(\mathbb{K}, \rho, \delta)$, captures the complexity of the space $\mathbb{K}$, and a lower bound on the $\delta$-entropy can be translated into a lower bound on the minimax estimation error rate. For a given $r \in [n]$ and $1 \leq \mu \leq n/r$, we consider the parameter set

$$\mathbb{K}(n, r, \sqrt{\mu r/n}) = \left\{\boldsymbol{U} \in \mathbb{R}^{n \times r} : \boldsymbol{U}^\top\boldsymbol{U} = \boldsymbol{I}_r, \ \|\boldsymbol{U}\|_{2,\infty} \leq \sqrt{\frac{r\mu}{n}}\right\}. \tag{20}$$

Below, we write $\mathbb{K}_{r,\mu}$ for $\mathbb{K}(n, r, \sqrt{\mu r/n})$. Lemma 4 lower bounds $\log \mathcal{M}(\mathbb{K}_{r,\mu}, d_{2,\infty}, \delta)$. We focus on the range $r/n \lesssim \delta^2 \lesssim \mu r/n$, as the lower bound in Equation (9) shows that $\delta^2 \ll r/n$ is not achievable when $|\lambda^\star| = O(\sigma\sqrt{n})$ and $\delta^2 \gg \mu r/n$ is achieved by the trivial all-zeros estimate.

**Lemma 4.** *Suppose that $n/\mu \geq \max\{4, r\}$ and $\mu \geq 12 \log(12n)$, and let $\delta > 0$ be such that*

$$\frac{c_0^2 r}{8e^2 n} \leq \delta^2 \leq \frac{c_0^2 \mu r}{96e^2 n \log(12n/\mu)}, \tag{21}$$

*where $c_0 > 0$ is a universal constant. Then when $n$ is sufficiently large,*

$$\log \mathcal{M}(\mathbb{K}_{r,\mu}, d_{2,\infty}, \delta) \gtrsim \frac{r^2}{\delta^2} \tag{22}$$

**Remark 5.** *Lemma 4 applies when $\log n \lesssim \mu \lesssim n/r$. A more careful analysis might relax the lower bound, but when $\mu \lesssim \log n$, any $U \in \mathbb{K}_{r,\mu}$ is nearly incoherent and Equation* (9) *is nearly optimal. An upper bound matching Lemma 4 up to log-factors can be found in the appendix.*

Lemma 4 implies an improved lower bound compared to Equation (9). Consider the parameter space

$$\Omega(\lambda^\star, \mu, r) = \{(\boldsymbol{\Lambda}^\star, \boldsymbol{U}^\star) : \boldsymbol{\Lambda}^\star = \lambda^\star \boldsymbol{I}_r, \boldsymbol{U}^\star \in \mathbb{K}_{r,\mu}\}.$$

Using the Yang-Barron method [59], we obtain a lower bound for eigenspace estimation in the rank-$r$ signal-plus-noise model for $|\lambda^\star| = O(\sigma\sqrt{n})$. A detailed proof can be found in the appendix.

**Theorem 2.** *Under Assumption 1 and the conditions of Lemma 4, for any $0 < \lambda^\star \le (6\sqrt{C_0})^{-1}\sigma\sqrt{n}$, where $C_0 > 0$ is a universal constant related to covering numbers of Grassmann manifolds, there is a universal constant $c > 0$ such that for all sufficiently large $n$,*

$$\inf_{\widehat{\boldsymbol{U}} \in \mathbb{R}^{n \times r}} \sup_{(\boldsymbol{\Lambda}^\star, \boldsymbol{U}^\star) \in \Omega(\lambda^\star, \mu, r)} \mathbb{E}_{\boldsymbol{\Lambda}^\star, \boldsymbol{U}^\star} d_{2,\infty}\left(\widehat{\boldsymbol{U}}, \boldsymbol{U}^\star\right) \ge c\left(\frac{\sigma\sqrt{r}}{\lambda^\star} \wedge \sqrt{\frac{\mu r}{n \log(n/\mu)}}\right).$$

**Remark 6.** *Theorem 2 removes the upper limit $\sqrt{r/n}$ from Equation* (9)*, suggesting that $\mu$ only comes to bear when $\lambda^\star \le \sigma\sqrt{n}$. This regime is not well studied, as the BBP transition [8] implies that spectral methods fail, but other algorithms might achieve our lower bound. For example, signal detection is possible if structure is present [6]. Unfortunately, any such algorithm is likely to be computationally expensive, given the general belief that no polynomial-time algorithm can succeed when $\lambda^\star \le \sigma\sqrt{n}$ [39, 9]. We leave further exploration of this small-$\lambda^\star$ regime to future work.*

**Remark 7.** *The parameter space $\Omega(\lambda^\star, \mu, r)$ considered in Theorem 2 contains only signal matrices with condition number $\kappa = 1$. In recent work, the authors have established lower bounds akin to Theorem 2 that show the role of condition number. A full accounting of the interplay between condition number and coherence is a promising area for future work.*

## 5 Numerical experiments

We turn to a brief experimental exploration of our theoretical results. All experiments were run in a distributed environment on commodity hardware without GPUs. In total, the experiments reported below used 3425 compute-hours. Mean memory usage was 3.5 GB, with a maximum of 11 GB.

### 5.1 Simulations for rank-one eigenspace estimation

We begin with the rank-one setting considered in Algorithm 1, in which we observe

$$\boldsymbol{Y} = \boldsymbol{M}^\star + \boldsymbol{W} = \lambda^\star \boldsymbol{u}^\star \boldsymbol{u}^{\star\top} + \boldsymbol{W},$$

and wish to recover $\boldsymbol{u}^\star \in \mathbb{S}^{n-1}$. We take $\lambda^\star = \sqrt{n \log n}$ in all experiments, matching the rate in Remark 1. We consider three distributions for the entries of $\boldsymbol{W}$: Gaussian, Laplacian and Rademacher, all scaled to have variance $\sigma^2 = 1$. We consider two approaches to generating $\boldsymbol{u}^\star$. In either case, we set a random entry of $\boldsymbol{u}^\star$ to be $a \in \{0.3, 0.55, 0.8\}$, then generate the remaining entries by either

1. drawing uniformly from $\sqrt{1 - a^2}\mathbb{S}^{n-2}$, or
2. drawing uniformly from $\{\pm 1\}^{n-1}$ then normalizing these to have $\ell_2$ norm $\sqrt{1 - a^2}$.

In both cases, $\|\boldsymbol{u}^\star\|_\infty = a$ and $\mu = a^2 n$ with high probability. We take $a = \Theta(1)$, since in finite samples, $Cn^{-1/2}\log n$ (which is nearly incoherent) is hard to discern from a constant (e.g., when $n = 20000$, $4n^{-1/2}\log n \approx 0.28$, nearly matching $a = 0.3$). Having generated $\boldsymbol{Y} = \boldsymbol{M}^\star + \boldsymbol{W}$, we estimate $\boldsymbol{u}^\star$ using both the spectral estimate $\boldsymbol{u}$ and Algorithm 1 and measure their estimation error under $d_\infty$. We report the mean of 20 independent trials for each combination of problem size $n$, magnitude $a$ and methods for generating $\boldsymbol{u}^\star$ and $\boldsymbol{W}$. We vary $n$ from 100 to 15100 in increments of 1000.

When running Algorithm 1, we use the debiased estimate $\widehat{\lambda}_c$ from Equation (14). This requires an estimate of $\sigma$, for which we use the plug-in estimator in Equation (17). We set $\beta = 0$, eliminating Equation (16). Similar to Algorithm 2, we conjugate $\boldsymbol{Y}$ by a random orthogonal matrix $\boldsymbol{H} \in \mathbb{O}_n$. We expect the top eigenvector $\widetilde{\boldsymbol{u}}$ of $\boldsymbol{H}\boldsymbol{Y}\boldsymbol{H}^\top$ to be approximately uniformly distributed on $\mathbb{S}^{n-1}$, as it is close to $\boldsymbol{H}\boldsymbol{u}^\star$ (see Theorem 2.1 in [44]). Thus, the median of the absolute values of $\widetilde{\boldsymbol{u}}$ should be $\Theta(n^{-1/2})$. Instead of selecting $\alpha_0$ according to Equation (15), we set $\alpha_0$ to be this median. The

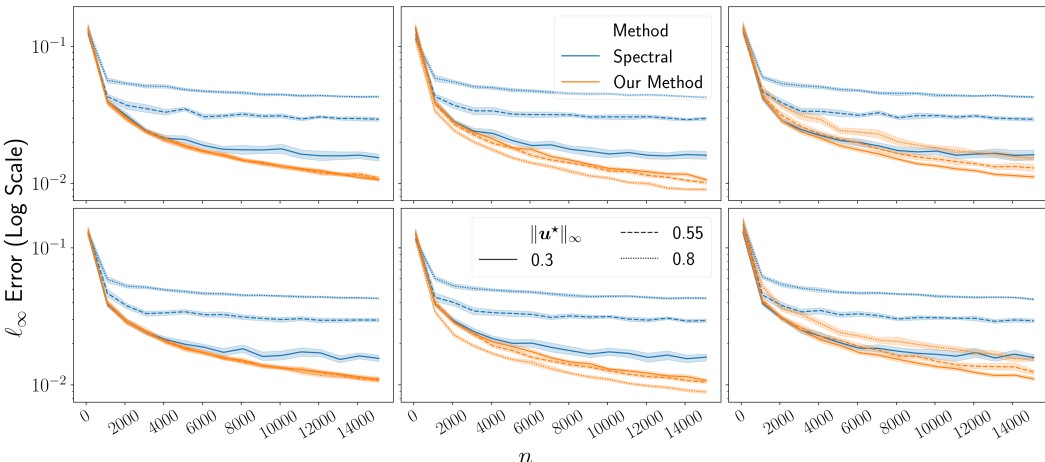

Figure 1: Estimation error measured in $d_\infty$ as a function of dimension $n$, by the leading eigenvector (blue) and Algorithm 1 (orange), for $\|\boldsymbol{u}^\star\|_\infty$ equal to $0.8, 0.55$ and $0.3$ (dotted, dashed and solid lines, respectively). We consider $\boldsymbol{u}^\star$ generated from the Bernoulli (top row) and Haar (bottom row) schemes, and we consider Gaussian (left), Rademacher (middle) and Laplacian (right) noise.

requirement in Equation (15) is then fulfilled, since there are roughly $n/2$ entries larger than the median. After obtaining $\widehat{\boldsymbol{u}}$ from Algorithm 1, we return $\boldsymbol{H}^\top \widehat{\boldsymbol{u}}$ as our estimate of $\boldsymbol{u}^\star$. We note that this random rotation further serves to illustrate that Assumption 2 is merely a technical requirement: after a random rotation, with high probability, $\boldsymbol{H}\boldsymbol{u}^\star$ does not satisfy Assumption 2.

Figure 1 compares the accuracy in estimating $\boldsymbol{u}^\star$ using the leading eigenvector of $\boldsymbol{Y}$ (blue) and Algorithm 1 (orange) under the three noise settings and two generating procedures for $\boldsymbol{u}^\star$. Shaded bands indicate 95% bootstrap confidence intervals (CIs). Across settings, Algorithm 1 recovers $\boldsymbol{u}^\star$ with a much smaller estimation error under $d_\infty$ compared to the naïve spectral estimate, especially when $\|\boldsymbol{u}^\star\|_\infty$ (i.e., the coherence $\mu$) is large. The spectral method degrades noticeably as coherence increases, while Algorithm 1 has far less dependence on $\mu$. Indeed, under Gaussian noise (the first column of Figure 1), it has no visible dependence on $\mu$. Under Rademacher noise (middle column of Figure 1), the dependence of Algorithm 1 on $\mu$ appears slightly reversed from that of the spectral estimator. Under Laplacian noise (right column of Figure 1), there seems to be a slight dependence on $\mu$. Further examination in the appendix suggests that this is due to estimating the entries *other* than the largest element of $\boldsymbol{u}^\star$, and is likely asymptotically smaller than the rate in Theorem 1.

### 5.2 Simulations for rank-$r$ eigenvector estimation

For the rank-$r$ setting, we have a signal matrix with eigenvalues $|\lambda_1^\star| \geq \ldots \geq |\lambda_r^\star|$. We observe
$$\boldsymbol{Y} = \boldsymbol{M}^\star + \boldsymbol{W} = \boldsymbol{U}^\star \boldsymbol{\Lambda}^\star \boldsymbol{U}^\star + \boldsymbol{W},$$
where $\boldsymbol{\Lambda}^\star = \operatorname{diag}(\lambda_1^\star, \ldots, \lambda_r^\star)$, and we wish to recover $\boldsymbol{U}^\star \in \mathbb{R}^{n \times r}$. We take $\sigma = 1$ and $\lambda_r^\star = \sqrt{n \log n}$. Recovering each column of $\boldsymbol{U}^\star$ separately requires an eigengap, defined for $k \in [r]$ as $\Delta_k := |\lambda_k^\star - \lambda_{k+1}^\star|$ and $\Delta_0 = \infty$. Typically, the estimation error of the $k$-th eigenvector has a $O(\min\{\Delta_k, \Delta_{k-1}\}^{-1})$ dependence on the eigengaps [57]. Here, we set $\Delta_k = 0.5\sqrt{n \log n}$, so $\lambda_k^\star = 0.5(r - k + 2)\sqrt{n \log n}$ for all $k \in [r]$. As in the rank-one case, we generate entries of $\boldsymbol{W}$ from Gaussian, Laplacian and Rademacher distributions, all scaled to have unit variance. The true eigenvectors $\boldsymbol{U}^\star$ are generated by repeating the following procedure for $k \in [r]$:

1. Randomly select an element of $\boldsymbol{v} \in \mathbb{R}^n$ and set it to be $a \in \{0.3, 0.55, 0.8\}$.
2. Generate the rest of $\boldsymbol{v}$ by drawing uniformly from $\sqrt{1 - a^2}\mathbb{S}^{n-2}$.
3. Set $\boldsymbol{u}_k^\star = \left(\boldsymbol{I}_n - \boldsymbol{U}_{\cdot,1:(k-1)}^\star \boldsymbol{U}_{\cdot,1:(k-1)}^{\star\top}\right)\boldsymbol{v}$, normalize $\boldsymbol{u}_k^\star$ to have unit $\ell_2$ norm, and set $\boldsymbol{U}_{\cdot,k}^\star = \boldsymbol{u}_k^\star$. If $k = 1$, then we take $\boldsymbol{U}_{\cdot,1:(k-1)}^\star \boldsymbol{U}_{\cdot,1:(k-1)}^{\star\top}$ to be the zero matrix.

Under this procedure, each $\boldsymbol{u}_k^\star$ has coherence approximately $a^2 n$. Since the large entries of the $\boldsymbol{u}_k^\star$ are unlikely to appear in the same rows, $\boldsymbol{U}^\star$ also has coherence $a^2 n$. We take $r = 2$ here.

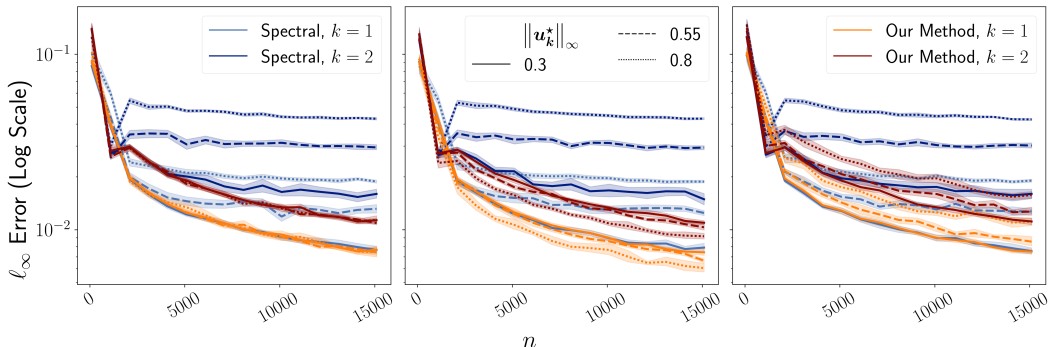

Figure 2: Estimation error under $d_\infty$ as a function of size $n$, by the $k$-th eigenvector (blue/purple) and the estimator in Algorithm 2 (orange/red) for $\|\boldsymbol{u}^\star\|_\infty$ equal to 0.8 (dotted lines), 0.55 (dashed lines) or 0.3 (solid lines) with Gaussian (left), Rademacher (center) or Laplacian (right) noise.

Having generated $\boldsymbol{Y} = \boldsymbol{M}^\star + \boldsymbol{W}$, we obtain estimates via the naïve spectral method and Algorithm 2. and measure their estimation error under $d_\infty$. We vary $n$ from 100 to 15100 in increments of 1000 and report the mean of 20 independent trials for each combination of $n$, $a$ and noise distribution. The results are summarized in Figure 2, showing the error for $\boldsymbol{u}_k^\star$, $k \in \{1, 2\}$ using the spectral estimate (blue/purple) and Algorithm 2 (orange/red), under Gaussian (left), Rademacher (middle) and Laplacian (right) noise. Shaded bands indicate 95% bootstrap CIs. In all settings, Algorithm 2 improves markedly on the spectral estimator. Algorithm 2 shows no visible dependence on $\mu$ under Gaussian noise. Under Rademacher noise, its $\mu$-dependence is the reverse of the spectral estimator. Under Laplacian noise, it shows slight dependence on $\mu$, which we again expect to be asymptotically smaller than the rate in Theorem 1. In the appendix, we consider $r = 3$ and measure error under $d_{2,\infty}$, and we again find that Algorithm 2 outperforms spectral estimators and is far less sensitive to $\mu$.

## 5.3 Comparison with other methods

To the best of our knowledge, we are the first paper to consider the task of non-spectral entrywise eigenvector estimation. The nearest obvious competing method might be based on approximate message passing (AMP; see [34] for an overview). Such a comparison is included in the appendix, where our experiments indicate that while AMP performs well in recovering the signal eigenvectors as measured by $\ell_2$ error, our method as specified in Algorithms 1 and 2 perform better under entrywise and $\ell_{2,\infty}$ error. Experimental details and further discussion can be found in the appendix.

## 6 Discussion, limitations and conclusion

We have presented new methods for eigenvector estimation in signal-plus-noise matrix models and new lower bounds for estimation rates in these models. The entrywise estimation error of our method has no dependence on the coherence $\mu$ for rank-one signal matrices, and achieves the optimal estimation rate up to log-factors. Simulations show that our method tolerates non-Gaussian noise and its extension to rank-$r$ signal matrices has little dependence on $\mu$. One limitation of our method is that it assumes homoscedastic noise. Future work will aim to relax this assumption and the technical condition in Assumption 2. In the rank-$r$ case, Algorithm 2 estimates each eigenvector separately, requiring an eigengap. Future work will avoid this by simultaneously or iteratively estimating multiple eigenvectors. We note in closing that inequitable social impacts from abuse or misuse of models and methods are common, but we see no particular such impacts in the present work.

## Acknowledgments and Disclosure of Funding

The authors gratefully acknowledge the support of the United States National Science Foundation via grants NSF DMS 2052918 and NSF DMS 2023239. KL was additionally supported by the University of Wisconsin–Madison Office of the Vice Chancellor for Research and Graduate Education with funding from the Wisconsin Alumni Research Foundation.

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

# A  Proof of Lemma 1

*Proof.* Let $\theta = \lim_{n \to \infty} (\sigma \sqrt{n})^{-1} \lambda^\star$. For any $\boldsymbol{u}^\star \in \mathbb{S}^{n-1}$, by the Baik-Ben Arous-Péché (BBP) phase transition ([8]; see also Theorem 2 in [38]), the top eigenvector $\boldsymbol{u}$ of $\boldsymbol{Y}$ obeys

$$\langle \boldsymbol{u}, \boldsymbol{u}^\star \rangle^2 \xrightarrow{\text{a.s.}} \begin{cases} 1 - \theta^{-2} & \text{if } \theta > 1 \\ 0 & \text{if } \theta \leqslant 1 \end{cases}. \tag{23}$$

Let $s = \lceil n/\mu \rceil$, and consider

$$\boldsymbol{u}^\star = \left( \frac{1}{\sqrt{s}} \boldsymbol{1}_s^\top, \boldsymbol{0}_{n-s}^\top \right)^\top \in \mathbb{R}^n. \tag{24}$$

Without loss of generality, we assume that $d_\infty(\boldsymbol{u}, \boldsymbol{u}^\star) = \|\boldsymbol{u} - \boldsymbol{u}^\star\|_\infty$, since otherwise we can repeat the following argument with $-\boldsymbol{u}$ instead of $\boldsymbol{u}$. We have

$$d_\infty(\boldsymbol{u}, \boldsymbol{u}^\star) \geq \max_{i \in [s]} |u_i - u_i^\star| \geq \frac{1}{\sqrt{s}} \left( \sum_{i=1}^s |u_i - u_i^\star|^2 \right)^{1/2}. \tag{25}$$

Expanding the term inside the square root,

$$\sum_{i=1}^s |u_i - u_i^\star|^2 = \sum_{i=1}^s u_i^2 + \sum_{i=1}^s u_i^{\star 2} - 2 \sum_{i=1}^s u_i^\star u_i = \sum_{i=1}^s u_i^2 + 1 - 2\langle \boldsymbol{u}^\star, \boldsymbol{u} \rangle, \tag{26}$$

where the last equality follows from the construction of $\boldsymbol{u}^\star$ in Equation (24). By the Cauchy-Schwarz inequality, we have

$$\sum_{i=1}^s u_i^2 \geq \frac{1}{s} \left( \sum_{i=1}^s u_i \right)^2 = \left( \sum_{i=1}^s \frac{1}{\sqrt{s}} u_i \right)^2 = \langle \boldsymbol{u}^\star, \boldsymbol{u} \rangle^2, \tag{27}$$

where the last equality follows from the construction of $\boldsymbol{u}^\star$ in Equation (24). Plugging Equation (27) into Equation (26), we obtain

$$\sum_{i=1}^s |u_i - u_i^\star|^2 \geq \langle \boldsymbol{u}^\star, \boldsymbol{u} \rangle^2 + 1 - 2\langle \boldsymbol{u}^\star, \boldsymbol{u} \rangle = \left( 1 - \langle \boldsymbol{u}^\star, \boldsymbol{u} \rangle \right)^2.$$

When $\theta > 1$, applying Equation (23) yields that

$$\liminf_{n \to \infty} \sum_{i=1}^s |u_i - u_i^\star|^2 \geq \liminf_{n \to \infty} \left( 1 - \langle \boldsymbol{u}^\star, \boldsymbol{u} \rangle \right)^2 = \left( 1 - \sqrt{1 - \frac{1}{\theta^2}} \right)^2 \tag{28}$$

holds almost surely. Again using the fact that $\theta > 1$, we have

$$1 - \sqrt{1 - \frac{1}{\theta^2}} = \left( 1 + \sqrt{1 - \frac{1}{\theta^2}} \right)^{-1} \left[ 1 - \left( 1 - \frac{1}{\theta^2} \right) \right] \geq \frac{1}{2\theta^2}. \tag{29}$$

Combining Equations (25), (28) and (29), it holds almost surely that

$$\liminf_{n \to \infty} d_\infty(\boldsymbol{u}, \boldsymbol{u}^\star) \geq \liminf_{n \to \infty} \frac{1}{\sqrt{s}} \left( |u_i - u_i^\star|^2 \right)^{1/2} \geq \lim_{n \to \infty} \frac{1}{2\theta^2} \cdot \frac{1}{\sqrt{1 + n/\mu}},$$

where the last inequality follows from $s = \lceil n/\mu \rceil$. Since $\mu \leq n$, it follows that

$$\liminf_{n \to \infty} d_\infty(\boldsymbol{u}, \boldsymbol{u}^\star) \geq \lim_{n \to \infty} \frac{1}{2\theta^2} \cdot \sqrt{\frac{\mu}{\mu + n}} \geq \lim_{n \to \infty} \frac{1}{2\sqrt{2}\,\theta^2} \cdot \sqrt{\frac{\mu}{n}}$$

almost surely. By the definition of $\theta$, we have

$$\liminf_{n \to \infty} d_\infty(\boldsymbol{u}, \boldsymbol{u}^\star) \geq \lim_{n \to \infty} \frac{\sigma^2 n}{2\sqrt{2}|\lambda^\star|^2} \sqrt{\frac{\mu}{n}} = \lim_{n \to \infty} \frac{\sigma^2 \sqrt{n\mu}}{2\sqrt{2}|\lambda^\star|^2},$$

completing the proof. $\qquad \square$

# B   Proof of Lemma 2

*Proof.* The upper bound

$$\left|\{i : |v_i| \geq \alpha_0\}\right| \leq \frac{1}{\alpha_0^2}$$

follows trivially from the fact that $\|v\|_2 = 1$.

To prove the lower bound, we consider the disjoint intervals

$$\mathscr{I}_\ell = \begin{cases} \left[\log^{-1/2} n, 1\right] & \text{if } \ell = 0 \\ \left[\log^{-(\ell+1)/2} n, \log^{-\ell/2} n\right) & \text{if } \ell \in \{1, 2, \ldots, \lceil L \rceil - 2\} \\ \left[\log^{-L/2} n, \log^{-(\lceil L \rceil - 1)/2} n\right) & \text{if } \ell = \lceil L \rceil - 1. \end{cases} \tag{30}$$

Recall the definition of $L$ from Equation (11),

$$L = \frac{\log(2n)}{\log \log n}.$$

Rearranging terms, we have

$$-\frac{L}{2} \log \log n = -\frac{1}{2} \log(2n).$$

Exponentiating both sides of the above display,

$$\log^{-L/2} n = \frac{1}{\sqrt{2n}}.$$

Noting that by Equation (30),

$$\bigcup_{\ell=0}^{\lceil L \rceil - 1} \mathscr{I}_\ell = \left[\log^{-L/2} n, 1\right] = \left[\frac{1}{\sqrt{2n}}, 1\right],$$

for any $i \in [n]$ such that $|v_i| \notin \cup_{\ell=0}^{\lceil L \rceil - 1} \mathscr{I}_\ell$, we have

$$|v_i| < \log^{-L/2} n = \frac{1}{\sqrt{2n}},$$

and therefore, summing over all $i \in [n]$ such that $|v_i| \notin \bigcup_{\ell=0}^{\lceil L \rceil - 1} \mathscr{I}_\ell$ yields that

$$\sum_{i:|v_i| \notin \cup_{\ell=0}^{\lceil L \rceil - 1} \mathscr{I}_\ell} v_i^2 < \sum_{i:|v_i| \notin \cup_{\ell=0}^{\lceil L \rceil - 1} \mathscr{I}_\ell} \frac{1}{2n} \leq n \cdot \frac{1}{2n} = \frac{1}{2}. \tag{31}$$

Let $x_\ell = \left|\{i : |v_i| \in \mathscr{I}_\ell\}\right|$ for all $\ell \in \{0, 1, \ldots, \lceil L \rceil - 1\}$. We have

$$\sum_{i:|v_i| \in \mathscr{I}_\ell} v_i^2 \leq x_\ell \max_{i:|v_i| \in \mathscr{I}_\ell} v_i^2 \leq \frac{x_\ell}{\log^\ell n},$$

where the last inequality follows from the definition of $\mathscr{I}_\ell$ in Equation (30). Summing over all $\ell \in \{0, 1, \ldots, \lceil L \rceil - 1\}$ on both sides , we have

$$\sum_{\ell=0}^{\lceil L \rceil - 1} \frac{x_\ell}{\log^\ell n} \geq \sum_{i:|v_i| \in \cup_{\ell=0}^{\lceil L \rceil - 1} \mathscr{I}_\ell} v_i^2 = 1 - \sum_{i:|v_i| \notin \cup_{\ell=0}^{\lceil L \rceil - 1} \mathscr{I}_\ell} v_i^2 > \frac{1}{2},$$

where the last inequality follows from Equation (31). The above further implies that

$$\lceil L \rceil \cdot \max_{\ell \in \{0, 1, \cdots, \lceil L \rceil - 1\}} \left\{\frac{x_\ell}{\log^\ell n}\right\} \geq \sum_{\ell=0}^{\lceil L \rceil - 1} \frac{x_\ell}{\log^\ell n} > \frac{1}{2}.$$

Hence, rearranging, there exists an $\ell_0 \in \{0, 1, \cdots, \lceil L \rceil - 1\}$ such that

$$x_{\ell_0} > \frac{\log^{\ell_0} n}{2 \lceil L \rceil} > \log^{\ell_0 - 1} n = \frac{\log^{\ell_0 + 1} n}{\log^2 n},$$

where the second inequality holds for suitably large $n$ by the definition of $L$ given in Equation (11).

Finally, note that for all $i \in [n]$ such that $|v_i| \in \mathscr{I}_{\ell_0}$, we have $|v_i| \geq \log^{-(\ell_0+1)/2} n$ by the definition of $\mathscr{I}_{\ell_0}$ in Equation (30). Taking

$$\alpha_0 = \begin{cases} \log^{-(\ell_0+1)/2} n & \text{if } \ell_0 \in \{0, 1, \cdots, \lceil L \rceil - 2\} \\ \log^{-L/2} n & \text{if } \ell_0 = \lceil L \rceil - 1 \end{cases}$$

yields that

$$|\{i : |v_i| \geq \alpha_0\}| \geq x_{\ell_0} > \frac{1}{\alpha_0^2 \log^2 n}.$$

We complete the proof by noting that $\alpha_0$ is in $\mathcal{A}$, where $\mathcal{A}$ is defined in Equation (10). $\qquad \square$

## C  Proof of Lemma 3

*Proof.* Expanding the right hand side of Equation (17), we have

$$\widehat{\sigma}^2 = \frac{2}{n(n+1)} \sum_{1 \leq i \leq j \leq n} \left( M_{ij}^\star - \widehat{M}_{ij} + W_{ij} \right)^2$$

$$= \frac{2}{n(n+1)} \sum_{1 \leq i \leq j \leq n} \left( M_{ij}^\star - \widehat{M}_{ij} \right)^2 + \frac{2}{n(n+1)} \sum_{1 \leq i \leq j \leq n} W_{ij}^2$$

$$+ \frac{4}{n(n+1)} \sum_{1 \leq i \leq j \leq n} W_{ij} \left( M_{ij}^\star - \widehat{M}_{ij} \right).$$

Subtracting $\sigma^2$ on both sides and applying the triangle inequality, we have

$$\left| \widehat{\sigma}^2 - \sigma^2 \right| \leq \frac{2}{n(n+1)} \sum_{1 \leq i \leq j \leq n} \left( M_{ij}^\star - \widehat{M}_{ij} \right)^2 + \left| \frac{2}{n(n+1)} \sum_{1 \leq i \leq j \leq n} W_{ij}^2 - \sigma^2 \right|$$

$$+ \left| \frac{4}{n(n+1)} \sum_{1 \leq i \leq j \leq n} W_{ij} \left( M_{ij}^\star - \widehat{M}_{ij} \right) \right|$$

$$\leq \frac{2}{n(n+1)} \left\| M^\star - \widehat{M} \right\|_{\mathrm{F}}^2 + \left| \frac{2}{n(n+1)} \sum_{1 \leq i \leq j \leq n} W_{ij}^2 - \sigma^2 \right|$$

$$+ \left| \frac{2}{n(n+1)} \operatorname{tr} W \left( M^\star - \widehat{M} \right) \right| + \frac{2}{n(n+1)} \left| \sum_{i=1}^n W_{ii} \left( M_{ii}^\star - \widehat{M}_{ii} \right) \right|. \tag{32}$$

We proceed to bound each term in the last inequality of Equation (32) separately.

For the first term in Equation (32), by an argument analogous to that of Equation (3.15) in [27],

$$\frac{2}{n(n+1)} \left\| \widehat{M} - M^\star \right\|_{\mathrm{F}}^2 \leq \frac{8r}{n(n+1)} \| W \|^2.$$

By standard matrix concentration inequalities [53, 54], with probability at least $1 - O(n^{-8})$,

$$\| W \| \leq 5 \sqrt{\nu_W n \log n}. \tag{33}$$

Combining the above two displays, we obtain an analogue of Equation (3.16) in [27], namely that with probability at least $1 - O(n^{-8})$,

$$\frac{2}{n(n+1)} \left\| \widehat{M} - M^\star \right\|_{\mathrm{F}}^2 \leq \frac{200 \nu_W r}{n+1}. \tag{34}$$

For the second term in Equation (32), applying standard concentration inequalities for subexponential random variables (see, e.g., Chapter 2 in [54]),

$$\mathbb{P}\left[\left|\sum_{1 \le i \le j \le n} \left(W_{ij}^2 - \sigma^2\right)\right| \ge t\right] \le 2\exp\left\{-c^2 \min\left\{\frac{2t^2}{n(n+1)\nu_W^2}, \frac{t}{\nu_W}\right\}\right\},$$

where $c > 0$ is an absolute constant. Taking $t = 2\nu_W\sqrt{n(n+1)\log n}/c$ and dividing through by $n(n+1)/2$, it holds with probability at least $1 - O(n^{-8})$ that

$$\left|\frac{2}{n(n+1)} \sum_{1 \le i \le j \le n} W_{ij}^2 - \sigma^2\right| \le \frac{4\nu_W\sqrt{\log n}}{cn}. \tag{35}$$

For the third term in Equation (32), using the matrix Hölder inequality (see Corollary IV.2.6 in [15]), we have

$$\left|\operatorname{tr} \boldsymbol{W}\left(\boldsymbol{M}^\star - \widehat{\boldsymbol{M}}\right)\right| \le \|\boldsymbol{W}\|\|\boldsymbol{M}^\star - \widehat{\boldsymbol{M}}\|_*,$$

where $\|\cdot\|_*$ denotes the nuclear norm, defined to be the sum of the singular values. Using the fact that $\boldsymbol{M}^\star - \widehat{\boldsymbol{M}}$ has at most rank $2r$, we have

$$\left|\operatorname{tr} \boldsymbol{W}\left(\boldsymbol{M}^\star - \widehat{\boldsymbol{M}}\right)\right| \le 2r\|\boldsymbol{W}\|\|\boldsymbol{M}^\star - \widehat{\boldsymbol{M}}\|.$$

Applying Equation (33) and following an argument analogous to that in Equations (3.12) and (3.15) in [27], we have that with probability at least $1 - O(n^{-8})$,

$$\|\boldsymbol{W}\| \le 5\sqrt{\nu_W n} \quad \text{and} \quad \|\boldsymbol{M}^\star - \widehat{\boldsymbol{M}}\| \le 10\sqrt{\nu_W n}.$$

It follows that with probability at least $1 - O(n^{-8})$,

$$\frac{2}{n(n+1)}\left|\operatorname{tr} \boldsymbol{W}\left(\boldsymbol{M}^\star - \widehat{\boldsymbol{M}}\right)\right| \le \frac{200\nu_W r}{n+1}. \tag{36}$$

For the fourth term in Equation (32), we have

$$\left|\sum_{i=1}^n W_{ii}\left(M_{ii}^\star - \widehat{M}_{ii}\right)\right| \le \max_{i \in [n]}\{|W_{ii}|\}\|\widehat{\boldsymbol{M}} - \boldsymbol{M}^\star\|_*.$$

$$\frac{2}{n(n+1)}\left|\sum_{i=1}^n W_{ii}\left(M_{ii}^\star - \widehat{M}_{ii}\right)\right| \le \frac{200\nu_W r\sqrt{\log n}}{n^{3/2}}. \tag{37}$$

Applying Equations (34), (35), (36) and (37) to Equation (32), with probability at least $1 - O(n^{-8})$,

$$|\widehat{\sigma}^2 - \sigma^2| \le \frac{400\nu_W r}{n} + \frac{4\nu_W\sqrt{\log n}}{cn} + \frac{200\nu_W r\sqrt{\log n}}{n^{3/2}},$$

as we set out to show. $\qquad\qquad\qquad\qquad\qquad\qquad\qquad\qquad\qquad\qquad\qquad\qquad\qquad\square$

## D   Proof of Theorem 1

Without loss of generality, we derive our results under the assumption that

$$d_\infty\left(\boldsymbol{u}, \boldsymbol{u}^\star\right) = \|\boldsymbol{u} - \boldsymbol{u}^\star\|_\infty.$$

Otherwise, we can replace $\boldsymbol{u}$ with $-\boldsymbol{u}$ and obtain the same results. We remind the reader that we use $c$ and $C$ to denote constants with respect to $n$, whose precise value might change from line to line. To prove Theorem 1, we first state and prove a few technical lemmas.

## D.1 Technical lemmas

**Lemma 5.** *If $|\lambda^\star| \geq 80\sqrt{\nu_W n}$, then under Assumption 1, it holds with probability at least $1 - O(n^{-8})$ that for all $\ell \in [n]$,*

$$\min\left\{|u_\ell - u_\ell^\star|, |u_\ell + u_\ell^\star|\right\} \leq \frac{80\sqrt{\nu_W \log n} + 120\sqrt{\nu_W n}|u_\ell^\star|}{|\lambda^\star|}. \tag{38}$$

*Proof.* We largely follow the proof of Theorem 4.1 in [27], albeit with a slightly more careful analysis. In particular, we note that the proof of Theorem 4.1 in [27] is written for the case where $\boldsymbol{W}$ has Gaussian entries. It is straightforward to extend this argument to subgaussian entries by applying standard subgaussian concentration inequalities and truncation arguments [54, 56], in essence replacing all appearances of $\sigma^2$ in their results with the subgaussian parameter (i.e., "variance proxy") $\nu_W$. Here, we only sketch out a few key points and refer the reader to Section 4.1.4 of [27] for a full argument. For each $\ell \in [n]$, we construct a leave-one-out copy $\boldsymbol{Y}^{(\ell)}$ as

$$\boldsymbol{Y}^{(\ell)} = \lambda^\star \boldsymbol{u}^\star \boldsymbol{u}^{\star\top} + \boldsymbol{W}^{(\ell)},$$

where

$$W_{ij}^{(\ell)} = \begin{cases} W_{ij}, & \text{if } \ell \notin \{i, j\} \\ 0 & \text{otherwise.} \end{cases}$$

For each $\ell \in [n]$, let $\lambda^{(\ell)}$ and $\boldsymbol{u}^{(\ell)}$ denote, respectively, the largest-magnitude eigenvalue of $\boldsymbol{Y}^{(\ell)}$ and its corresponding eigenvector. Without loss of generality, flipping signs if necessary, we assume that

$$d_\infty(\boldsymbol{u}, \boldsymbol{u}^\star) = \|\boldsymbol{u} - \boldsymbol{u}^\star\|_\infty.$$

By Equation (4.20) in [27], with probability at least $1 - O(n^{-8})$ we have

$$\left|u_\ell^{(\ell)} - u_\ell^\star\right| \leq \frac{20\sqrt{\nu_W n}}{|\lambda^\star|}|u_\ell^\star|. \tag{39}$$

By Equation (4.15) in [27], we have

$$\left\|\boldsymbol{u} - \boldsymbol{u}^{(\ell)}\right\|_2 \leq \frac{4\left\|\left(\boldsymbol{Y} - \boldsymbol{Y}^{(\ell)}\right)\boldsymbol{u}^{(\ell)}\right\|_2}{|\lambda^\star|}.$$

Following the argument just after Equation (4.18) in [27], replacing the Gaussian concentration inequalities with subgaussian concentration inequalities, one can show that

$$\left\|\left(\boldsymbol{Y} - \boldsymbol{Y}^{(\ell)}\right)\boldsymbol{u}^{(\ell)}\right\|_2 \leq 5\sqrt{\nu_W \log n} + 5\sqrt{\nu_W n}\,|u_\ell| + 5\sqrt{\nu_W n}\left\|\boldsymbol{u} - \boldsymbol{u}^{(\ell)}\right\|_2,$$

with probability at least $1 - O(n^{-8})$. Combining the above two displays and rearranging slightly,

$$\left(1 - \frac{5\sqrt{\nu_W n}}{|\lambda^\star|}\right)\|\boldsymbol{u} - \boldsymbol{u}^{(\ell)}\|_2 \leq \frac{5\sqrt{\nu_W \log n} + 5\sqrt{\nu_W n}\,|u_\ell|}{|\lambda^\star|}.$$

Since $\lambda^\star \geq 40\sqrt{\nu_W n}$ by assumption, it follows that

$$\|\boldsymbol{u} - \boldsymbol{u}^{(\ell)}\|_2 \leq \frac{40\sqrt{\nu_W \log n} + 40\sqrt{\nu_W n}\,|u_\ell|}{|\lambda^\star|}. \tag{40}$$

Combining the triangle inequality with the trivial upper bound $\|\boldsymbol{u} - \boldsymbol{u}^{(\ell)}\|_\infty \leq \|\boldsymbol{u} - \boldsymbol{u}^{(\ell)}\|_2$, we have for any $\ell \in [n]$,

$$|u_\ell - u_\ell^\star| \leq \left|u_\ell^\star - u_\ell^{(\ell)}\right| + \left\|\boldsymbol{u} - \boldsymbol{u}^{(\ell)}\right\|_2.$$

Applying Equations (39) and (40), it holds with probability at least $1 - O(n^{-8})$ that

$$|u_\ell - u_\ell^\star| \leq \frac{20\sqrt{\nu_W n}}{|\lambda^\star|}|u_\ell^\star| + \frac{40\sqrt{\nu_W \log n} + 40\sqrt{\nu_W n}\,|u_\ell|}{|\lambda^\star|}$$

$$\leq \frac{20\sqrt{\nu_W n}}{|\lambda^\star|}|u_\ell^\star| + \frac{40\sqrt{\nu_W \log n} + 40\sqrt{\nu_W n}|u_\ell^\star| + 40\sqrt{\nu_W n}\,|u_\ell - u_\ell^\star|}{|\lambda^\star|},$$

where the second inequality follows from the triangle inequality. After rearranging terms and using the fact that $|\lambda^\star| \geq 80\sqrt{\nu_W n}$, we obtain the desired bound

$$|u_\ell - u_\ell^\star| \leq \frac{80\sqrt{\nu_W \log n} + 120\sqrt{\nu_W n}|u_\ell^\star|}{|\lambda^\star|},$$

for all $\ell \in [n]$, completing the proof. $\qquad\square$

Lemma 6, which controls the behavior of the index set $\hat{I}$ in Algorithm 1, follows from an application of Lemma 5.

**Lemma 6.** *Under the model in Equation* (1) *with $\boldsymbol{M}^\star = \lambda^\star u^\star u^{\star\top}$, suppose that Assumption 1 holds and Assumption 3 holds with constant $C_1 > 2400/\epsilon_0$, where $\epsilon_0 \in (0, 1)$ is fixed. For $\alpha \in \mathcal{A}$, define*

$$\hat{I}_\alpha = \{i : |u_i| \geq \alpha\},$$

*where $\boldsymbol{u}$ is the leading eigenvector of $\boldsymbol{Y}$, as in Step 1 of Algorithm 1. For all sufficiently large $n$, the following all hold with probability at least $1 - O(n^{-8} \log n)$:*

$$\{i : |u_i^\star| \geq (1 + \epsilon_0/2)\alpha\} \subseteq \hat{I}_\alpha \subseteq \{i : |u_i^\star| \geq (1 - \epsilon_0/2)\alpha\}, \tag{41}$$

$$\{i : (1 - \epsilon_0/2)\alpha < |u_i| < (1 + \epsilon_0/2)\alpha\} \subseteq \{i : (1 - \epsilon_0)\alpha < |u_i^\star| < (1 + \epsilon_0)\alpha\}, \tag{42}$$

$$\{i : |u_i^\star| \geq \alpha\} \subseteq \{i : |u_i| > (1 - \epsilon_0/2)\alpha\} \tag{43}$$

*and*

$$\{i : |u_i| \geq (1 + \epsilon_0/2)\alpha\} \subseteq \{i : |u_i^\star| \geq \alpha\}. \tag{44}$$

*Proof.* Fix $\alpha \in \mathcal{A}$. For any $i \in \hat{I}_\alpha$, we have $|u_i| \geq \alpha$ by definition. We have

$$\mathbb{P}\left( \bigcup_{i \in \hat{I}_\alpha} \left\{ |u_i^\star| < \alpha - \frac{80\sqrt{\nu_W \log n} + 120\sqrt{\nu_W n}|u_i^\star|}{|\lambda^\star|} \right\} \right)$$

$$\leq \mathbb{P}\left( \bigcup_{i \in \hat{I}_\alpha} \left\{ |u_i^\star| < |u_i| - \frac{80\sqrt{\nu_W \log n} + 120\sqrt{\nu_W n}|u_i^\star|}{|\lambda^\star|} \right\} \right) \tag{45}$$

$$\leq \mathbb{P}\left( \bigcup_{i \in [n]} \left\{ |u_i^\star| < |u_i| - \frac{80\sqrt{\nu_W \log n} + 120\sqrt{\nu_W n}|u_i^\star|}{|\lambda^\star|} \right\} \right) \leq O(n^{-8}).$$

where the first inequality follows from the fact that $|u_i| \geq \alpha$ for $i \in \hat{I}_\alpha$, the second inequality follows from set inclusion, and the last inequality follows from combining the triangle inequality with Lemma 5. If

$$|u_i^\star| \geq \alpha - \frac{80\sqrt{\nu_W \log n} + 120\sqrt{\nu_W n}|u_i^\star|}{|\lambda^\star|} \tag{46}$$

and we take $C_1 > 2400/\epsilon_0$ in Assumption 3, then we have

$$|u_i^\star| \geq \alpha - \frac{80\epsilon_0\sqrt{\nu_W \log n}}{2400\sqrt{\nu_W n \log n}} - \frac{120\epsilon_0\sqrt{\nu_W n}}{2400\sqrt{\nu_W n \log n}}|u_i^\star| \geq \alpha - \frac{\epsilon_0}{30}\sqrt{\frac{1}{n}} - \frac{\epsilon_0}{20\sqrt{\log n}}|u_i^\star|.$$

Since $\alpha \geq \sqrt{1/2n}$, after rearranging terms,

$$|u_i^\star| \geq \left(1 + \frac{\epsilon_0}{20\sqrt{\log n}}\right)^{-1}\left(\alpha - \frac{\epsilon_0}{30}\sqrt{\frac{1}{n}}\right) \geq \left(1 + \frac{\epsilon_0}{20\sqrt{\log n}}\right)^{-1}\left(1 - \frac{\epsilon_0}{15}\right)\alpha \tag{47}$$

$$> (1 - \epsilon_0/2)\alpha,$$

where the last inequality holds for all $i \in [n]$ satisfying Equation (46) (i.e., all $i \in \hat{I}_\alpha$) when $n$ is sufficiently large. Thus, we conclude that, applying a union bound followed by Equation (45),

$$
\begin{aligned}
\mathbb{P}\left(\hat{I}_\alpha \subseteq \{i : |u_i^\star| \geq (1-\epsilon_0/2)\alpha\}\right) &= \mathbb{P}\left(\bigcap_{i \in \hat{I}_\alpha} \{|u_i^\star| \geq (1 - \epsilon_0/2)\alpha\}\right) \\
&\geq 1 - \mathbb{P}\left(\bigcup_{i \in \hat{I}_\alpha} \left\{|u_i^\star| < \alpha - \frac{80\sqrt{\nu_W \log n} + 120\sqrt{\nu_W n}|u_i^\star|}{|\lambda^\star|}\right\}\right) \\
&\geq 1 - O(n^{-8}).
\end{aligned}
\tag{48}
$$

Following a similar argument, we can show that Equation (44) holds with probability at least $1 - O(n^{-8})$, and that, also with probability at least $1 - O(n^{-8})$,

$$
\{i : (1 - \epsilon_0/2)\,\alpha < |u_i|\} \subseteq \{i : (1 - \epsilon_0)\alpha < |u_i^\star|\}.
\tag{49}
$$

On the other hand, combining the triangle inequality with the bound in Equation (38) in Lemma 5, it holds with probability at least $1 - O(n^{-8})$ that for all $i \in [n]$ satisfying $|u_i^\star| \geq (1 + \epsilon_0/2)\alpha$,

$$
|u_i| \geq |u_i^\star| - |u_i - u_i^\star| \geq \left(1 + \frac{\epsilon_0}{2}\right)\alpha - \frac{\epsilon_0}{30\sqrt{n}} - \frac{\epsilon_0(1 + \epsilon_0/2)\alpha}{20\sqrt{\log n}} > \alpha,
$$

where the last inequality holds when $n$ is sufficiently large. It follows that

$$
\begin{aligned}
\mathbb{P}\left(\hat{I}_\alpha \supseteq \{i : |u_i^\star| \geq (1 + \epsilon_0/2)\alpha\}\right) &= \mathbb{P}\left(\bigcap_{i \in \{i:|u_i^\star| \geq (1+\epsilon_0/2)\alpha\}} \{|u_i| \geq \alpha\}\right) \\
&\geq 1 - O(n^{-8}).
\end{aligned}
\tag{50}
$$

Following the same argument, we have that Equation (43) holds with probability at least $1 - O(n^{-8})$ and that

$$
\{i : |u_i^\star| \geq (1 + \epsilon_0)\alpha\} \subseteq \{i : |u_i| \geq (1 + \epsilon_0/2)\alpha\}
\tag{51}
$$

also with probability at least $1 - O(n^{-8})$.

Combining Equation (48) and Equation (50) yields that Equation (41) holds with probability at least $1 - O(n^{-8})$. Similarly, combining Equations (49) and (51) implies that Equation (42) holds with probability at least $1 - O(n^{-8})$. A union bound over all $\alpha \in \mathcal{A}$ yields that Equations (41), (42) (43) and (44) hold simultaneously for all $\alpha \in \mathcal{A}$ with probability at least $1 - O(|\mathcal{A}|n^{-8})$. Recalling from Equation (10) that $|\mathcal{A}| = O(\log n)$ completes the proof. $\qquad\square$

The results in Lemma 6 lead to Lemma 7.

**Lemma 7.** *Consider the model in Equation* (1) *with $M^\star = \lambda^\star u^\star u^{\star\top}$ and suppose that Assumptions 1, 2 and 3 hold. For each $\alpha \in \mathcal{A}$, let*

$$
I_\alpha := \{i : |u_i^\star| \geq \alpha\}
\tag{52}
$$

*and define the set*

$$
\mathcal{A}_0 = \{\alpha \in \mathcal{A} : |I_\alpha| \geq (\alpha^2 \log^2 n)^{-1}\},
\tag{53}
$$

*where $\mathcal{A}$ is as defined in Equation* (10). *Let $\hat{I}$ be the set constructed in Step 2 of Algorithm 1 with $\beta = \epsilon_0/2$. With probability at least $1 - O(n^{-8} \log n)$, there exists an $\alpha \in \mathcal{A}_0$ such that $\hat{I} = I_\alpha$.*

*Proof.* We first show that with probability at least $1 - O(n^{-8} \log n)$, for any $\alpha \in \mathcal{A}$ satisfying Equations (15) and (16), we have $\alpha \in \mathcal{A}_0$ and $\hat{I}_\alpha = I_\alpha$, where $\hat{I}_\alpha$ is as defined in Lemma 6. We then prove that there exists at least one $\alpha \in \mathcal{A}_0 \subseteq \mathcal{A}$ such that both Equations (15) and (16) hold. Having established this, the value $\hat{\alpha} \in \mathcal{A}$ chosen in Step 2 of Algorithm 1, which satisfies Equations (15) and (16) by definition, must be an element of $\mathcal{A}_0 \subseteq \mathcal{A}$. Since $\hat{I} = \hat{I}_{\hat{\alpha}}$ by definition, we must have $\hat{I} = I_{\hat{\alpha}}$ with probability at least $1 - O(n^{-8} \log n)$, which will complete the proof.

Suppose that $\alpha \in \mathcal{A}$ satisfies Equations (15) and (16). Trivially, by the definition of $\mathcal{A}_0$ in Equation (53), Equation (15) implies that $\alpha \in \mathcal{A}_0$. By Lemma 6, with probability at least $1 - O(n^{-8} \log n)$, Equations (43) and (44) hold for all elements of $\mathcal{A}$. Thus, in particular,

$$\{i : |u_i| \geq (1 + \epsilon_0/2)\widehat{\alpha}\} \subseteq \{i : |u_i^\star| \geq \widehat{\alpha}\} \subseteq \{i : |u_i| > (1 - \epsilon_0/2)\widehat{\alpha}\}.$$

By Equation (16) and the choice $\beta = \epsilon_0/2$ in Algorithm 1,

$$\hat{I}_\alpha = \{i : |u_i| > (1 - \epsilon_0/2)\alpha\} = \{i : |u_i| \geq (1 + \epsilon_0/2)\alpha\},$$

from which it follows that $\hat{I}_\alpha = I_\alpha$ with probability at least $1 - O(n^{-8} \log n)$.

We now establish the existence of at least one $\alpha \in \mathcal{A}_0 \subseteq \mathcal{A}$ satisfying Equations (15) and (16). By Assumption 2, for sufficiently large $n$, there exists $\alpha \in \mathcal{A}$ such that Equation (12) holds, from which $\alpha \in \mathcal{A}_0$ immediately. Also by Assumption 2, there are no $i \in [n]$ for which

$$(1 - \epsilon_0)\alpha \leq |u_i^\star| \leq (1 + \epsilon_0)\alpha.$$

By Lemma 6, Equation (41) holds with probability at least $1 - O(n^{-8} \log n)$ for all elements of $\mathcal{A}_0 \subseteq \mathcal{A}$, and thus $\hat{I}_\alpha = I_\alpha$. Therefore, we have $|\hat{I}_\alpha| = |I_\alpha| \geq (\alpha^2 \log^2 n)^{-1}$, where the lower-bound follows from Equation (12). As a result, any such $\alpha$ guaranteed by Assumption 2 satisfies Equation (15).

By Equation (42) in Lemma 6 and the fact that $\alpha$ satisfies Assumption 2, with probability at least $1 - O(n^{-8} \log n)$ we have

$$\{i : (1 - \epsilon_0/2)\alpha < |u_i| < (1 + \epsilon_0/2)\alpha\} = \emptyset.$$

Thus, with $\beta = \epsilon_0/2$, this particular $\alpha$ also satisfies Equation (16) and we conclude that with probability at least $1 - O(n^{-8} \log n)$, there exists at least one $\alpha \in \mathcal{A}_0$ such that both Equations (15) and (16) hold.

Our argument above ensures that with probability at least $1 - O(n^{-8} \log n)$, there exists $\alpha \in \mathcal{A}_0$ satisfying Equations (15) and (16). Thus, $\widehat{\alpha} \in \mathcal{A}_0$ as constructed in Step 2 exists, and our argument above ensures that $\hat{I}$ as constructed in Step 2 satisfies $\hat{I} = \hat{I}_{\widehat{\alpha}} = I_{\widehat{\alpha}}$, completing the proof. $\qquad\square$

**Lemma 8.** *Under Assumptions 1 and 3, with probability at least $1 - O(n^{-8})$ it holds for all $i \in [n]$ that*

$$|u_i^\star| \geq \frac{1}{24\sqrt{n}} \quad \text{implies} \quad \operatorname{sgn}(u_i) = \operatorname{sgn}(u_i^\star).$$

*Proof of Lemma 8.* Let $i \in [n]$ and suppose that $|u_i^\star| \geq 1/24\sqrt{n}$ with $u_i^\star \geq 0$. The case for $u_i^\star < 0$ follows by an analogous argument and details are omitted. By Equation (38) in Lemma 5, it holds with probability at least $1 - O(n^{-8})$ that for all $i \in [n]$ such that $u_i^\star \geq 0$,

$$u_i \geq u_i^\star - |u_i - u_i^\star| \geq \left(1 - \frac{120\sqrt{\nu_W n}}{|\lambda^\star|}\right) u_i^\star - \frac{80\sqrt{\nu_W \log n}}{|\lambda^\star|}.$$

By Assumption 3, it follows that when $|\lambda^\star| \geq 2400\sqrt{\nu_W n}$

$$u_i \geq \frac{19}{20} u_i^\star - \frac{80\sqrt{\nu_W \log n}}{|\lambda^\star|}.$$

Since $u_i^\star \geq 1/24\sqrt{n}$ by assumption, it follows that when $|\lambda^\star| \geq 2400\sqrt{\nu_W n}$, we have

$$u_i^\star \geq \frac{100\sqrt{\nu_W \log n}}{|\lambda^\star|},$$

from which we have $u_i > 0$ and therefore $\operatorname{sgn}(u_i) = \operatorname{sgn}(u_i^\star)$. $\qquad\square$

**Lemma 9.** *Fix $\alpha \in \mathcal{A}$ and $s \in \{\pm 1\}^n$. For a random noise matrix $W \in \mathbb{R}^{n \times n}$ satisfying Assumption 1, it holds with probability at least $1 - O(n^{-8})$ that*

$$\left| \sum_{j,k \in I_\alpha} s_j s_k W_{jk} \right| \leq C |I_\alpha| \sqrt{\nu_W \log n} \tag{54}$$

*and*

$$\max_{j \in [n]} \left| \sum_{k \in I_\alpha} s_j s_k W_{jk} \right| \leq C \sqrt{|I_\alpha| \nu_W \log n}. \tag{55}$$

*Proof.* Since $s \in \{\pm 1\}^n$ is fixed and the entries of $W$ are symmetric about zero, we assume without loss of generality that $s$ is a vector of all 1's. By standard concentration inequalities [54], we have that for any $t > 0$,

$$\mathbb{P}\left(\left|\sum_{j,k \in I_\alpha} W_{jk}\right| \geq t\right) \leq 2 \exp\left(\frac{-t^2}{2|I_\alpha|^2 \nu_W}\right).$$

Setting $t = C|I_\alpha|\sqrt{\nu_W \log n}$ for suitably-chosen $C > 0$, Equation (54) holds with probability at least $1 - O(n^{-8})$. Similarly, by standard subgaussian concentration inequalities [54] and a union bound over $j \in [n]$, it holds for all $t > 0$ that

$$\mathbb{P}\left(\max_{j \in [n]}\left|\sum_{k \in I_\alpha} W_{jk}\right| > t\right) \leq 2n \exp\left(\frac{-t^2}{2|I_\alpha|\nu_W}\right).$$

Taking $t^2 = C|I_\alpha|\nu_W \log n$ for $C > 0$ chosen suitably large, Equation (55) holds with probability at least $1 - O(n^{-8})$, completing the proof. $\square$

### D.2 Proof of Theorem 1

*Proof.* In this proof, we control the estimation error of $\widehat{u}$ under $d_\infty$. There are two types of entries of $\widehat{u}$, as stated in Step 5 of Algorithm 1. One type is derived from $\widehat{v}$ constructed in Step 4 of Algorithm 1, and corresponds to large entries of $u^\star$. The other type is given by the top eigenvector $u$ of $Y$, corresponding to small entries of $u^\star$. We handle these two types of entries separately. In **Part I** of the proof, we control the estimation error corresponding to the first type of entries, derived from $\widehat{v}$. In **Part II** of the proof, we control the estimation error corresponding to the second type of entries, derived from $u$. Combining these two parts, we obtain a high probability bound for the estimation error $d_\infty(\widehat{u}, u^\star)$.

**Part I. Estimation error related to $\widehat{v}$.**

We define the event $\mathcal{E}_1$ according to

$$\mathcal{E}_1 = \{|\widehat{\lambda} - \lambda^\star| \leq C_2\sqrt{\nu_W}\log^{5/2} n\}. \tag{56}$$

On event $\mathcal{E}_1$, if $\lambda^\star > 0$, then by the triangle inequality, we have

$$\widehat{\lambda} \geq \lambda^\star - \left|\lambda^\star - \widehat{\lambda}\right| \geq \lambda^\star - C_2\sqrt{\nu_W}\log^{5/2} n > 0, \tag{57}$$

where the second inequality holds for all sufficiently large $n$ by Assumption 3. Similarly, for all sufficiently large $n$ we have $\widehat{\lambda} < 0$ if $\lambda^\star < 0$. In other words, for all sufficiently large $n$, we have

$$\mathrm{sgn}\left(\widehat{\lambda}\right) = \mathrm{sgn}\left(\lambda^\star\right)$$

on event $\mathcal{E}_1$. Hence, Step 1 in Algorithm 1 guarantees that we work with the original $Y$ when $\lambda^\star > 0$, or with $-Y$ when $\lambda^\star < 0$. In either case, the algorithm proceeds with a signal matrix that has a positive leading eigenvalue on $\mathcal{E}_1$. Without loss of generality, we assume that $\lambda^\star > 0$.

On event $\mathcal{E}_1$, we have

$$\left|\frac{\lambda^\star}{\widehat{\lambda}} - 1\right| \leq \frac{C_2\sqrt{\nu_W}(\log n)^{5/2}}{\lambda^\star - |\lambda^\star - \widehat{\lambda}|} \leq \frac{C\sqrt{\nu_W}(\log n)^{5/2}}{\lambda^\star}. \tag{58}$$

where the second inequality holds for all sufficiently large $n$ by Assumption 3. Similarly, for $n$ sufficiently large, following Equation (58), event $\mathcal{E}_1$ also implies

$$1/2 \leq |\lambda^\star/\widehat{\lambda}| \leq 2. \tag{59}$$

Recalling the set $\mathcal{A}_0$ from Equation (53) and $I_\alpha$ from Equation (52) above, for each $\alpha \in \mathcal{A}_0$, define the events $\mathcal{E}_{2,\alpha}$ and $\mathcal{E}_{3,\alpha}$ according to

$$\mathcal{E}_{2,\alpha} := \{\widehat{I} = I_\alpha\}, \tag{60}$$

where $\hat{I}$ is as constructed in Step 2 of Algorithm 1, and

$$\mathcal{E}_{3,\alpha} := \{Q_{kk} = \operatorname{sgn}(u_k^\star) \text{ for all } k \in I_\alpha, \text{ and } Q_{kk} = 1 \text{ for all } k \in I_\alpha^c\}. \tag{61}$$

Define $s \in \{\pm 1\}^n$ according to

$$s_k := \begin{cases} \operatorname{sgn}(u_k^\star) & \text{if } k \in I_\alpha, \\ 1 & \text{otherwise,} \end{cases} \tag{62}$$

and for each $\alpha \in \mathcal{A}_0$, define the event $\mathcal{E}_{4,\alpha}$ as

$$\mathcal{E}_{4,\alpha} = \left\{ \left| \sum_{j,k \in I_\alpha} s_j s_k W_{jk} \right| \le C|I_\alpha|\sqrt{\nu_W \log n} \right\}. \tag{63}$$

For $\widehat{S}$ given in Step 4 of Algorithm 1, plugging in $\widetilde{Y} = QYQ$ and expanding, it follows that on the event $\mathcal{E}_{2,\alpha} \cap \mathcal{E}_{3,\alpha}$,

$$\widehat{S}^2 = \sum_{j,k \in \hat{I}} Q_{jj} Q_{kk} Y_{jk} = \lambda^\star \left( \sum_{k \in I_\alpha} |u_k^\star| \right)^2 + \sum_{j,k \in I_\alpha} s_j s_k W_{jk}. \tag{64}$$

On the event $\mathcal{E}_{2,\alpha} \cap \mathcal{E}_{3,\alpha} \cap \mathcal{E}_{4,\alpha}$, we can lower-bound this right-hand side, obtaining

$$\widehat{S}^2 \ge \lambda^\star \left( \sum_{k \in I_\alpha} |u_k^\star| \right)^2 - C|I_\alpha|\sqrt{\nu_W \log n} \ge \lambda^\star |I_\alpha|^2 \alpha^2 - C|I_\alpha|\sqrt{\nu_W \log n}, \tag{65}$$

where the first inequality follows from the fact that event $\mathcal{E}_{4,\alpha}$ holds and the second inequality follows from the definition of $I_\alpha$ in Equation (52). Using Equation (53) and the fact that $\alpha \in \mathcal{A}_0$, we may further bound this by

$$\widehat{S}^2 \ge \frac{|I_\alpha|}{\log^2 n} \left( \lambda^\star - C\sqrt{\nu_W} \log^{5/2} n \right) > 0,$$

where the second inequality holds under Assumption 3 when $n$ is sufficiently large. Thus, on the event $\mathcal{E}_{2,\alpha} \cap \mathcal{E}_{3,\alpha} \cap \mathcal{E}_{4,\alpha}$, we have $\sum_{j,k \in \hat{I}} \widetilde{Y}_{jk} > 0$ and thus $\widehat{S}$ is well-defined (i.e., it is the square root of a positive number).

We now show that $\widehat{S}$ is close to its population target. Rearranging the terms in Equation (64), we have that on the event $\mathcal{E}_{2,\alpha} \cap \mathcal{E}_{3,\alpha} \cap \mathcal{E}_{4,\alpha}$,

$$\left| \widehat{S}^2 - \lambda^\star \left( \sum_{k \in I_\alpha} |u_k^\star| \right)^2 \right| \le C|I_\alpha|\sqrt{\nu_W \log n}. \tag{66}$$

Dividing both sides by $\widehat{S} + \sqrt{\lambda^\star} \sum_{k \in I_\alpha} |u_k^\star|$, when the event $\mathcal{E}_{2,\alpha} \cap \mathcal{E}_{3,\alpha} \cap \mathcal{E}_{4,\alpha}$ holds, we have

$$\left| \widehat{S} - \sqrt{\lambda^\star} \sum_{k \in I_\alpha} |u_k^\star| \right| \le \frac{C|I_\alpha|\sqrt{\nu_W \log n}}{\left( \widehat{S} + \sqrt{\lambda^\star} \sum_{k \in I_\alpha} |u_k^\star| \right)} \le \frac{C|I_\alpha|\sqrt{\nu_W \log n}}{\sqrt{\lambda^\star} \sum_{k \in I_\alpha} |u_k^\star|} \le \frac{C\sqrt{\nu_W \log n}}{\sqrt{\lambda^\star}\alpha}, \tag{67}$$

where the second inequality follows from the fact that $\widehat{S} > 0$ and the last inequality follows from the definition of $I_\alpha$ in Equation (52). Dividing by $\sqrt{\lambda^\star} \sum_{k \in I_\alpha} |u_k^\star|$ on both sides (note that this quantity is positive by definition of $I_\alpha$ and the fact that $\alpha \in \mathcal{A}_0$), it follows that on the event $\mathcal{E}_{2,\alpha} \cap \mathcal{E}_{3,\alpha} \cap \mathcal{E}_{4,\alpha}$,

$$\left| \frac{\widehat{S}}{\sqrt{\lambda^\star} \sum_{k \in I_\alpha} |u_k^\star|} - 1 \right| \le \frac{C\sqrt{\nu_W \log n}}{\lambda^\star \alpha \sum_{k \in I_\alpha} |u_k^\star|} \le \frac{C\sqrt{\nu_W \log n}}{\lambda^\star |I_\alpha| \alpha^2} \le \frac{C\sqrt{\nu_W}(\log n)^{5/2}}{\lambda^\star}, \tag{68}$$

where the second inequality follows from the definition of $I_\alpha$ in Equation (52) and the last inequality follows from Equation (53).

We move on to Step 4 of Algorithm 1, from which we recall, for ease of reference, that

$$\widehat{v}_j = \frac{\sum_{k\in\widehat{I}} \widetilde{Y}_{jk}}{\sqrt{\widehat{\lambda}}\widehat{S}} \quad \text{for } j \in [n]. \tag{69}$$

Note that by Equation (57), the square root of $\widehat{\lambda}$ is well-defined on $\mathcal{E}_1$. Rearranging the definition of $\widehat{v}_j$ in Equation (69), plugging in $\widetilde{Y} = QYQ$ and expanding, we have for $j \in [n]$,

$$\frac{\widehat{v}_j\widehat{S}}{\sqrt{\lambda^\star}} = \sqrt{\frac{\lambda^\star}{\widehat{\lambda}}} Q_{jj} u_j^\star \sum_{k\in\widehat{I}} Q_{kk} u_k^\star + \frac{\sum_{k\in\widehat{I}} Q_{jj}Q_{kk}W_{jk}}{\sqrt{\widehat{\lambda}\lambda^\star}}.$$

Rearranging terms and adding $\widehat{v}_j \sum_{k\in\widehat{I}} Q_{kk} u_k^\star$ to both sides, we have for any $j \in [n]$

$$\left(\widehat{v}_j - Q_{jj}u_j^\star\right) \sum_{k\in\widehat{I}} Q_{kk} u_k^\star = \widehat{v}_j \left(\sum_{k\in\widehat{I}} Q_{kk} u_k^\star - \frac{\widehat{S}}{\sqrt{\lambda^\star}}\right)$$

$$+ \left(\sqrt{\frac{\lambda^\star}{\widehat{\lambda}}} - 1\right) Q_{jj} u_j^\star \sum_{k\in\widehat{I}} Q_{kk} u_k^\star + \frac{\sum_{k\in\widehat{I}} Q_{jj}Q_{kk}W_{jk}}{\sqrt{\widehat{\lambda}\lambda^\star}}.$$

Dividing by $\sum_{k\in\widehat{I}} Q_{kk} u_k^\star$ on both sides, again noting that this quantity is positive on the event $\mathcal{E}_{2,\alpha} \cap \mathcal{E}_{3,\alpha}$,

$$\widehat{v}_j - Q_{jj}u_j^\star = \widehat{v}_j \left(1 - \frac{\widehat{S}}{\sqrt{\lambda^\star}\sum_{k\in\widehat{I}} Q_{kk} u_k^\star}\right) + \left(\sqrt{\frac{\lambda^\star}{\widehat{\lambda}}} - 1\right) Q_{jj}u_j^\star + \frac{\sum_{k\in\widehat{I}} Q_{jj}Q_{kk}W_{jk}}{\sqrt{\widehat{\lambda}\lambda^\star}\sum_{k\in\widehat{I}} Q_{kk} u_k^\star}. \tag{70}$$

Taking absolute values in Equation (70) and applying the triangle inequality,

$$\left|\widehat{v}_j - Q_{jj}u_j^\star\right| \le |\widehat{v}_j| \left|1 - \frac{\widehat{S}}{\sqrt{\lambda^\star}\sum_{k\in\widehat{I}} Q_{kk} u_k^\star}\right| + \left|\sqrt{\frac{\lambda^\star}{\widehat{\lambda}}} - 1\right| |Q_{jj}u_j^\star| + \left|\frac{\sum_{k\in\widehat{I}} Q_{jj}Q_{kk}W_{jk}}{\sqrt{\widehat{\lambda}\lambda^\star}\sum_{k\in\widehat{I}} Q_{kk} u_k^\star}\right|.$$

Trivially, $|Q_{jj}u_j^\star| \le 1$. Further, on the event $\mathcal{E}_{2,\alpha} \cap \mathcal{E}_{3,\alpha}$, we have $\widehat{I} = I_\alpha$ and $Q_{jj} = \operatorname{sgn}\left(u_j^\star\right) = s_j$ for all $j \in I_\alpha$. Thus, we may write

$$\left|\widehat{v}_j - Q_{jj}u_j^\star\right| \le |\widehat{v}_j| \left|1 - \frac{\widehat{S}}{\sqrt{\lambda^\star}\sum_{k\in I_\alpha} |u_k^\star|}\right| + \left|\sqrt{\frac{\lambda^\star}{\widehat{\lambda}}} - 1\right| + \left|\frac{\sum_{k\in I_\alpha} s_j s_k W_{jk}}{\lambda^\star \sum_{k\in I_\alpha} |u_k^\star|}\right| \sqrt{\frac{\lambda^\star}{\widehat{\lambda}}}. \tag{71}$$

On the event $\mathcal{E}_1$, applying Equations (58) and (59) yields that

$$\left|\sqrt{\frac{\lambda^\star}{\widehat{\lambda}}} - 1\right| = \frac{\left|\frac{\lambda^\star}{\widehat{\lambda}} - 1\right|}{\sqrt{\frac{\lambda^\star}{\widehat{\lambda}}} + 1} \le \frac{C\sqrt{\nu_W}\log^{5/2} n}{\lambda^\star}. \tag{72}$$

Applying this bound in Equation (71), on the event $\mathcal{E}_1 \cap \mathcal{E}_{2,\alpha} \cap \mathcal{E}_{3,\alpha}$, we have

$$\left|\widehat{v}_j - Q_{jj}u_j^\star\right| \le |\widehat{v}_j| \left|1 - \frac{\widehat{S}}{\sqrt{\lambda^\star}\sum_{k\in I_\alpha} |u_k^\star|}\right| + \frac{C\sqrt{\nu_W}\log^{5/2} n}{\lambda^\star} + \left|\frac{\sum_{k\in I_\alpha} s_j s_k W_{jk}}{\lambda^\star \sum_{k\in I_\alpha} |u_k^\star|}\right| \sqrt{\frac{\lambda^\star}{\widehat{\lambda}}}.$$

We remind the reader that in the proof, constant $C$ may change its precise value from line to line. Another application of Equation (72) and using Assumption 3 yields

$$\left|\widehat{v}_j - Q_{jj}u_j^\star\right| \le |\widehat{v}_j| \left|1 - \frac{\widehat{S}}{\sqrt{\lambda^\star}\sum_{k\in I_\alpha} |u_k^\star|}\right| + \frac{C\sqrt{\nu_W}\log^{5/2} n}{\lambda^\star} + C\left|\frac{\sum_{k\in I_\alpha} s_j s_k W_{jk}}{\lambda^\star \sum_{k\in I_\alpha} |u_k^\star|}\right|.$$

On the event $\mathcal{E}_1 \cap \mathcal{E}_{2,\alpha} \cap \mathcal{E}_{3,\alpha} \cap \mathcal{E}_{4,\alpha}$, Equation (68) holds and we have

$$\left|\widehat{v}_j - Q_{jj}u_j^\star\right| \le |\widehat{v}_j|\frac{C\sqrt{\nu_W}\log^{5/2} n}{\lambda^\star} + \frac{C\sqrt{\nu_W}\log^{5/2} n}{\lambda^\star} + C\left|\frac{\sum_{k\in I_\alpha} s_j s_k W_{jk}}{\lambda^\star \sum_{k\in I_\alpha} |u_k^\star|}\right|. \tag{73}$$

Define the event

$$\mathcal{E}_{5,\alpha} = \left\{ \max_{j \in [n]} \left| \sum_{k \in I_\alpha} s_j s_k W_{jk} \right| \le C\sqrt{|I_\alpha|\nu_W \log n} \right\}. \tag{74}$$

Under the event $\mathcal{E}_1 \cap \mathcal{E}_{2,\alpha} \cap \mathcal{E}_{3,\alpha} \cap \mathcal{E}_{4,\alpha} \cap \mathcal{E}_{5,\alpha}$, we can further bound Equation (73) by

$$\left| \widehat{v}_j - Q_{jj}u_j^\star \right| \le |\widehat{v}_j| \frac{C\sqrt{\nu_W}\log^{5/2} n}{\lambda^\star} + \frac{C\sqrt{\nu_W}\log^{5/2} n}{\lambda^\star} + \frac{C\sqrt{|I_\alpha|\nu_W \log n}}{\lambda^\star|I_\alpha|\alpha}.$$

Applying the definition of $\mathcal{A}_0$ in Equation (53), on the event $\mathcal{E}_1 \cap \mathcal{E}_{2,\alpha} \cap \mathcal{E}_{3,\alpha} \cap \mathcal{E}_{4,\alpha} \cap \mathcal{E}_{5,\alpha}$, it holds for all $j \in [n]$ that

$$\left| \widehat{v}_j - Q_{jj}u_j^\star \right| \le |\widehat{v}_j| \frac{C\sqrt{\nu_W}\log^{5/2} n}{\lambda^\star} + \frac{C\sqrt{\nu_W}\log^{5/2} n}{\lambda^\star}. \tag{75}$$

Applying the triangle inequality and using the fact that $|Q_{jj}u_j^\star| \le 1$ by definition,

$$(|\widehat{v}_j| + 1)\frac{C\sqrt{\nu_W}\log^{5/2} n}{\lambda^\star} \le |\widehat{v}_j - Q_{jj}u_j^\star|\frac{C\sqrt{\nu_W}\log^{5/2} n}{\lambda^\star} + \left(|Q_{jj}u_j^\star| + 1\right)\frac{C\sqrt{\nu_W}\log^{5/2} n}{\lambda^\star}$$

$$\le |\widehat{v}_j - Q_{jj}u_j^\star|\frac{C\sqrt{\nu_W}\log^{5/2} n}{\lambda^\star} + \frac{C\sqrt{\nu_W}\log^{5/2} n}{\lambda^\star}.$$

Applying this bound to Equation (75),

$$\left| \widehat{v}_j - Q_{jj}u_j^\star \right| \le |\widehat{v}_j - Q_{jj}u_j^\star|\frac{C\sqrt{\nu_W}(\log n)^{5/2}}{\lambda^\star} + \frac{C\sqrt{\nu_W}(\log n)^{5/2}}{\lambda^\star}.$$

Rearranging the terms, under Assumption 3 for $n$ sufficiently large and when the event $\mathcal{E}_1 \cap \mathcal{E}_{2,\alpha} \cap \mathcal{E}_{3,\alpha} \cap \mathcal{E}_{4,\alpha} \cap \mathcal{E}_{5,\alpha}$ holds, we have for all $j \in [n]$,

$$\left| \widehat{v}_j - Q_{jj}u_j^\star \right| \le \frac{C\sqrt{\nu_W}(\log n)^{5/2}}{\lambda^\star}. \tag{76}$$

Recall from Step 5 of Algorithm 1 that we set $\widehat{u}_j = Q_{jj}\widehat{v}_j$ for all $j \in [n]$ such that $|u_j| > (\sigma/\widehat{\lambda})\log n$. For any such $j$, on the event $\mathcal{E}_1 \cap \mathcal{E}_{2,\alpha} \cap \mathcal{E}_{3,\alpha} \cap \mathcal{E}_{4,\alpha} \cap \mathcal{E}_{5,\alpha}$, Equation (76) holds, and we have

$$|\widehat{u}_j - u_j^\star| = |Q_{jj}\widehat{v}_j - Q_{jj}^2 u_j^\star| = |Q_{jj}||\widehat{v}_j - Q_{jj}u_j^\star| \le \frac{C\sqrt{\nu_W}\log^{5/2} n}{\lambda^\star}. \tag{77}$$

Consider the event

$$\mathcal{E} = \left\{ \|Q\widehat{v} - u^\star\|_\infty \le \frac{C\sqrt{\nu_W}\log^{5/2} n}{\lambda^\star} \right\}.$$

Recalling the events $\mathcal{E}_1, \mathcal{E}_{2,\alpha}, \mathcal{E}_{3,\alpha}, \mathcal{E}_{4,\alpha}$ and $\mathcal{E}_{5,\alpha}$ defined in Equations (56) (60) (61) (63) and (74), respectively, we have

$$\mathbb{P}(\mathcal{E}) \ge \mathbb{P}\left( \mathcal{E} \cap \mathcal{E}_1 \cap \left( \bigcup_{\alpha \in \mathcal{A}_0} \mathcal{E}_{2,\alpha} \cap \mathcal{E}_{3,\alpha} \cap \mathcal{E}_{4,\alpha} \cap \mathcal{E}_{5,\alpha} \right) \right)$$

$$= \mathbb{P}\left( \bigcup_{\alpha \in \mathcal{A}_0} \mathcal{E}_1 \cap \mathcal{E}_{2,\alpha} \cap \mathcal{E}_{3,\alpha} \cap \mathcal{E}_{4,\alpha} \cap \mathcal{E}_{5,\alpha} \right), \tag{78}$$

where the last equality holds because for any $\alpha \in \mathcal{A}_0$, the event $\mathcal{E}$ is implied by the event $\mathcal{E}_1 \cap \mathcal{E}_{2,\alpha} \cap \mathcal{E}_{3,\alpha} \cap \mathcal{E}_{4,\alpha} \cap \mathcal{E}_{5,\alpha}$, which is a fact established in the proof of Equations (76) and (77).

Since by the definition in Equation (60), $\{\mathcal{E}_{2,\alpha}\}_{\alpha \in \mathcal{A}_0}$ are disjoint, it follows from Equation (78) that

$$\mathbb{P}(\mathcal{E}) \ge \sum_{\alpha \in \mathcal{A}_0} \mathbb{P}(\mathcal{E}_1 \cap \mathcal{E}_{2,\alpha} \cap \mathcal{E}_{3,\alpha} \cap \mathcal{E}_{4,\alpha} \cap \mathcal{E}_{5,\alpha})$$

$$= \sum_{\alpha \in \mathcal{A}_0} \left[ \mathbb{P}(\mathcal{E}_1 \cap \mathcal{E}_{2,\alpha}) - \mathbb{P}(\mathcal{E}_1 \cap \mathcal{E}_{2,\alpha} \cap (\mathcal{E}_{3,\alpha} \cap \mathcal{E}_{4,\alpha} \cap \mathcal{E}_{5,\alpha})^c) \right]. \tag{79}$$

By the union bound and basic set inclusions,

$$
\begin{aligned}
\mathbb{P}\left(\mathcal{E}_1 \cap \mathcal{E}_{2,\alpha} \cap (\mathcal{E}_{3,\alpha} \cap \mathcal{E}_{4,\alpha} \cap \mathcal{E}_{5,\alpha})^c\right) &= \mathbb{P}\left[\mathcal{E}_1 \cap \mathcal{E}_{2,\alpha} \cap \left(\mathcal{E}_{3,\alpha}^c \cup \mathcal{E}_{4,\alpha}^c \cup \mathcal{E}_{5,\alpha})^c\right)\right] \\
&\leq \mathbb{P}\left(\mathcal{E}_1 \cap \mathcal{E}_{2,\alpha} \cap \mathcal{E}_{3,\alpha}^c\right) + \mathbb{P}\left(\mathcal{E}_1 \cap \mathcal{E}_{2,\alpha} \cap \mathcal{E}_{4,\alpha}^c\right) \\
&\quad + \mathbb{P}\left(\mathcal{E}_1 \cap \mathcal{E}_{2,\alpha} \cap \mathcal{E}_{5,\alpha}^c\right) \\
&\leq \mathbb{P}\left(\mathcal{E}_{2,\alpha} \cap \mathcal{E}_{3,\alpha}^c\right) + \mathbb{P}\left(\mathcal{E}_{4,\alpha}^c\right) + \mathbb{P}\left(\mathcal{E}_{5,\alpha}^c\right).
\end{aligned}
$$

Applying this bound in Equation (79),

$$
\mathbb{P}\left(\mathcal{E}\right) \geq \sum_{\alpha \in \mathcal{A}_0} \left[\mathbb{P}\left(\mathcal{E}_1 \cap \mathcal{E}_{2,\alpha}\right) - \left(\mathbb{P}(\mathcal{E}_{2,\alpha} \cap \mathcal{E}_{3,\alpha}^c) + \mathbb{P}(\mathcal{E}_{4,\alpha}^c) + \mathbb{P}(\mathcal{E}_{5,\alpha}^c)\right)\right] \tag{80}
$$

The terms $\mathbb{P}(\mathcal{E}_{4,\alpha}^c)$ and $\mathbb{P}(\mathcal{E}_{5,\alpha}^c)$ are bounded in Lemma 9 as

$$
\mathbb{P}(\mathcal{E}_{4,\alpha}^c) + \mathbb{P}(\mathcal{E}_{5,\alpha}^c) \leq O(n^{-8}). \tag{81}
$$

For the term $\mathbb{P}(\mathcal{E}_{2,\alpha} \cap \mathcal{E}_{3,\alpha}^c)$, we have

$$
\mathbb{P}(\mathcal{E}_{2,\alpha} \cap \mathcal{E}_{3,\alpha}^c) = \mathbb{P}\left(\bigcup_{k \in I_\alpha} \left\{Q_{kk} \neq \mathrm{sgn}\left(u_k^\star\right), \hat{I} = I_\alpha\right\} \cup \bigcup_{k \in I_\alpha^c} \left\{Q_{kk} \neq 1, \hat{I} = I_\alpha\right\}\right).
$$

By Step 3 in Algorithm 1, we must have

$$
\bigcup_{k \in I_\alpha^c} \left\{Q_{kk} \neq 1, \hat{I} = I_\alpha\right\} = \bigcup_{k \in \hat{I}^c} \left\{Q_{kk} \neq 1, \hat{I} = I_\alpha\right\} = \emptyset.
$$

It follows that

$$
\begin{aligned}
\mathbb{P}(\mathcal{E}_{2,\alpha} \cap \mathcal{E}_{3,\alpha}^c) &= \mathbb{P}\left(\bigcup_{k \in I_\alpha} \left\{\mathrm{sgn}\left(u_k\right) \neq \mathrm{sgn}\left(u_k^\star\right), \hat{I} = I_\alpha\right\}\right) \\
&\leq \mathbb{P}\left(\bigcup_{k \in I_\alpha} \left\{\mathrm{sgn}\left(u_k\right) \neq \mathrm{sgn}\left(u_k^\star\right)\right\}\right).
\end{aligned} \tag{82}
$$

For any $k \in I_\alpha$, by the construction of $\mathcal{A}$ in Equation (10), we must have $|u_k^\star| \geq \alpha \geq 1/\sqrt{2n}$. Thus, by Lemma 8, we can further bound Equation (82) as

$$
\mathbb{P}(\mathcal{E}_{2,\alpha} \cap \mathcal{E}_{3,\alpha}^c) \leq O(n^{-8}). \tag{83}
$$

Applying Equations (81) and (83) to Equation (80), we have

$$
\begin{aligned}
\mathbb{P}(\mathcal{E}) &\geq \sum_{\alpha \in \mathcal{A}_0} \mathbb{P}\left(\mathcal{E}_1 \cap \mathcal{E}_{2,\alpha}\right) - O(|\mathcal{A}_0|n^{-8}) \\
&= 1 - \mathbb{P}\left(\mathcal{E}_1^c \cup \left(\bigcap_{\alpha \in \mathcal{A}_0} \mathcal{E}_{2,\alpha}^c\right)\right) - O(|\mathcal{A}_0|n^{-8}) \\
&\geq 1 - \mathbb{P}(\mathcal{E}_1^c) - \mathbb{P}\left(\bigcap_{\alpha \in \mathcal{A}_0} \left\{\hat{I} \neq I_\alpha\right\}\right) - O(|\mathcal{A}_0|n^{-8}).
\end{aligned}
$$

Under Assumption 4, we have $\mathbb{P}(\mathcal{E}_1^c) \leq O(n^{-8} \log n)$, and it follows that

$$
\mathbb{P}(\mathcal{E}) \geq 1 - O(n^{-8} \log n) - \mathbb{P}\left(\bigcap_{\alpha \in \mathcal{A}_0} \left\{\hat{I} \neq I_\alpha\right\}\right) - O(|\mathcal{A}_0|n^{-8}).
$$

Applying Lemma 7 to the right hand side yields that

$$
\mathbb{P}(\mathcal{E}) \geq 1 - O(n^{-8} \log n) - O(n^{-8}) - O(|\mathcal{A}_0|n^{-8}) \geq 1 - O(n^{-8} \log n),
$$

where the last inequality follows from the fact that $|\mathcal{A}_0| \leq |\mathcal{A}|$ and $|\mathcal{A}| = O(\log n)$ by Equation (10).

At this stage, we have shown that with probability at least $1 - O(n^{-8} \log n)$,

$$\|\boldsymbol{Q}\widehat{\boldsymbol{v}} - \boldsymbol{u}^\star\|_\infty \leq \frac{C\sqrt{\nu_W} \log^{5/2} n}{\lambda^\star}. \tag{84}$$

**Part II. Estimation error related to $u$.**

Note that Step 5 in Algorithm 1 further refines $\widehat{\boldsymbol{v}}$ to adjust the estimates of small entries of $\boldsymbol{u}^\star$. In the remainder of the proof, we show that this refinement does not affect our bound in Equation (84). By Lemma 5, it holds with probability at least $1 - O(n^{-8})$ that for all $j \in [n]$,

$$|u_j - u_j^\star| \leq \frac{80\sqrt{\nu_W \log n} + 120\sqrt{\nu_W n}\,|u_j^\star|}{\lambda^\star}$$

$$\leq \frac{80\sqrt{\nu_W \log n}}{\lambda^\star} + \frac{120\sqrt{\nu_W n}}{\lambda^\star}\left(|u_j| + |u_j - u_j^\star|\right),$$

where the second line follows from the triangle inequality. Rearranging terms and noting that $\lambda^\star > 240\sqrt{\nu_W n \log n}$ under Assumption 3, we have that with probability at least $1 - O(n^{-8})$, it holds for all $j \in [n]$ that

$$|u_j - u_j^\star| \leq \frac{1}{1 - 120\sqrt{\nu_W n}/\lambda^\star}\left(\frac{80\sqrt{\nu_W \log n}}{\lambda^\star} + \frac{120\sqrt{\nu_W n}}{\lambda^\star}\,|u_j|\right)$$

$$\leq \frac{160\sqrt{\nu_W \log n}}{\lambda^\star} + \frac{|u_j|}{\sqrt{\log n}}.$$

That is, defining the event

$$\mathcal{E}_6 = \bigcap_{j=1}^{n}\left\{|u_j - u_j^\star| \leq \frac{160\sqrt{\nu_W \log n}}{\lambda^\star} + \frac{|u_j|}{\sqrt{\log n}}\right\}, \tag{85}$$

we have

$$\mathbb{P}(\mathcal{E}_6^c) \leq O(n^{-8}). \tag{86}$$

Note that by Equation (58) and Assumption 3, when $\mathcal{E}_1$ holds, we have $|\lambda^\star/\widehat{\lambda}| \leq 2$ for sufficiently large $n$. It follows that when $\mathcal{E}_1$ holds, for all $j \in [n]$ such that $|u_j| \leq (\sqrt{\nu_W}/\widehat{\lambda}) \log n$, we have

$$|u_j| = \frac{\sqrt{\nu_W} \log n}{\lambda^\star} \cdot \frac{\lambda^\star}{\widehat{\lambda}} \leq \frac{2\sqrt{\nu_W} \log n}{\lambda^\star}, \tag{87}$$

where the inequality follows from Equation (58). Define the set $\hat{J} = \{j : |u_j| \leq (\sqrt{\nu_W}/\widehat{\lambda}) \log n\}$. We have

$$\mathbb{P}\left(\bigcup_{j \in \hat{J}}\left\{|u_j - u_j^\star| > \frac{162\sqrt{\nu_W \log n}}{\lambda^\star}\right\}\right)$$

$$\leq \mathbb{P}\left(\bigcup_{j \in \hat{J}}\left\{|u_j - u_j^\star| > \frac{162\sqrt{\nu_W \log n}}{\lambda^\star}\right\} \cap \mathcal{E}_1\right) + \mathbb{P}(\mathcal{E}_1^c)$$

$$\leq \mathbb{P}\left(\bigcup_{j \in \hat{J}}\left\{|u_j - u_j^\star| > \frac{160\sqrt{\nu_W \log n}}{\lambda^\star} + \frac{|u_j|}{\sqrt{\log n}}\right\} \cap \mathcal{E}_1\right) + O(n^{-8}),$$

where the second inequality follows from Equation (87) and Assumption 4. Recalling the definition of $\mathcal{E}_6$ in Equation (85) and using the fact that $\hat{J} \subseteq [n]$, we have that the set in the last inequality of the above display is a subset of $\mathcal{E}_6^c$. Following the above bound and basic set inclusions, we have

$$\mathbb{P}\left(\bigcup_{j \in \hat{J}}\left\{|u_j - u_j^\star| > \frac{162\sqrt{\nu_W \log n}}{\lambda^\star}\right\}\right) \leq \mathbb{P}(\mathcal{E}_6^c) + O(n^{-8}) \leq O(n^{-8}).$$

Combining this with Equation (84) and recalling the definition of $\widehat{\boldsymbol{u}}$ in Step 5, we have that with probability at least $1 - O(n^{-8})$,

$$d_\infty(\widehat{\boldsymbol{u}}, \boldsymbol{u}^\star) \leq \frac{C\sqrt{\nu_W} \log^{5/2} n}{\lambda^\star},$$

as we set out to show. $\qquad\square$

# E   A new estimator for $\lambda^\star$

Consider $\widetilde{Y}$ and $\widehat{S}$ as constructed in Steps 3 and 4, respectively, of Algorithm 1. We can estimate $\lambda^\star$ by

$$\widehat{\lambda} = \frac{\sum_{j=1}^n \left(\sum_{k \in \hat{I}} \widetilde{Y}_{jk}\right)^2 - |\hat{I}| n \sigma^2}{\widehat{S}^2}. \tag{88}$$

Proposition 1 controls the estimation error of $\widehat{\lambda}$ given in Equation (88).

**Proposition 1.** *Under the model in Equation* (1)*, suppose that Assumptions 1, 2 and 3 hold and consider the eigenvalue estimate $\widehat{\lambda}$ as defined in Equation* (88)*. For $n$ sufficiently large, we have*

$$\left|\widehat{\lambda} - \lambda^\star\right| \lesssim \sqrt{\nu_W} \log^{5/2} n$$

*with probability at least $1 - O(n^{-8} \log n)$.*

## E.1   Technical lemmas for Proposition 1

Our proof of Proposition 1, which appears in Section E.2 below, makes use of two technical lemmas, which we establish first.

**Lemma 10.** *Let $W \in \mathbb{R}^{n \times n}$ be a random matrix satisfying Assumption 1. For a fixed $s \in \{\pm 1\}^n$, for any $\alpha \in \mathcal{A}_0$, with probability at least $1 - O(n^{-8})$,*

$$\left|\sum_{j=1}^n \left(\sum_{k \in I_\alpha} s_k W_{jk}\right)^2 - n|I_\alpha|\sigma^2\right| \leq C\nu_W |I_\alpha| \sqrt{n} \log n. \tag{89}$$

*Proof.* Since $s \in \{\pm 1\}^n$ is fixed and the entries of $W$ are symmetric about zero, without loss of generality, we assume that $s$ is a vector of all 1's. Reindexing if necessary, we assume without loss of generality that $I_\alpha = [K]$, where $K = |I_\alpha| > 0$. We have $K > 0$ since $\alpha \in \mathcal{A}_0$ and by the definition of $\mathcal{A}_0$ in Equation (53), $|I_\alpha| \geq \left(\alpha^2 \log^2 n\right)^{-1} > 0$ . It follows that

$$\sum_{j=1}^n \left(\sum_{k \in I_\alpha} W_{jk}\right)^2 - n|I_\alpha|\sigma^2 = \sum_{j=1}^n \sum_{k=1}^K W_{jk}^2 - nK\sigma^2 + 2\sum_{j=1}^n \sum_{1 \leq k < \ell \leq K} W_{jk} W_{j\ell} \tag{90}$$

$$= \gamma_1 + \gamma_2 + \gamma_3,$$

where

$$\gamma_1 := \sum_{j=K+1}^n \sum_{k=1}^K W_{jk}^2 + \sum_{k=1}^K W_{kk}^2 + 2\sum_{1 \leq j < k \leq K} W_{jk}^2 - nK\sigma^2,$$

$$\gamma_2 := 2\sum_{j=K+1}^n \sum_{1 \leq k < \ell \leq K} W_{jk} W_{j\ell} \quad \text{and}$$

$$\gamma_3 := \sum_{j=1}^K \sum_{1 \leq k \neq \ell \leq K} W_{jk} W_{j\ell}.$$

We will bound these three quantities separately.

We begin by bounding $\gamma_1$ in Equation (90). For ease of notation, define $N = K(n - K/2 + 1/2)$. $\gamma_1$ is a sum of $N$ independent sub-exponential random variables. By Equation (2.18) in [56], for any $t \in (0, 2)$,

$$\mathbb{P}\left(\frac{|\gamma_1|}{N} \geq \nu_W t\right) \leq 2e^{-Nt^2/32}.$$

Taking $t = 16\sqrt{N^{-1} \log n}$ yields that for all suitably large $n$, with probability at least $1 - O(n^{-8})$,

$$|\gamma_1| \leq 16\nu_W \sqrt{N \log n} \leq 16\nu_W \sqrt{Kn \log n}, \tag{91}$$

where the second inequality follows from the trivial bound $N \leq Kn$.

Turning to $\gamma_2$ in Equation (90), define

$$\boldsymbol{A} = \boldsymbol{1}_K \boldsymbol{1}_K^\top - \boldsymbol{I}_K \in \mathbb{R}^{K \times K},$$

where $\boldsymbol{1}_K \in \mathbb{R}^K$ is a vector of all 1's and $\boldsymbol{I}_K$ is the $K \times K$ identity matrix. It follows that

$$\gamma_2 = \sum_{j=K+1}^{n} \sum_{1 \leq k \neq \ell \leq K} W_{jk} W_{j\ell} = \sum_{j=K+1}^{n} \sum_{1 \leq k, \ell \leq K} W_{jk} A_{k\ell} W_{j\ell}.$$

By the Hanson-Wright inequality (see Theorem 6.2.1 in [54]), for $t \geq 0$, we have

$$\mathbb{P}\left( \left| \sum_{j=K+1}^{n} \sum_{1 \leq k, \ell \leq K} W_{jk} A_{k\ell} W_{j\ell} \right| \geq t \right)$$

$$\leq 2 \exp\left\{ -c \min\left( \frac{t^2}{(n-K)\nu_W^2 \|\boldsymbol{A}\|_{\mathrm{F}}^2}, \frac{t}{\sqrt{n-K}\nu_W \|\boldsymbol{A}\|_{\mathrm{F}}} \right) \right\},$$

where $c > 0$ is a universal constant. We note that $\|\boldsymbol{A}\|_{\mathrm{F}} \leq K^2$ by construction. Taking $t = (8\nu_W/c)K\sqrt{n-K}\log n$, with probability at least $1 - O(n^{-8})$ we have

$$|\gamma_2| = \left| \sum_{j=K+1}^{n} \sum_{1 \leq k, \ell \leq K} W_{jk} A_{k\ell} W_{j\ell} \right| \leq \frac{8\nu_W}{c} K\sqrt{n-K}\log n. \tag{92}$$

Finally, to bound $\gamma_3$ in Equation (90), we define

$$\boldsymbol{w} = \mathrm{vech}([W_{ij}]_{1 \leq i, j \leq K}) \in \mathbb{R}^{K(K+1)/2},$$

where $\mathrm{vech}(\cdot)$ is the half-vectorization operator that vectorizes the upper triangular part (including the diagonal) of a given matrix.

We identify the elements of $\boldsymbol{w}$ with the elements of

$$\mathcal{J}_K = \{(i_1, i_2) : 1 \leq i_1 \leq i_2 \leq K\},$$

that is, pairs $(i_1, i_2)$ satisfying $1 \leq i_1 \leq i_2 \leq K$. We note that $\gamma_3$ is a sum of products of the form $w_{(i_1, i_2)} w_{(j_1, j_2)}$ such that $(i_1, i_2), (j_1, j_2) \in \mathcal{J}_K$ and one element of $(i_1, i_2)$ agrees with one element of $(j_1, j_2)$, while the others disagree. For $(i_1, i_2) \in \mathcal{J}_K$ with $i_1 < i_2$, there are $2(K-1)$ other pairs $(j_1, j_2) \in \mathcal{J}_K$ satisfying this requirement, while for $(i_1, i_2) \in \mathcal{J}_K$ with $i_1 = i_2$, there are $(K-1)$ other elements of $\mathcal{J}_K$ satisfying the requirement. In total, there are $K^2(K-1)$ such pairs, which agrees with the number of terms in $\gamma_3$. We define matrix $\boldsymbol{B} \in \mathbb{R}^{K(K+1)/2 \times K(K+1)/2}$ by identifying its rows and columns with elements of $\mathcal{J}_K$ and setting

$$B_{(i_1, i_2), (j_1, j_2)} = \max\left\{ \delta_{i_1 j_1}(1 - \delta_{i_2 j_2}), \delta_{i_1 j_2}(1 - \delta_{i_2 j_1}), \delta_{i_2 j_1}(1 - \delta_{i_1 j_2}), \delta_{i_2 j_2}(1 - \delta_{i_1 j_2}) \right\}$$

for $(i_1, i_2), (j_1, j_2) \in \mathcal{J}_K$, where

$$\delta_{ij} = \begin{cases} 1 & \text{if } i = j, \\ 0 & \text{otherwise.} \end{cases}$$

There are $K^2(K-1)$ entries in $\boldsymbol{B}$ that are equal to 1, while all others are equal to zero. Thus, we have $\|\boldsymbol{B}\|_{\mathrm{F}} = K\sqrt{K-1} \leq K^{3/2}$. By construction, one can verify that

$$\boldsymbol{w}^\top \boldsymbol{B} \boldsymbol{w} = \sum_{(i_1, i_2) \in \mathcal{J}} \sum_{(j_1, j_2) \in \mathcal{J}} w_{(i_1, i_2)} B_{(i_1, i_2), (j_1, j_2)} w_{(j_1, j_2)} = \sum_{j=1}^{K} \sum_{1 \leq k \neq \ell \leq K} W_{jk} W_{j\ell} = \gamma_3.$$

Again, by the Hanson-Wright inequality, for every $t \geq 0$, we have

$$\mathbb{P}\left( \left| \boldsymbol{w}^\top \boldsymbol{B} \boldsymbol{w} \right| \geq t \right) \leq 2 \exp\left\{ -c \min\left( \frac{t^2}{\nu_W^2 \|\boldsymbol{B}\|_{\mathrm{F}}^2}, \frac{t}{\nu_W \|\boldsymbol{B}\|_{\mathrm{F}}} \right) \right\},$$

where $c > 0$ is a universal constant. Taking $t = (8\nu_W/c)\|\boldsymbol{B}\|_\mathrm{F} \log n$ yields that with probability at least $1 - O(n^{-8})$,

$$|\gamma_3| = \left|\boldsymbol{w}^\top \boldsymbol{B}\boldsymbol{w}\right| \leq \frac{8\nu_W}{c}\|\boldsymbol{B}\|_\mathrm{F} \log n \leq \frac{8\nu_W}{c} K^{3/2} \log n, \tag{93}$$

where the second inequality follows from our bound on $\|\boldsymbol{B}\|_\mathrm{F}$.

Finally, plugging the bounds in Equations (91) (92) and (93) into Equation (90), we have that with probability at least $1 - O(n^{-8})$, Equation (89) holds, as we set out to show. $\qquad\square$

**Lemma 11.** *Under Assumption 1, for any fixed $\boldsymbol{s} \in \{\pm 1\}^n$ and any fixed $\boldsymbol{u}^\star \in \mathbb{S}^{n-1}$, for any $\alpha \in \mathcal{A}$, it holds with probability at least $1 - O(n^{-8})$ that*

$$\left|\sum_{k \in I_\alpha} \sum_{j=1}^n u_j^\star s_k W_{jk}\right| \leq 4\sqrt{2\nu_W |I_\alpha| \log n}.$$

*Proof.* Since $\boldsymbol{s} \in \{\pm 1\}^n$ is fixed and the entries of $\boldsymbol{W}$ are symmetric about zero, without loss of generality, we assume that $\boldsymbol{s}$ is a vector of all 1's. Reindexing if necessary, we again assume that $I_\alpha = [K]$ without loss of generality, where $K = |I_\alpha|$. Rearranging sums, we have

$$\sum_{k=1}^K \sum_{j=1}^n u_j^\star s_k W_{jk} = \sum_{k=1}^K \sum_{j=1}^n u_j^\star W_{jk}$$
$$= \sum_{j=1}^K \sum_{k=j+1}^K (u_j^\star + u_k^\star) W_{jk} + \sum_{j=1}^K u_j^\star W_{jj} + \sum_{k=1}^K \sum_{j=K+1}^n u_j^\star W_{jk}.$$

By standard subgaussian concentration inequalities [54, 56], using the fact that $\boldsymbol{u}^\star$ is unit-norm,

$$\mathbb{P}\left(\left|\sum_{k=1}^K \sum_{j=1}^n u_j^\star s_k W_{jk}\right| \geq 4\sqrt{\nu_W 2K \log n}\right) \leq O(n^{-8}),$$

completing the proof. $\qquad\square$

### E.2   Proof of Proposition 1

*Proof.* Similar to the proof of Theorem 1, we assume that $\lambda^\star > 0$ without loss of generality. Recalling the quantity $\widehat{S}$ as defined in Step 4 of Algorithm 1 and the fact that $\widehat{S} > 0$ from Equation (65), we define

$$R_{\hat{I}} = \frac{\sqrt{\lambda^\star} \left(\sum_{k \in \hat{I}} Q_{kk} u_k^\star\right)}{\widehat{S}}, \tag{94}$$

where we recall that $\hat{I}$ is the set given in Step 2 of Algorithm 1 and $\boldsymbol{Q} \in \mathbb{R}^{n \times n}$ is a diagonal matrix given in Step 3 of Algorithm 1.

Plugging in $\widetilde{\boldsymbol{Y}} = \boldsymbol{QYQ}$ and expanding the right hand side of Equation (88), we have

$$\widehat{\lambda} = \frac{1}{\widehat{S}^2} \left(\sum_{j=1}^n \left(\lambda^\star Q_{jj} u_j^\star \sum_{k \in \hat{I}} Q_{kk} u_k^\star + \sum_{k \in \hat{I}} Q_{jj} Q_{kk} W_{jk}\right)^2\right) - \frac{n|\hat{I}|\sigma^2}{\widehat{S}^2}$$
$$= \frac{1}{\widehat{S}^2} \sum_{j=1}^n (\lambda^\star u_j^\star)^2 \left(\sum_{k \in \hat{I}} Q_{kk} u_k^\star\right)^2 + \frac{1}{\widehat{S}^2} \sum_{j=1}^n \left(\sum_{k \in \hat{I}} Q_{kk} W_{jk}\right)^2$$
$$+ \frac{2\lambda^\star}{\widehat{S}^2} \sum_{j=1}^n \left(u_j^\star \sum_{k \in \hat{I}} Q_{kk} u_k^\star \sum_{\ell \in \hat{I}} Q_{\ell\ell} W_{j\ell}\right) - \frac{n|\hat{I}|\sigma^2}{\widehat{S}^2}.$$

Applying the definition of $R_{\hat{I}}$ in Equation (94) and rearranging terms,

$$\widehat{\lambda} = \lambda^{\star} R_{\hat{I}}^2 + R_{\hat{I}}^2 \frac{\sum_{j=1}^{n} \left(\sum_{k \in \hat{I}} Q_{kk} W_{jk}\right)^2 - n|\hat{I}|\sigma^2}{\left(\sqrt{\lambda^{\star}} \sum_{k \in \hat{I}} Q_{kk} u_k^{\star}\right)^2} + 2R_{\hat{I}}^2 \sum_{j=1}^{n} u_j^{\star} \frac{\sum_{k \in \hat{I}} Q_{kk} W_{jk}}{\sum_{k \in \hat{I}} Q_{kk} u_k^{\star}}. \tag{95}$$

To control the term $R_{\hat{I}}$ in Equation (95), we proceed to bound a related term $R_{\alpha}$ for all $\alpha \in \mathcal{A}_0$, defined as

$$R_{\alpha} = \frac{\sqrt{\lambda^{\star}} \left(\sum_{k \in I_{\alpha}} |u_k^{\star}|\right)}{\widehat{S}}.$$

We recall the events $\mathcal{E}_{2,\alpha}$, $\mathcal{E}_{3,\alpha}$ and $\mathcal{E}_{4,\alpha}$ for all $\alpha \in \mathcal{A}_0$ as defined in Equations (60), (61) and (63) in the proof of Theorem 1. For any $\alpha \in \mathcal{A}_0$, to control the term $R_{\alpha}$, we divide $\widehat{S}$ on both sides of Equation (67) stated in the proof of Theorem 1. It follows from Equation (67) that

$$|R_{\alpha} - 1| = \left| \frac{\sqrt{\lambda^{\star}} \sum_{k \in I_{\alpha}} |u_k^{\star}|}{\widehat{S}} - 1 \right| \leq \frac{C\sqrt{\nu_W \log n}}{\alpha \sqrt{\lambda^{\star}} \widehat{S}}$$

holds on the event $\mathcal{E}_{2,\alpha} \cap \mathcal{E}_{3,\alpha} \cap \mathcal{E}_{4,\alpha}$. Following the previous bound, on the event $\mathcal{E}_{2,\alpha} \cap \mathcal{E}_{3,\alpha} \cap \mathcal{E}_{4,\alpha}$, we have that

$$|R_{\alpha} - 1| \leq \frac{C\sqrt{\nu_W \log n}}{\alpha \sqrt{\lambda^{\star}} \widehat{S}} \leq \frac{C\sqrt{\nu_W \log n}}{\alpha \lambda^{\star} \left(\sum_{k \in I_{\alpha}} |u_k^{\star}|\right) - \alpha \sqrt{\lambda^{\star}} \left| \widehat{S} - \sqrt{\lambda^{\star}} \sum_{k \in I_{\alpha}} |u_k^{\star}| \right|}$$

$$\leq \frac{C\sqrt{\nu_W \log n}}{\lambda^{\star} |I_{\alpha}| \alpha^2 - C\sqrt{\nu_W \log n}} \leq \frac{C\sqrt{\nu_W \log^{5/2} n}}{\lambda^{\star} - C\sqrt{\nu_W \log^{5/2} n}},$$

where the second inequality holds by the triangle inequality, the third inequality follows from Equations (66) and (52), the last inequality holds by Equation (53). We remind the reader that we allow constant $C$ to change its precise value from line to line in the proof. Under Assumption 3, it follows from the previous bound that for $n$ sufficiently large, we have on the event $\mathcal{E}_{2,\alpha} \cap \mathcal{E}_{3,\alpha} \cap \mathcal{E}_{4,\alpha}$ that

$$|R_{\alpha} - 1| \leq \frac{C\sqrt{\nu_W \log^{5/2} n}}{\lambda^{\star}}. \tag{96}$$

By Equation (96) and Assumption 3, for $n$ sufficiently large, we have

$$1/2 \leq R_{\alpha} \leq 2 \tag{97}$$

on the event $\mathcal{E}_{2,\alpha} \cap \mathcal{E}_{3,\alpha} \cap \mathcal{E}_{4,\alpha}$.

Recall $s \in \{\pm 1\}^n$ as defined in Equation (62). On the event $\mathcal{E}_{2,\alpha} \cap \mathcal{E}_{3,\alpha} \cap \mathcal{E}_{4,\alpha}$, we find that, using Equation (95) to substitute for $\widehat{\lambda}$ and applying the triangle inequality,

$$\left| \widehat{\lambda} - \lambda^{\star} \right| \leq \lambda^{\star} \left| R_{\alpha}^2 - 1 \right| + R_{\alpha}^2 \frac{\left| \sum_{j=1}^{n} \left(\sum_{k \in I_{\alpha}} s_k W_{jk}\right)^2 - n|I_{\alpha}|\sigma^2 \right|}{\left(\sqrt{\lambda^{\star}} \sum_{k \in I_{\alpha}} |u_k^{\star}|\right)^2}$$

$$+ 2R_{\alpha}^2 \left| \sum_{j=1}^{n} u_j^{\star} \frac{\sum_{k \in I_{\alpha}} s_k W_{jk}}{\sum_{k \in I_{\alpha}} |u_k^{\star}|} \right|,$$

where we substitute $R_{\hat{I}}$ by $R_{\alpha}$ using the fact that on $\mathcal{E}_{2,\alpha}$, we have $\hat{I} = I_{\alpha}$. Continuing from the previous bound, it follows from Equation (96) that

$$\left| \widehat{\lambda} - \lambda^{\star} \right| \leq \lambda^{\star} \cdot \frac{C\sqrt{\nu_W \log^{5/2} n}}{\lambda^{\star}} \cdot |R_{\alpha} + 1| + R_{\alpha}^2 \frac{\left| \sum_{j=1}^{n} \left(\sum_{k \in I_{\alpha}} s_k W_{jk}\right)^2 - n|I_{\alpha}|\sigma^2 \right|}{\left(\sqrt{\lambda^{\star}} \sum_{k \in I_{\alpha}} |u_k^{\star}|\right)^2}$$

$$+ 2R_{\alpha}^2 \left| \sum_{j=1}^{n} u_j^{\star} \frac{\sum_{k \in I_{\alpha}} s_k W_{jk}}{\sum_{k \in I_{\alpha}} |u_k^{\star}|} \right|.$$

Applying Equation (97) to $R_\alpha$ in the above display, we have

$$\left|\widehat{\lambda} - \lambda^\star\right| \leq C\sqrt{\nu_W} \log^{5/2} n + \frac{C\left|\sum_{j=1}^n \left(\sum_{k \in I_\alpha} s_k W_{jk}\right)^2 - n|I_\alpha|\sigma^2\right|}{\left(\sqrt{\lambda^\star} \sum_{k \in I_\alpha} |u_k^\star|\right)^2} \tag{98}$$
$$+ C\left|\frac{\sum_{k \in I_\alpha} \sum_{j=1}^n u_j^\star s_k W_{jk}}{\sum_{k \in I_\alpha} |u_k^\star|}\right|.$$

For all $\alpha \in \mathcal{A}_0$, define the events

$$\mathcal{G}_{1,\alpha} = \left\{\left|\sum_{j=1}^n \left(\sum_{k \in I_\alpha} s_k W_{jk}\right)^2 - n|I_\alpha|\sigma^2\right| \leq C\nu_W |I_\alpha| \sqrt{n} \log n\right\} \quad \text{and}$$
$$\mathcal{G}_{2,\alpha} = \left\{\left|\sum_{k \in I_\alpha} \sum_{j=1}^n u_j^\star s_k W_{jk}\right| \leq 4\sqrt{2\nu_W |I_\alpha| \log n}\right\}. \tag{99}$$

On the event $\mathcal{E}_{2,\alpha} \cap \mathcal{E}_{3,\alpha} \cap \mathcal{E}_{4,\alpha} \cap \mathcal{G}_{1,\alpha} \cap \mathcal{G}_{2,\alpha}$, we can further bound Equation (98) according to

$$\left|\widehat{\lambda} - \lambda^\star\right| \leq C\sqrt{\nu_W} \log^{5/2} n + \frac{C\nu_W |I_\alpha| \sqrt{n} \log n}{\left(\sqrt{\lambda^\star} \sum_{k \in I_\alpha} |u_k^\star|\right)^2} + \frac{C\sqrt{\nu_W |I_\alpha| \log n}}{\sum_{k \in I_\alpha} |u_k^\star|},$$

following the definition of $\mathcal{G}_{1,\alpha}$ and $\mathcal{G}_{2,\alpha}$ in Equation (99). By the definition of $I_\alpha$ in Equation (52), it follows that

$$\left|\widehat{\lambda} - \lambda^\star\right| \leq C\sqrt{\nu_W} \log^{5/2} n + \frac{C\nu_W |I_\alpha| \sqrt{n} \log n}{\lambda^\star |I_\alpha|^2 \alpha^2} + \frac{C\sqrt{\nu_W |I_\alpha| \log n}}{|I_\alpha|\alpha}$$

holds on the event $\mathcal{E}_{2,\alpha} \cap \mathcal{E}_{3,\alpha} \cap \mathcal{E}_{4,\alpha} \cap \mathcal{G}_{1,\alpha} \cap \mathcal{G}_{2,\alpha}$. Applying Equation (53) to lower bound $|I_\alpha|$, we have

$$\left|\widehat{\lambda} - \lambda^\star\right| \leq C\sqrt{\nu_W} \log^{5/2} n + \frac{C\nu_W \sqrt{n} \log^3 n}{\lambda^\star} + C\sqrt{\nu_W} \log^{3/2} n \leq C\sqrt{\nu_W} \log^{5/2} n, \tag{100}$$

holds on the event $\mathcal{E}_{2,\alpha} \cap \mathcal{E}_{3,\alpha} \cap \mathcal{E}_{4,\alpha} \cap \mathcal{G}_{1,\alpha} \cap \mathcal{G}_{2,\alpha}$, where the last inequality holds under Assumption 3 when $n$ is sufficiently large.

Consider the event

$$\mathcal{G} = \left\{\left|\widehat{\lambda} - \lambda^\star\right| \leq C\sqrt{\nu_W} \log^{5/2} n\right\}.$$

By the proof leading to Equation (100), we have for all $\alpha \in \mathcal{A}_0$,
$$\mathcal{E}_{2,\alpha} \cap \mathcal{E}_{3,\alpha} \cap \mathcal{E}_{4,\alpha} \cap \mathcal{G}_{1,\alpha} \cap \mathcal{G}_{2,\alpha} \subseteq \mathcal{G}.$$

It follows that

$$\mathbb{P}(\mathcal{G}) \geq \mathbb{P}\left(\bigcup_{\alpha \in \mathcal{A}_0} \mathcal{E}_{2,\alpha} \cap \mathcal{E}_{3,\alpha} \cap \mathcal{E}_{4,\alpha} \cap \mathcal{G}_{1,\alpha} \cap \mathcal{G}_{2,\alpha}\right)$$
$$= \sum_{\alpha \in \mathcal{A}_0} \mathbb{P}\left(\mathcal{E}_{2,\alpha} \cap \mathcal{E}_{3,\alpha} \cap \mathcal{E}_{4,\alpha} \cap \mathcal{G}_{1,\alpha} \cap \mathcal{G}_{2,\alpha}\right),$$

where the first inequality follows from set inclusion and the last equality follows from the fact that $\{\mathcal{E}_{2,\alpha}\}_{\alpha \in \mathcal{A}_0}$ are disjoint events according to Equation (60). By basic set inclusions, it follows that

$$\mathbb{P}(\mathcal{G}) \geq \sum_{\alpha \in \mathcal{A}_0} \left(\mathbb{P}(\mathcal{E}_{2,\alpha}) - \mathbb{P}(\mathcal{E}_{2,\alpha} \cap \mathcal{E}_{3,\alpha}^c) - \mathbb{P}(\mathcal{E}_{4,\alpha}^c) - \sum_{j=1}^2 \mathbb{P}(\mathcal{G}_{j,\alpha}^c)\right).$$

Applying Lemma 9, Equations (83), Lemma 10 and Lemma 11 to the above display yields that

$$\mathbb{P}(\mathcal{G}) \geq \sum_{\alpha \in \mathcal{A}_0} \mathbb{P}(\mathcal{E}_{2,\alpha}) - O(|\mathcal{A}_0|n^{-8}) = 1 - \mathbb{P}\left(\bigcap_{\alpha \in \mathcal{A}_0} \left\{\widehat{I} \neq I_\alpha\right\}\right) - O(|\mathcal{A}_0|n^{-8}).$$

Applying Lemma 7 and using the fact that $|\mathcal{A}_0| \leq |\mathcal{A}|$ and $|\mathcal{A}| = O(\log n)$ by its definition in Equation (10), we have

$$\mathbb{P}(\mathcal{G}) \geq 1 - O(n^{-8}) - O(|\mathcal{A}_0|n^{-8}) \geq 1 - O(n^{-8} \log n),$$

completing the proof. $\qquad\qquad\square$

# F  Proofs for lower bounds of metric entropy under $d_{2,\infty}$

To prove Lemma 4, we first state a few technical lemmas.

## F.1  Technical Lemmas

For a semi-metric $\rho$ defined over a set $\mathbb{K}$, we let $\mathcal{N}(\mathbb{K}, \rho, \delta)$ denote the $\delta$-covering number of $\mathbb{K}$ under $\rho$ (see Chapter 15 in [56]). The collection of all linear subspaces of fixed dimension $r$ of the Euclidean space $\mathbb{R}^n$ forms the Grassmann manifold $\mathbb{G}_{n,r}$, also termed the Grassmannian. For points on the Grassmannian, we adopt the projector perspective [13]: a subspace $\mathcal{U} \in \mathbb{G}_{n,r}$ is identified with the (unique) orthogonal projector $\boldsymbol{P} \in \mathbb{R}^{n \times n}$ onto $\mathcal{U}$, which in turn is uniquely represented by $\boldsymbol{P} = \boldsymbol{U}\boldsymbol{U}^T$, where $\boldsymbol{U} \in \mathbb{R}^{n \times r}$ whose columns form an orthonormal basis of $\mathcal{U}$. For any matrix $\boldsymbol{A} \in \mathbb{R}^{m \times n}$ with singular values denoted by $\sigma_i(\boldsymbol{A})$ for $1 \leq i \leq \min\{m,n\}$, the Schatten-$q$ norm $\|\cdot\|_{S_q}$ is defined for any $1 \leq q \leq \infty$

$$\|\boldsymbol{A}\|_{S_q} := \left( \sum_{i=1}^{\min\{m,n\}} \sigma_i^q(\boldsymbol{A}) \right)^{1/q}.$$

For a pair of subspaces $\mathcal{U}_1, \mathcal{U}_2 \in \mathbb{G}_{n,r}$ identified with projectors $\boldsymbol{P}_1 = \boldsymbol{U}_1\boldsymbol{U}_1^\top, \boldsymbol{P}_2 = \boldsymbol{U}_2\boldsymbol{U}_2^\top \in \mathbb{R}^{n \times n}$, respectively, we consider the distance $d_{S_q}(\cdot, \cdot)$ induced by the Schatten-$q$ norm

$$d_{S_q}(\mathcal{U}_1, \mathcal{U}_2) := \|\boldsymbol{P}_1 - \boldsymbol{P}_2\|_{S_q} = \|\boldsymbol{U}_1\boldsymbol{U}_1^\top - \boldsymbol{U}_2\boldsymbol{U}_2^\top\|_{S_q}. \tag{101}$$

Lemma 12 controls the covering number of $\mathbb{G}_{n,r}$ under $d_{S_q}$.

**Lemma 12** ([46] Proposition 8). *For any integer $1 \leq r \leq n$ such that $2r \leq n$, any $q$ such that $1 \leq q \leq \infty$ and any $\delta > 0$, we have*

$$\left( \frac{c_0}{\delta} \right)^{r(n-r)} \leq \mathcal{N}(\mathbb{G}_{n,r}, \ d_{S_q}, \ \delta r^{1/q}) \leq \left( \frac{C_0}{\delta} \right)^{r(n-r)}$$

*where $c_0, C_0 > 0$ are universal constants, $d_{S_q}$ is the distance defined in Equation (101) induced by the Schatten-$q$ norm.*

Let $\mathbb{V}_{n,r} := \{\boldsymbol{U} \in \mathbb{R}^{n \times r} : \boldsymbol{U}^\top \boldsymbol{U} = \boldsymbol{I}_r\}$ be the $n \times r$ Stiefel manifold [16]. The distance $d_{S_q}(\cdot, \cdot)$ can also be viewed as a distance defined on $\mathbb{V}_{n,r}$. For a pair of orthogonal matrices $\boldsymbol{U}_1, \boldsymbol{U}_2 \in \mathbb{V}_{n,r}$, we let

$$d_{S_q}(\boldsymbol{U}_1, \boldsymbol{U}_2) := \|\boldsymbol{U}_1\boldsymbol{U}_1^\top - \boldsymbol{U}_2\boldsymbol{U}_2^\top\|_{S_q}.$$

When $q = 2$, the Schatten-$q$ norm coincides with the Frobenius norm, and we have

$$d_{S_2}(\boldsymbol{U}_1, \boldsymbol{U}_2) = \left\| \boldsymbol{U}_1\boldsymbol{U}_1^\top - \boldsymbol{U}_2\boldsymbol{U}_2^\top \right\|_{\mathrm{F}}.$$

Define a distance $d_{\mathrm{F}}$ over $\mathbb{V}_{n,r}$ as

$$d_{\mathrm{F}}(\boldsymbol{U}_1, \boldsymbol{U}_2) := \min_{\boldsymbol{\Gamma} \in \mathbb{O}_r} \|\boldsymbol{U}_1 - \boldsymbol{U}_2\boldsymbol{\Gamma}\|_{\mathrm{F}}. \tag{102}$$

Lemma 13 controls the covering number of $\mathbb{V}_{n,r}$ under $d_{\mathrm{F}}$, which follows immediately from Lemma 12.

**Lemma 13.** *For any integer $1 \leq r \leq n/2$ and for every $\delta > 0$, we have*

$$\left( \frac{c_0 \sqrt{r}}{\delta} \right)^{r(n-r)} \leq \mathcal{N}(\mathbb{V}_{n,r}, d_{\mathrm{F}}, \delta) \leq \left( \frac{C_0 \sqrt{2r}}{\delta} \right)^{r(n-r)},$$

*where $c_0, C_0 > 0$ are universal constants and $d_{\mathrm{F}}$ is the*

*Proof of Lemma 13.* We identify each element $\boldsymbol{U}$ of $\mathbb{V}_{n,r}$ with the element $\boldsymbol{U}\boldsymbol{U}^\top$ in $\mathbb{G}_{n,r}$ and apply Lemma 12 to $\mathbb{V}_{n,r}$ under $d_{\mathrm{F}}$. By the fact that (see Lemma 2.6 in [27])

$$\frac{1}{\sqrt{2}} \left\| \boldsymbol{U}_1\boldsymbol{U}_1^\top - \boldsymbol{U}_2\boldsymbol{U}_2^\top \right\|_{\mathrm{F}} \leq d_{\mathrm{F}}(\boldsymbol{U}_1, \boldsymbol{U}_2) \leq \left\| \boldsymbol{U}_1\boldsymbol{U}_1^\top - \boldsymbol{U}_2\boldsymbol{U}_2^\top \right\|_{\mathrm{F}},$$

we have

$$\mathcal{N}\left(\mathbb{V}_{n,r}, d_{S_2}, \delta\right) \leq \mathcal{N}(\mathbb{V}_{n,r}, d_{\mathrm{F}}, \delta) \leq \mathcal{N}\left(\mathbb{V}_{n,r}, d_{S_2}, \frac{\delta}{\sqrt{2}}\right)$$

and it follows from Lemma 12 that

$$\left(\frac{c_0\sqrt{r}}{\delta}\right)^{r(n-r)} \leq \mathcal{N}(\mathbb{V}_{s,r}, d_{\mathrm{F}}, \delta) \leq \left(\frac{C_0\sqrt{2r}}{\delta}\right)^{r(n-r)},$$

completing the proof. $\qquad\qquad\qquad\square$

For any $s \geq r$, Lemma 14 relates the packing number of any subset $\mathbb{K} \subseteq \mathbb{V}_{s,r}$ to its Haar measure and the covering number of $\mathbb{V}_{s,r}$. Recall that the Haar measure on $\mathbb{V}_{s,r}$ is the invariant measure under both left- and right-orthogonal transformation [29]. In other words, for any subset $\mathbb{K} \subseteq \mathbb{V}_{s,r}$ and orthogonal matrices $\boldsymbol{\Gamma}_1 \in \mathbb{O}_s$ and $\boldsymbol{\Gamma}_2 \in \mathbb{O}_r$, we have

$$\xi_H\left(\boldsymbol{\Gamma}_1 \cdot \mathbb{K}\right) = \xi_H\left(\mathbb{K}\right) \quad \text{and} \quad \xi_H\left(\mathbb{K} \cdot \boldsymbol{\Gamma}_2\right) = \xi_H\left(\mathbb{K}\right),$$

where

$$\Gamma_1 \cdot \mathbb{K} := \{\boldsymbol{\Gamma}_1\boldsymbol{U} : \boldsymbol{U} \in \mathbb{K}\} \quad \text{and} \quad \mathbb{K} \cdot \boldsymbol{\Gamma}_2 := \{\boldsymbol{U}\boldsymbol{\Gamma}_2 : \boldsymbol{U} \in \mathbb{K}\}.$$

For any norm $\vert\!\vert\!\vert\cdot\vert\!\vert\!\vert$ defined over a set $\mathbb{K}$, we also use the notation $\mathcal{M}(\mathbb{K}, \vert\!\vert\!\vert\cdot\vert\!\vert\!\vert, \delta)$ and $\mathcal{N}(\mathbb{K}, \vert\!\vert\!\vert\cdot\vert\!\vert\!\vert, \delta)$ to denote the $\delta$-packing number and $\delta$-covering number of $\mathbb{K}$ under the distance induced by $\vert\!\vert\!\vert\cdot\vert\!\vert\!\vert$, respectively. With the above setup, we state Lemma 14 below.

**Lemma 14.** *For $1 \leq r \leq s$, let $\xi_H$ denote the Haar measure on $\mathbb{V}_{s,r}$. For $1 \leq r \leq s$ and any $\mathbb{K} \subseteq \mathbb{V}_{s,r}$ such that $\xi_H(\mathbb{K}) \geq \gamma > 0$, for any unitarily invariant norm $\vert\!\vert\!\vert\cdot\vert\!\vert\!\vert$ on $\mathbb{R}^{s\times r}$, we have*

$$\mathcal{M}\left(\mathbb{K}, \vert\!\vert\!\vert\cdot\vert\!\vert\!\vert, \frac{\delta}{2}\right) \geq \gamma\mathcal{N}\left(\mathbb{V}_{s,r}, \vert\!\vert\!\vert\cdot\vert\!\vert\!\vert, \delta\right), \tag{103}$$

*and*

$$\mathcal{M}\left(\mathbb{K}, \vert\!\vert\!\vert\cdot\vert\!\vert\!\vert, \frac{\delta}{2}\right) \leq \mathcal{N}\left(\mathbb{V}_{s,r}, \vert\!\vert\!\vert\cdot\vert\!\vert\!\vert, \frac{\delta}{8}\right). \tag{104}$$

*Proof of Lemma 14.* The bound in Equation (104) follows from the fact that $\mathbb{K} \subseteq \mathbb{V}_{s,r}$, and (see Exercise 4.2.10 in [54]),

$$\mathcal{M}\left(\mathbb{K}, \vert\!\vert\!\vert\cdot\vert\!\vert\!\vert, \frac{\delta}{2}\right) \leq \mathcal{N}\left(\mathbb{K}, \vert\!\vert\!\vert\cdot\vert\!\vert\!\vert, \frac{\delta}{4}\right) \leq \mathcal{N}\left(\mathbb{V}_{s,r}, \vert\!\vert\!\vert\cdot\vert\!\vert\!\vert, \frac{\delta}{8}\right).$$

To establish the bound in Equation (103), suppose that $\mathbb{V}_{s,r}$ has a $\delta$-packing set $\mathcal{P} = \{\boldsymbol{U}^{(i)}\}_{i=1}^{M}$ under $\vert\!\vert\!\vert\cdot\vert\!\vert\!\vert$, then

$$\bigcup_{i=1}^{M} \mathbb{B}\left(\boldsymbol{U}^{(i)}, \vert\!\vert\!\vert\cdot\vert\!\vert\!\vert, \frac{\delta}{2}\right) \subseteq \mathbb{V}_{s,r}$$

where

$$\mathbb{B}\left(\boldsymbol{U}^{(i)}, \vert\!\vert\!\vert\cdot\vert\!\vert\!\vert, \frac{\delta}{2}\right) := \left\{\boldsymbol{U} \in \mathbb{V}_{s,r} : \left\vert\!\left\vert\!\left\vert\boldsymbol{U} - \boldsymbol{U}^{(i)}\right\vert\!\right\vert\!\right\vert \leq \delta/2\right\}.$$

Since $\mathcal{P}$ is a $\delta$-packing set, the sets $\mathbb{B}\left(\boldsymbol{U}^{(i)}, \vert\!\vert\!\vert\cdot\vert\!\vert\!\vert, \delta/2\right)$ for $i = 1, 2, \ldots, M$ are disjoint and we have

$$\sum_{i=1}^{M} \xi_H\left(\mathbb{B}\left(\boldsymbol{U}^{(i)}, \vert\!\vert\!\vert\cdot\vert\!\vert\!\vert, \frac{\delta}{2}\right)\right) = \xi_H\left(\bigcup_{i=1}^{M} \mathbb{B}\left(\boldsymbol{U}^{(i)}, \vert\!\vert\!\vert\cdot\vert\!\vert\!\vert, \frac{\delta}{2}\right)\right) \leq \xi_H(\mathbb{V}_{s,r}) = 1. \tag{105}$$

For any $\boldsymbol{U}, \widetilde{\boldsymbol{U}} \in \mathbb{V}_{s,r}$, there exists $\boldsymbol{\Gamma} \in \mathbb{O}_s$ such that $\boldsymbol{\Gamma}\boldsymbol{U} = \widetilde{\boldsymbol{U}}$. Since $\vert\!\vert\!\vert\cdot\vert\!\vert\!\vert$ is invariant under $\mathbb{O}_s$, we have

$$\boldsymbol{\Gamma} \cdot \mathbb{B}\left(\boldsymbol{U}, \vert\!\vert\!\vert\cdot\vert\!\vert\!\vert, \frac{\delta}{2}\right) = \mathbb{B}\left(\boldsymbol{\Gamma}\boldsymbol{U}, \vert\!\vert\!\vert\cdot\vert\!\vert\!\vert, \frac{\delta}{2}\right) = \mathbb{B}\left(\widetilde{\boldsymbol{U}}, \vert\!\vert\!\vert\cdot\vert\!\vert\!\vert, \frac{\delta}{2}\right).$$

Since $\xi_H$ is invariant under left-orthogonal transformation, it follows that

$$\xi_H\left(\mathbb{B}\left(\boldsymbol{U}, \vert\!\vert\!\vert\cdot\vert\!\vert\!\vert, \frac{\delta}{2}\right)\right) = \xi_H\left(\boldsymbol{\Gamma} \cdot \mathbb{B}\left(\boldsymbol{U}, \vert\!\vert\!\vert\cdot\vert\!\vert\!\vert, \frac{\delta}{2}\right)\right) = \xi_H\left(\mathbb{B}\left(\widetilde{\boldsymbol{U}}, \vert\!\vert\!\vert\cdot\vert\!\vert\!\vert, \frac{\delta}{2}\right)\right).$$

Therefore, all $\|\!|\!|\cdot\|\!|\!|$-balls with the same radius have the same Haar measure, which does not depend on the particular choice of $\boldsymbol{U}$ due to the invariance properties of $\|\!|\!|\cdot\|\!|\!|$ and $\xi_H$. We denote the Haar measure of a $\|\!|\!|\cdot\|\!|\!|$-balls with radius $\delta/2$ as

$$\psi(\delta/2) := \xi_H\left(\mathbb{B}\left(\boldsymbol{U}, \|\!|\!|\cdot\|\!|\!|, \frac{\delta}{2}\right)\right).$$

It then follows from Equation (105) that

$$M\psi(\delta/2) = \sum_{i=1}^{M} \xi_H\left(\mathbb{B}\left(\boldsymbol{U}^{(i)}, \|\!|\!|\cdot\|\!|\!|, \frac{\delta}{2}\right)\right) \leq 1,$$

from which we conclude that $M \leq 1/\psi(\delta/2)$. Taking the maximal such $\delta$-packing set yields that

$$\mathcal{M}\left(\mathbb{V}_{s,r}, \|\!|\!|\cdot\|\!|\!|, \delta\right) \leq \frac{1}{\psi(\delta/2)}.$$

Recall that $\mathbb{K}$ is a subset of $\mathbb{V}_{s,r}$ and $\xi_H(\mathbb{K}) \geq \gamma$. Suppose that $\mathscr{C}_{\mathbb{K}} = \{\boldsymbol{V}^{(i)}\}_{i=1}^{N}$ is a $\delta/2$-covering set of $\mathbb{K}$ under $\|\!|\!|\cdot\|\!|\!|$, then

$$\mathbb{K} \subseteq \bigcup_{i=1}^{N} \mathbb{B}\left(\boldsymbol{V}^{(i)}, \|\!|\!|\cdot\|\!|\!|, \frac{\delta}{2}\right)$$

and by the subadditivity of measures,

$$\sum_{i=1}^{N} \xi_H\left(\mathbb{B}\left(\boldsymbol{V}^{(i)}, \|\!|\!|\cdot\|\!|\!|, \frac{\delta}{2}\right)\right) \geq \xi_H\left(\bigcup_{i=1}^{N} \mathbb{B}\left(\boldsymbol{V}^{(i)}, \|\!|\!|\cdot\|\!|\!|, \frac{\delta}{2}\right)\right) \geq \xi_H(\mathbb{K}) \geq \gamma.$$

Thus, it follows that

$$N\psi(\delta/2) \geq \gamma \implies N \geq \gamma\mathcal{M}\left(\mathbb{V}_{s,r}, \|\!|\!|\cdot\|\!|\!|, \delta\right).$$

Taking the maximal $\delta/2$-covering set yields that

$$\mathcal{N}\left(\mathbb{K}, \|\!|\!|\cdot\|\!|\!|, \frac{\delta}{2}\right) \geq \gamma\mathcal{M}\left(\mathbb{V}_{s,r}, \|\!|\!|\cdot\|\!|\!|, \delta\right).$$

Finally, by Lemma 4.2.8 in [54], we have

$$\mathcal{M}\left(\mathbb{K}, \|\!|\!|\cdot\|\!|\!|, \frac{\delta}{2}\right) \geq \mathcal{N}\left(\mathbb{K}, \|\!|\!|\cdot\|\!|\!|, \frac{\delta}{2}\right).$$

and

$$\mathcal{M}\left(\mathbb{V}_{s,r}, \|\!|\!|\cdot\|\!|\!|, \delta\right) \geq \mathcal{N}\left(\mathbb{V}_{s,r}, \|\!|\!|\cdot\|\!|\!|, \delta\right)$$

Combining the three displays above yields Equation (103), completing the proof. $\square$

Lemma 15 controls the Haar measure of $\mathbb{K}(s, r, \sqrt{\mu r/n})$. We remind the reader that $\mathbb{K}(s, r, \sqrt{\mu r/n})$ is defined in Equation (20).

**Lemma 15.** *For* $n/\mu \geq \max\{4, r\}$ *and* $s \geq (12n/\mu)\log(12n/\mu)$ *we have*

$$\xi_H(\mathbb{K}(s, r, \sqrt{\mu r/n})) \geq \frac{1}{2}.$$

*Proof of Lemma 15.* Define the set

$$\mathbb{L}\left(s, r, \sqrt{\frac{\mu}{n}}\right) = \left\{\boldsymbol{U} \in \mathbb{V}_{s,r} : \|\boldsymbol{U}\|_\infty \geq \sqrt{\frac{\mu}{n}}\right\},$$

where $\|\boldsymbol{U}\|_\infty$ denotes the entrywise $\ell_\infty$ norm (i.e., $\|\boldsymbol{U}\|_\infty := \max_{i,j}|U_{i,j}|$). In what follows, we omit the parameters and abbreviate $\mathbb{L}(s, r, \sqrt{\mu/n})$ to $\mathbb{L}$. Noting that for any $\boldsymbol{U} \in \mathbb{L}^c$, we have $\|\boldsymbol{U}\|_\infty \leq \sqrt{\mu/n}$, which implies that $\|\boldsymbol{U}\|_{2,\infty} \leq \sqrt{\mu r/n}$. Therefore, $\mathbb{L}^c \subseteq \mathbb{K}(s, r, \sqrt{\mu r/n})$ and

$$\xi_H\left(\mathbb{K}(s, r, \sqrt{\mu r/n})\right) \geq 1 - \xi_H(\mathbb{L}). \tag{106}$$

Since
$$\mathbb{L} = \bigcup_{i=1}^{r} \left\{ \boldsymbol{U} \in \mathbb{V}_{s,r} : \|\boldsymbol{U}_{\cdot,i}\|_{\infty} \geq \sqrt{\frac{\mu}{n}} \right\},$$

it follows that
$$\xi_H(\mathbb{L}) \leq r\, \xi_H\left( \left\{ \boldsymbol{U} \in \mathbb{V}_{s,r} : \|\boldsymbol{U}_{\cdot,1}\|_{\infty} \geq \sqrt{\frac{\mu}{n}} \right\} \right).$$

Since $\boldsymbol{U}$ is distributed according to the Haar measure on $\mathbb{V}_{s,r}$, the column $\boldsymbol{U}_{\cdot,1} \in \mathbb{R}^s$ is uniformly distribution on $\mathbb{S}^{s-1}$. As a result, we have
$$\xi_H\left( \left\{ \boldsymbol{U} \in \mathbb{V}_{s,r} : \|\boldsymbol{U}_{\cdot,1}\|_{\infty} \geq \sqrt{\frac{\mu}{n}} \right\} \right) = \mathbb{P}\left( \frac{\|\boldsymbol{g}\|_{\infty}}{\|\boldsymbol{g}\|_2} \geq \sqrt{\frac{\mu}{n}} \right),$$

where $\boldsymbol{g} \sim N(0, \boldsymbol{I}_s)$ (see Lemma 10.1 in [44]). Combining the above two displays,
$$\xi_H(\mathbb{L}) \leq r\, \mathbb{P}\left( \frac{\|\boldsymbol{g}\|_{\infty}}{\|\boldsymbol{g}\|_2} \geq \sqrt{\frac{\mu}{n}} \right). \tag{107}$$

We introduce two events
$$\mathcal{E}_{0,1} := \left\{ \|\boldsymbol{g}\|_2^2 \geq \frac{3n}{\mu} \log s \right\} \quad \text{and} \quad \mathcal{E}_{0,2} := \left\{ \|\boldsymbol{g}\|_{\infty} \leq \sqrt{3 \log s} \right\}.$$

By Lemma 10.2 in [44], we have for all $t > 0$
$$\mathbb{P}\left( \|\boldsymbol{g}\|_2^2 \leq s - 2\sqrt{st} \right) \leq \exp(-t).$$

Taking $t = 2 \log s$ yields that
$$\mathbb{P}\left( \|\boldsymbol{g}\|_2^2 \leq s - 2\sqrt{2s \log s} \right) \leq \frac{1}{s^2}.$$

When $n \geq 4\mu$ and $s \geq (12n/\mu) \log(12n/\mu)$, one can verify that
$$s - 2\sqrt{2s \log s} \geq s/2 \geq \frac{3n}{\mu} \log s.$$

Thus, it follows that
$$\mathbb{P}(\mathcal{E}_{0,1}^c) \leq \mathbb{P}\left( \|\boldsymbol{g}\|_2^2 \leq s - 2\sqrt{2s \log s} \right) \leq \frac{1}{s^2}. \tag{108}$$

By standard Gaussian concentration inequalities and the union bound, we have
$$\mathbb{P}\left( \mathcal{E}_{0,2}^c \right) = \mathbb{P}\left( \|\boldsymbol{g}\|_{\infty} \geq \sqrt{3 \log s} \right) \leq 2s \exp(-3 \log s) = \frac{2}{s^2}. \tag{109}$$

On the event $\mathcal{E}_{0,1} \cap \mathcal{E}_{0,2}$, one has
$$\frac{\|\boldsymbol{g}\|_{\infty}}{\|\boldsymbol{g}\|_2} \leq \frac{\sqrt{3 \log s}}{\sqrt{3n \log s/\mu}} = \sqrt{\frac{\mu}{n}}.$$

Combining Equations (108) and Equations (109), it follows that
$$\mathbb{P}\left( \frac{\|\boldsymbol{g}\|_{\infty}}{\|\boldsymbol{g}\|_2} \geq \sqrt{\frac{\mu}{n}} \right) \leq \mathbb{P}\left( \mathcal{E}_{0,1}^c \cup \mathcal{E}_{0,2}^c \right) \leq \frac{3}{s^2}.$$

Applying this bound to Equation (107), we obtain
$$\xi_H(\mathbb{L}) \leq r\, \mathbb{P}\left( \frac{\|\boldsymbol{g}\|_{\infty}}{\|\boldsymbol{g}\|_2} \geq \sqrt{\frac{\mu}{n}} \right) \leq \frac{3r}{s^2}. \tag{110}$$

By the assumption that $n/\mu \geq \max\{4, r\}$, for any $r \geq 1$ we have
$$s \geq (12n/\mu) \log(12n/\mu) > 12r > \sqrt{6r}.$$

Combining the above bound with Equation (110), we have
$$\xi_H(\mathbb{L}) \leq \frac{1}{2}.$$

Finally, applying this upper bound to Equation (106), we have
$$\xi_H\left( \mathbb{K}(s, r, \sqrt{\mu r/n}) \right) \geq 1 - \xi_H(\mathbb{L}) \geq \frac{1}{2},$$

as desired. $\qquad \square$

### F.2 Proof of Lemma 4

*Proof.* Set $s = \lceil c_0^2 r / 8e^2 \delta^2 \rceil$. Under our upper bound assumption on $\delta$ in Equation (21) and $n/\mu \geq \max\{4, r\}$, we have

$$s \geq \frac{c_0^2 r}{8e^2} \cdot \frac{96e^2 n \log(12n/\mu)}{c_0^2 \mu r} = (12n/\mu) \log(12n/\mu) > 12r, \tag{111}$$

so that we always have

$$s \geq \max\left\{ \frac{12n}{\mu} \log\left(\frac{12n}{\mu}\right), \frac{c_0^2 r}{8e^2 \delta^2} \right\}. \tag{112}$$

Note that owing to our assumptions that $\mu \geq 12 \log(12n)$ and $\delta^2 \geq c_0^2 r / 8e^2 n$, we have $s \leq n$. Consider the subset

$$\mathbb{K}_s(n, r, \sqrt{\mu r/n}) := \left\{ \boldsymbol{U} \in \mathbb{K}_{r,\mu} : \boldsymbol{U}_{i,\cdot} = 0, \text{ for } s+1 \leq i \leq n \right\},$$

which is a subset related (but different) to $\mathbb{K}(s, r, \sqrt{\mu r/n})$ previously considered in Lemma 15. Our lower bound relies on the key observation that

$$\mathcal{M}(\mathbb{K}_{r,\mu}, d_{2,\infty}, \delta) \geq \mathcal{M}(\mathbb{K}_s(n, r, \sqrt{\mu r/n}), d_{2,\infty}, 2\delta) \geq \mathcal{M}(\mathbb{K}_s(n, r, \sqrt{\mu r/n}), d_{\mathrm{F}}, 2\sqrt{s}\delta), \tag{113}$$

where the first inequality follows from Exercise 4.2.10 in [54], and the second inequality follows from the fact that for any $\boldsymbol{U}_1, \boldsymbol{U}_2 \in \mathbb{K}_s(n, r, \sqrt{\mu r/n})$, we have

$$d_{2,\infty}(\boldsymbol{U}_1, \boldsymbol{U}_2) \geq \frac{1}{\sqrt{s}} d_{\mathrm{F}}(\boldsymbol{U}_1, \boldsymbol{U}_2).$$

Starting from Equation (113), we proceed to obtain a $\sqrt{s}\delta$-packing for $\mathbb{K}_s(n, r, \sqrt{\mu r/n})$ under $d_{\mathrm{F}}$. Since any $\boldsymbol{U} \in \mathbb{K}_s(n, r, \sqrt{\mu r/n})$ can be uniquely identified with an element in $\mathbb{K}(s, r, \sqrt{\mu r/n})$ by restricting it to its first $s$ rows. By the assumption that $n/\mu \geq \max\{4, r\}$ and the lower bound on $s$ in Equation (111), we verify that the conditions of Lemma 15 are all satisfied. Thus, following directly from the lower bound of $\xi_H(\mathbb{K}(s, r, \sqrt{\mu r/n}))$ in Lemma 15 and Lemma 14, we have

$$\mathcal{M}\left(\mathbb{K}_s(n, r, \sqrt{\mu r/n}), d_{\mathrm{F}}, 2\sqrt{s}\delta\right) \geq \frac{1}{2} \mathcal{N}\left(\mathbb{V}_{s,r}, d_{\mathrm{F}}, 4\sqrt{s}\delta\right).$$

Applying the lower bound in Lemma 13 to the right hand side of the above bound, we have

$$\mathcal{M}\left(\mathbb{K}_s(n, r, \sqrt{\mu r/n}), d_{\mathrm{F}}, 2\sqrt{s}\delta\right) \geq \frac{1}{2} \left(\frac{c_0 \sqrt{r}}{4\sqrt{s}\delta}\right)^{r(s-r)}. \tag{114}$$

Under our upper bound assumption on $\delta$ in Equation (21) and $n/\mu \geq \max\{4, r\}$,

$$\frac{c_0^2 r}{384 e^2 \delta^2} \geq \frac{c_0^2 r}{384 e^2} \cdot \frac{96 e^2 n \log(12n/\mu)}{c_0^2 \mu r} = \frac{n}{4\mu} \log(12n/\mu) \geq 1,$$

from our choice of $s$, it follows that

$$s \leq \frac{c_0^2 r}{8e^2 \delta^2} + 1 \leq \frac{c_0^2 r}{8e^2 \delta^2} + \frac{c_0^2 r}{384 e^2 \delta^2} \leq \frac{c_0^2 r}{7e^2 \delta^2},$$

which implies $\sqrt{s}\delta \leq (\sqrt{7}e)^{-1} c_0 \sqrt{r}$. Thus, it follows from Equation (114) that

$$\mathcal{M}\left(\mathbb{K}_s(n, r, \sqrt{\mu r/n}), d_{\mathrm{F}}, 2\sqrt{s}\delta\right) \geq \frac{1}{2} \left(\frac{\sqrt{7}e}{4}\right)^{r(s-r)} \geq \frac{1}{2} \left(\frac{\sqrt{7}e}{4}\right)^{\frac{rs}{2}},$$

where the last inequality follows from Equation (111). Noting that $\sqrt{7}e/4 \geq \sqrt{e}$, we have

$$\mathcal{M}\left(\mathbb{K}_s(n, r, \sqrt{\mu r/n}), d_{\mathrm{F}}, 2\sqrt{s}\delta\right) \geq \frac{1}{2} \exp\left(\frac{rs}{4}\right)$$

Applying Equation (113) followed by Equation (112), we obtain

$$\mathcal{M}(\mathbb{K}_{r,\mu}, d_{2,\infty}, \delta) \geq \mathcal{M}\left(\mathbb{K}_s(n, r, \sqrt{\mu r/n}), d_{\mathrm{F}}, 2\sqrt{s}\delta\right)$$
$$\geq \frac{1}{2} \exp\left(\max\left\{\frac{3nr}{\mu} \log\left(\frac{12n}{\mu}\right), \frac{c_0^2 r^2}{32 e^2 \delta^2}\right\}\right) \geq \frac{1}{2} \exp\left(\frac{c_0^2 r^2}{32 e^2 \delta^2}\right). \tag{115}$$

Taking the logarithm on both sides of the above display yields Equation (22). $\qquad\square$

# G   Upper bounds of metric entropy under $d_{2,\infty}$

To provide a complete picture, Lemma 16 establishes an upper bound on the packing $\delta$-entropy of $\mathbb{K}_{r,\mu}$ under $d_{2,\infty}$ that matches the lower bound in Lemma 4 up to log-factors when $\delta$ satisfies Equation (21).

**Lemma 16.** *Assume that $n \geq \mu r$ for $r \in [n]$ and $\mu > 0$. For all $n$ sufficiently large and any $\sqrt{4/(n-1)} < \delta < \sqrt{\mu r/n}$, we have*

$$\log \mathcal{M}\left(\mathbb{K}_{r,\mu}, d_{2,\infty}, \delta\right) \lesssim \frac{r^2}{\delta^2} \log n. \tag{116}$$

*Proof of Lemma 16.* We obtain an upper bound of $\mathcal{M}(\mathbb{K}_{r,\mu}, d_{2,\infty}, \delta)$ by finding an upper bound of $\mathcal{N}(\mathbb{K}_{r,\mu}, d_{2,\infty}, \delta)$. Since for any $\boldsymbol{U}_1$ and $\boldsymbol{U}_2 \in \mathbb{R}^{n \times r}$,

$$d_{2,\infty}\left(\boldsymbol{U}_1, \boldsymbol{U}_2\right) \leq \|\boldsymbol{U}_1 - \boldsymbol{U}_2\|_{2,\infty},$$

we have

$$\mathcal{N}\left(\mathbb{K}_{r,\mu}, d_{2,\infty}, \delta\right) \leq \mathcal{N}\left(\mathbb{K}_{r,\mu}, \|\cdot\|_{2,\infty}, \delta\right). \tag{117}$$

Thus, it suffices to obtain an upper bound for $\mathcal{N}\left(\mathbb{K}_{r,\mu}, \|\cdot\|_{2,\infty}, \delta\right)$ instead. Let

$$\mathbb{T} := \left\{\boldsymbol{u} \geq 0 : \boldsymbol{u} \in \sqrt{r}\mathbb{S}^{n-1}, \|\boldsymbol{u}\|_\infty \leq 1\right\}. \tag{118}$$

For any $\boldsymbol{U} \in \mathbb{K}_{r,\mu}$, noting that $\sqrt{\mu r/n} \leq 1$, we define a mapping $h : \mathbb{K}_{r,\mu} \to \mathbb{T}$ given by

$$h(\boldsymbol{U}) := \left(\|\boldsymbol{U}_{1,.}\|_2, \|\boldsymbol{U}_{2,.}\|_2, \cdots, \|\boldsymbol{U}_{n,.}\|_2\right)^\top.$$

Indeed, $h(\boldsymbol{U}) \in \mathbb{T}$ since by definition, we have $h(\boldsymbol{U}) \geq 0$, $\|h(\boldsymbol{U})\|_2 = \sqrt{r}$ and $\|h(\boldsymbol{U})\|_\infty \leq 1$.

We pause to give a roadmap of our proof. Recall that for a set $\mathbb{K}$, its exterior $\delta$-covering is a $\delta$-covering, except that the exterior $\delta$-covering allows elements not in $\mathbb{K}$ to form the covering (see Exercise 4.2.9 in [54]). To obtain $\mathcal{N}\left(\mathbb{K}_{r,\mu}, \|\cdot\|_{2,\infty}, \delta\right)$, we will explicitly construct an exterior $\delta$-covering of $\mathbb{K}_{r,\mu}$ under $\|\cdot\|_{2,\infty}$. To do so, we proceed in three steps. In the first step, we construct an exterior $\delta$-covering $\mathscr{C}$ of $\mathbb{T}$ under $\|\cdot\|_\infty$. In the second step, we divide the set $\mathbb{K}_{r,\mu}$ into separate subsets $\{\mathbb{U}_{\boldsymbol{v}}\}_{\boldsymbol{v} \in \mathbb{T}}$ via the mapping $h : \mathbb{K}_{r,\mu} \to \mathbb{T}$. In this way, each element $\boldsymbol{v}$ in $\mathbb{T}$ is associated with a subset $\mathbb{U}_{\boldsymbol{v}} \subseteq \mathbb{K}_{r,\mu}$. For every $\boldsymbol{v} \in \mathscr{C}$ (the exterior $(\delta/2)$-covering set of $\mathbb{T}$), we construct an exterior $(\delta/2)$-covering $\mathscr{C}_{\boldsymbol{v}}$ of $\mathbb{U}_{\boldsymbol{v}}$ under $\|\cdot\|_{2,\infty}$. In the last step, we take the union of $\mathscr{C}_{\boldsymbol{v}}$ over $\boldsymbol{v} \in \mathscr{C}$ to form a set $\mathscr{C}_{\mathbb{K}}$, and show this set is an exterior $\delta$-covering set for $\mathbb{K}_{r,\mu}$ under $\|\cdot\|_{2,\infty}$. Finally, we control the cardinality of $\mathscr{C}_{\mathbb{K}}$ to obtain an upper bound of $\mathcal{N}\left(\mathbb{K}_{r,\mu}, \|\cdot\|_{2,\infty}, \delta\right)$, which in turn will yield our desired upper bound on $\mathcal{M}(\mathbb{K}_{r,\mu}, d_{2,\infty}, \delta)$.

**Step 1. Construct an exterior $(\delta/2)$-covering set of $\mathbb{T}$ under $\|\cdot\|_\infty$.**

Recall that $\mathbb{T}$ is given in Equation (118). Let $s = \lceil 4/\delta^2 \rceil$. Under the assumption that $\delta > \sqrt{4/(n-1)}$, we have $s \leq n$. We consider the nonnegative integer solutions to the indeterminate equation

$$z_1 + z_2 + \cdots + z_n = rs, \quad \|\boldsymbol{z}\|_\infty \leq s + 1 \tag{119}$$

and let

$$\mathscr{C} := \left\{\sqrt{\frac{1}{s}}\left(\sqrt{z_1}, \cdots, \sqrt{z_n}\right) : \boldsymbol{z} \text{ is a solution of Equation (119)}\right\}. \tag{120}$$

By Lemma 17, we have $\mathscr{C}$ forms an exterior $\delta/2$-covering set of $\mathbb{T}$ under $\|\cdot\|_\infty$. Since without the constraint $\|\boldsymbol{z}\|_\infty \leq s + 1$, Equation (119) has $\binom{n+rs-1}{rs}$ solutions, we have

$$|\mathscr{C}| \leq \binom{n + rs - 1}{rs} \leq \left(\frac{e(n+rs)}{rs}\right)^{rs} = \left(\frac{e(n + r\lceil 4/\delta^2 \rceil)}{r\lceil 4/\delta^2 \rceil}\right)^{r\lceil 4/\delta^2 \rceil}. \tag{121}$$

**Step 2. Construct an exterior $(\delta/2)$-covering set of $\mathbb{U}_{\boldsymbol{v}}$ under $\|\cdot\|_{2,\infty}$.**

For a given $\boldsymbol{v} \in \mathscr{C}$, consider the subset

$$\mathbb{U}_{\boldsymbol{v}} := \{\boldsymbol{U} \in \mathbb{K}_{r,\mu} : h(\boldsymbol{U}) = \boldsymbol{v}\}, \tag{122}$$

our next step is to construct an exterior $(\delta/2)$-covering set under $\|\cdot\|_{2,\infty}$ for $\mathbb{U}_{\boldsymbol{v}}$. For every $i \in [n]$, consider a $(\delta/2)$-covering set $\mathscr{C}_{v_i}$ of $v_i \mathbb{S}^{r-1}$ under the $\ell_2$ norm. By Corollary 4.2.13 in [54],

$$|\mathscr{C}_{v_i}| \leq \left( \frac{6 v_i}{\delta} \right)^r \quad \text{for all } i \in [n]. \tag{123}$$

Consider the set

$$\mathscr{C}_{\boldsymbol{v}} := \left\{ \boldsymbol{\Theta} \in \mathbb{R}^{n \times r} : \boldsymbol{\Theta} = (\boldsymbol{\theta}_1, \boldsymbol{\theta}_2, \dots, \boldsymbol{\theta}_n)^\top, \boldsymbol{\theta}_i \in \mathscr{C}_{v_i} \text{ for } i \in [n] \right\}. \tag{124}$$

We claim that $\mathscr{C}_{\boldsymbol{v}}$ is an exterior $(\delta/2)$-covering set for $\mathbb{U}_{\boldsymbol{v}}$ under $\|\cdot\|_{2,\infty}$. To see this, note that for any $\boldsymbol{U} \in \mathbb{U}_{\boldsymbol{v}}$ and any $i \in [n]$, since $\boldsymbol{U}_{i,\cdot}^\top \in v_i \mathbb{S}^{r-1}$ and $\mathscr{C}_{v_i}$ is a $(\delta/2)$-covering of $v_i \mathbb{S}^{r-1}$ under the $\ell_2$ norm, there exists a $\boldsymbol{\theta}_i \in \mathscr{C}_{v_i}$ such that

$$\left\| \boldsymbol{\theta}_i - \boldsymbol{U}_{i,\cdot}^\top \right\|_2 \leq \frac{\delta}{2}, \quad \text{for all } i \in [n].$$

Let $\boldsymbol{\Theta} = (\boldsymbol{\theta}_1, \boldsymbol{\theta}_2, \dots, \boldsymbol{\theta}_n)^\top \in \mathscr{C}_{\boldsymbol{v}}$. It follows that

$$\|\boldsymbol{\Theta} - \boldsymbol{U}\|_{2,\infty} = \max_{i \in [n]} \leq \left\| \boldsymbol{\theta}_i - \boldsymbol{U}_{i,\cdot}^\top \right\|_2 \leq \frac{\delta}{2}.$$

Since $\boldsymbol{v} \in \mathscr{C}$, from Equation (119), we have

$$\|\boldsymbol{v}\|_0 \leq rs \tag{125}$$

as any solution $\boldsymbol{z}$ to Equation (119) has at most $rs$ nonzero entries. By Equations (119) and (120), we also have

$$\|\boldsymbol{v}\|_\infty \leq \frac{\sqrt{s+1}}{\sqrt{s}} \leq 2. \tag{126}$$

By Equations (123) and (124),

$$|\mathscr{C}_{\boldsymbol{v}}| \leq \prod_{i=1}^n \left( \frac{6 v_i}{\delta} \right)^r = \prod_{i:v_i > 0} \left( \frac{6 v_i}{\delta} \right)^r \leq \left( \frac{12}{\delta} \right)^{sr^2}.$$

where the last inequality follows from Equations (125) and (126). By the choice of $s = \lceil 4/\delta^2 \rceil$, we have

$$|\mathscr{C}_{\boldsymbol{v}}| \leq \left( \frac{12}{\delta} \right)^{r^2 \lceil 4/\delta^2 \rceil}. \tag{127}$$

**Step 3. Construct an exterior $\delta$-covering set for $\mathbb{K}_{r,\mu}$ under $\|\cdot\|_{2,\infty}$.**

Consider the set

$$\mathscr{C}_\mathbb{K} := \bigcup_{\boldsymbol{v} \in \mathscr{C}} \mathscr{C}_{\boldsymbol{v}}, \tag{128}$$

we claim that this set forms an exterior $\delta$-covering set for $\mathbb{K}_{r,\mu}$ under $\|\cdot\|_{2,\infty}$.

For any $\boldsymbol{U} \in \mathbb{K}_{r,\mu}$, since $\mathscr{C}$ is an exterior $(\delta/2)$-covering of $\mathbb{T}$ under $\|\cdot\|_\infty$, we can find $\boldsymbol{v} \in \mathscr{C}$ such that $\|h(\boldsymbol{U}) - \boldsymbol{v}\|_\infty \leq \delta/2$. Consider a matrix $\widetilde{U} \in \mathbb{R}^{n \times r}$ with rows given by

$$\widetilde{\boldsymbol{U}}_{i,\cdot} = \begin{cases} v_i \boldsymbol{U}_{i,\cdot} / h_i(\boldsymbol{U}) & \text{if } h_i(\boldsymbol{U}) > 0, \\ \boldsymbol{\theta}_i^\top & \text{if } h_i(\boldsymbol{U}) = 0, \end{cases}$$

for all $i \in [n]$, where $\boldsymbol{\theta}_i$ is an arbitrary element of $\mathscr{C}_{v_i}$. Recall the definition of $\mathbb{U}_{\boldsymbol{v}}$ from Equation (122). By construction, $\widetilde{U} \in \mathbb{U}_{\boldsymbol{v}}$, which implies that there exists $\boldsymbol{\Theta} \in \mathscr{C}_{\boldsymbol{v}} \subseteq \mathscr{C}_\mathbb{K}$ such that

$$\left\| \boldsymbol{\Theta} - \widetilde{U} \right\|_{2,\infty} \leq \frac{\delta}{2},$$

where we use the fact that $\mathscr{C}_{\boldsymbol{v}}$ defined in Equation (124) is an exterior $(\delta/2)$-covering of $\mathbb{U}_{\boldsymbol{v}}$. For $i \in [n]$ such that $h_i(\boldsymbol{U}) > 0$, we have

$$\left\| \widetilde{\boldsymbol{U}}_{i,\cdot} - \boldsymbol{U}_{i,\cdot} \right\|_2 = \left| \frac{v_i}{h_i(\boldsymbol{U})} - 1 \right| \|\boldsymbol{U}_{i,\cdot}\|_2 = |v_i - h_i(\boldsymbol{U})| \leq \frac{\delta}{2}$$

and for $i \in [n]$ such that $h_i(\boldsymbol{U}) = 0$, we have

$$\left\|\widetilde{\boldsymbol{U}}_{i,\cdot} - \boldsymbol{U}_{i,\cdot}\right\|_2 = \|\boldsymbol{\theta}_i\|_2 = v_i = |v_i - h_i(\boldsymbol{U})| \leq \frac{\delta}{2}.$$

Combining the above two displays, it follows that

$$\left\|\widetilde{\boldsymbol{U}} - \boldsymbol{U}\right\|_{2,\infty} = \max_{i \in [n]} \left\|\widetilde{\boldsymbol{U}}_{i,\cdot} - \boldsymbol{U}_{i,\cdot}\right\|_2 \leq \frac{\delta}{2}.$$

Thus, we have found $\boldsymbol{\Theta} \in \mathscr{C}_{\mathbb{K}}$ such that

$$\|\boldsymbol{\Theta} - \boldsymbol{U}\|_{2,\infty} \leq \|\boldsymbol{\Theta} - \widetilde{\boldsymbol{U}}\|_{2,\infty} + \|\widetilde{\boldsymbol{U}} - \boldsymbol{U}\|_{2,\infty} \leq \delta,$$

and it follows that $\mathscr{C}_{\mathbb{K}}$ is an exterior $\delta$-covering set for $\mathbb{K}_{r,\mu}$ under $\|\cdot\|_{2,\infty}$.

**Step 4. An upper bound on $\mathcal{M}(\mathbb{K}_{r,\mu}, \|\cdot\|_{2,\infty}, \delta)$.**

Recalling the definition of $\mathscr{C}_{\mathbb{K}}$ from Equation (128), Equations (121) and (127) imply that

$$|\mathscr{C}_{\mathbb{K}}| \leq \sum_{\boldsymbol{v} \in \mathscr{C}} |\mathscr{C}_{\boldsymbol{v}}| \leq \left(\frac{e\left(n + r\left\lceil 4/\delta^2\right\rceil\right)}{r\left\lceil 4/\delta^2\right\rceil}\right)^{r\left\lceil 4/\delta^2\right\rceil} \cdot \left(\frac{12}{\delta}\right)^{r^2\left\lceil 4/\delta^2\right\rceil}.$$

Using this bound and the fact that $\mathscr{C}_{\mathbb{K}}$ is an exterior $\delta$-covering set for $\mathbb{K}_{r,\mu}$ under $\|\cdot\|_{2,\infty}$, Exercise 4.2.9 in [54] implies that

$$\mathcal{N}(\mathbb{K}_{r,\mu}, \|\cdot\|_{2,\infty}, 2\delta) \leq |\mathscr{C}_{\mathbb{K}}| \leq \left(\frac{e\left(n + r\left\lceil 4/\delta^2\right\rceil\right)}{r\left\lceil 4/\delta^2\right\rceil}\right)^{r\left\lceil 4/\delta^2\right\rceil} \left(\frac{12}{\delta}\right)^{r^2\left\lceil 4/\delta^2\right\rceil}$$

$$\leq \left(\frac{en\delta^2}{4r} + e\right)^{\frac{4r}{\delta^2} + r} \left(\frac{12}{\delta}\right)^{\frac{4r^2}{\delta^2} + r^2}.$$

Combining this with with Equation (117) and Lemma 4.2.8 in [54], we conclude that

$$\mathcal{M}\left(\mathbb{K}_{r,\mu}, d_{2,\infty}, \delta\right) \leq \mathcal{N}\left(\mathbb{K}_{r,\mu}, d_{2,\infty}, \delta/2\right) \leq \left(\frac{en\delta^2}{64r} + e\right)^{\frac{64r}{\delta^2} + r} \left(\frac{48}{\delta}\right)^{\frac{64r^2}{\delta^2} + r^2}. \tag{129}$$

Taking the logarithm on both sides of Equation (129), we have

$$\log \mathcal{M}\left(\mathbb{K}_{r,\mu}, d_{2,\infty}, \delta\right) \leq r\left(\frac{64 + \delta^2}{\delta^2}\right) \log\left(\frac{en\delta^2}{64r} + e\right) + r^2\left(\frac{64 + \delta^2}{\delta^2}\right) \log\left(\frac{48}{\delta}\right)$$

$$\leq \frac{65r}{\delta^2} \log\left(\frac{e\mu}{64} + e\right) + \frac{65r^2}{\delta^2} \log\left(\frac{48}{\delta}\right)$$

where the last inequality follows from $\delta < \sqrt{\mu r/n} \leq 1$. Under the assumption that $\delta > \sqrt{4/n}$ and $\mu r \leq n$, the right hand side of the above display can be further written as

$$\log \mathcal{M}\left(\mathbb{K}_{r,\mu}, d_{2,\infty}, \delta\right) \leq \frac{65r}{\delta^2} \log\left(\frac{en}{64} + e\right) + \frac{65r^2}{\delta^2} \log\left(24\sqrt{n}\right),$$

yields Equation (116) for all $n$ sufficiently large, completing the proof. $\qquad\square$

Recall the definition of $\mathbb{T}$ in Equation (118) and $\mathscr{C}$ defined in Equation (120), we have Lemma 17.

**Lemma 17.** *Let the set $\mathbb{T} \subseteq \sqrt{r}\mathbb{S}^{n-1}$ be as defined in Equation (118), and let $\boldsymbol{h}$ be an arbitrary element in $\mathbb{T}$. Then there exists $\boldsymbol{v} \in \mathscr{C}$ such that*

$$\|\boldsymbol{h} - \boldsymbol{v}\|_\infty \leq \frac{1}{\sqrt{s}} \leq \frac{\delta}{2}, \tag{130}$$

*where $\delta$ is given in Lemma 16 and $s = \left\lceil 4/\delta^2\right\rceil$.*

*Proof.* The inequality $1/\sqrt{s} \leq \delta/2$ in Equation (130) follows directly from $s = \lceil 4/\delta^2 \rceil$. To construct a $\boldsymbol{v} \in \mathscr{C}$ satisfying Equation (130), we first construct a $\boldsymbol{\zeta} \in \mathbb{R}^n$ and then make a slight modification to $\boldsymbol{\zeta}$ to obtain $\boldsymbol{v}$.

To construct $\boldsymbol{\zeta}$, we set $\zeta_1^2$ to be either $\lfloor h_1^2 s \rfloor /s$ or $\lceil h_1^2 s \rceil /s$, and follow the recursive procedure to obtain an entrywise positive $\boldsymbol{\zeta} \geq 0$:

$$\zeta_i^2 = \begin{cases} \lfloor h_i^2 s \rfloor /s & \text{if } \sum_{j=1}^{i-1} h_j^2 - \sum_{j=1}^{i-1} \zeta_j^2 < 0, \\ \lceil h_i^2 s \rceil /s & \text{if } \sum_{j=1}^{i-1} h_j^2 - \sum_{j=1}^{i-1} \zeta_j^2 \geq 0 \end{cases} \tag{131}$$

for $2 \leq i \leq n$. The construction given in Equation (131) guarantees that

$$\|\boldsymbol{h} - \boldsymbol{\zeta}\|_\infty \leq \max_{i \in [n]} \left\{ h_i - \sqrt{\lfloor h_i^2 s \rfloor /s}, \sqrt{\lceil h_i^2 s \rceil /s} - h_i \right\} \leq \frac{1}{\sqrt{s}}, \tag{132}$$

where the last inequality follows from the fact that $\sqrt{a} - \sqrt{b} \leq \sqrt{a-b}$ for any $a \geq b > 0$. Now we show that following the procedure in Equation (131), we have

$$\left| \sum_{i=1}^k \zeta_i^2 - \sum_{i=1}^k h_i^2 \right| \leq \frac{1}{s}, \tag{133}$$

for all $k \in [n]$. We prove the above bound by induction on $k \in [n]$. When $k = 1$, we trivially have

$$\left| \zeta_1^2 - h_1^2 \right| \leq \frac{1}{s}.$$

For the inductive step, suppose that for $k > 1$, we have

$$\left| \sum_{i=1}^{k-1} h_i^2 - \sum_{i=1}^{k-1} \zeta_i^2 \right| \leq \frac{1}{s}.$$

By definition in Equation (131), if

$$0 \leq \sum_{i=1}^{k-1} h_i^2 - \sum_{i=1}^{k-1} \zeta_i^2 \leq \frac{1}{s}, \tag{134}$$

then

$$-\frac{1}{s} \leq \sum_{i=1}^{k-1} h_i^2 - \sum_{i=1}^{k-1} \zeta_i^2 + h_k^2 - \frac{\lceil h_k^2 s \rceil}{s} \leq \frac{1}{s}.$$

Thus, $\zeta_k^2 = \lceil h_k^2 s \rceil /s$ satisfies

$$\left| \sum_{i=1}^k h_i^2 - \sum_{i=1}^k \zeta_i^2 \right| \leq \frac{1}{s}.$$

If, contrary to Equation (134), we have

$$-\frac{1}{s} \leq \sum_{i=1}^{k-1} h_i^2 - \sum_{i=1}^{k-1} \zeta_i^2 < 0,$$

then

$$-\frac{1}{s} \leq \sum_{i=1}^{k-1} h_i^2 - \sum_{i=1}^{k-1} \zeta_i^2 + h_k^2 - \frac{\lfloor h_k^2 s \rfloor}{s} \leq \frac{1}{s},$$

and thus $\zeta_k^2 = \lfloor h_k^2 s \rfloor /s$ again satisfies

$$\left| \sum_{i=1}^k h_i^2 - \sum_{i=1}^k \zeta_i^2 \right| \leq \frac{1}{s}.$$

Therefore, Equation (133) holds for all $k \in [n]$. Taking $k = n$ in Equation (133) and by the fact that $\|\boldsymbol{h}\|_2 = r$, we obtain that

$$\left| s\|\boldsymbol{\zeta}\|_2^2 - sr \right| \leq 1. \tag{135}$$

Following Equation (131), we also have that $s\zeta_i^2$ is an integer for every $i \in [n]$. Combined with Equation (135), we have a stronger statement that

$$(s\|\boldsymbol{\zeta}\|_2^2 - sr) \in \{-1, 0, 1\}.$$

If $(s\|\boldsymbol{\zeta}\|_2^2 - sr) = 0$, setting $\boldsymbol{v} = \boldsymbol{\zeta}$ already guarantees that $\|\boldsymbol{v}\|_2^2 = r$ and $\|\boldsymbol{v} - \boldsymbol{h}\|_\infty \le 1/\sqrt{s}$. Otherwise, if $(s\|\boldsymbol{\zeta}\|_2^2 - sr) = -1$, then by Equation (131), there must be an $i_0 \in [n]$ such that

$$s\zeta_{i_0}^2 = \lfloor h_{i_0}^2 s \rfloor < h_{i_0}^2 s < \lceil h_{i_0}^2 s \rceil = \lfloor h_{i_0}^2 s \rfloor + 1.$$

Setting

$$v_i = \begin{cases} \sqrt{\zeta_{i_0}^2 + 1/s} = \sqrt{\lceil h_{i_0}^2 s \rceil / s}, & \text{if } i = i_0 \\ \zeta_i, & \text{otherwise} \end{cases}$$

guarantees that $\|\boldsymbol{v}\|_2^2 = r$ and

$$\|\boldsymbol{v} - \boldsymbol{h}\|_\infty = \max\left\{ \max_{i \ne i_0} |\zeta_i - h_i|, \left| \sqrt{\lceil h_{i_0}^2 s \rceil / s} - h_{i_0} \right| \right\} \le \frac{1}{\sqrt{s}},$$

where the last inequality follows from Equation (132). Similarly, if $(s\|\boldsymbol{\zeta}\|_2^2 - sr) = 1$, then we can alter one element of $\boldsymbol{\zeta}$ to obtain $\boldsymbol{v}$, such that $\|\boldsymbol{v}\|_2^2 = r$ and $\|\boldsymbol{v} - \boldsymbol{h}\|_\infty \le 1/\sqrt{s}$.

Since $sv_i^2$ are integers for all $i \in [n]$, $s\|\boldsymbol{v}\|_2^2 = sr$, and the fact that

$$s\|\boldsymbol{v}\|_\infty^2 \le \max_{i \in [n]} \lceil h_i^2 s \rceil \le s + 1,$$

where the last inequality follows from $\|\boldsymbol{h}\|_\infty \le 1$ for all $\boldsymbol{h} \in \mathbb{T}$, we see that $\boldsymbol{v} \in \mathscr{C}$. $\qquad\square$

## H   Proof of Theorem 2

In this section, we use the Yang-Barron method [59] along with a Fano lower bound (see Proposition 15.12 and Lemma 15.21 in [56]) to prove the minimax lower bound for eigenspace estimation stated in Theorem 2. We follow the notation used in [56]. Given a class of distributions $\mathcal{P}$, we let $\theta$ denote a functional on the space $\mathcal{P}$, that is, a mapping from a distribution $\mathbb{P}$ to a parameter $\theta(\mathbb{P})$ taking values in some space $\Omega$. We let $\rho : \Omega \times \Omega \to [0, \infty)$ be a semi-metric. Proposition 2 states the Fano lower bound (Proposition 15.12 in [56]).

**Proposition 2** (Fano lower bound). *Let $\{\theta_1, \theta_2, \ldots, \theta_M\} \subseteq \Omega$ be a $2\delta$-separated set under the $\rho$ semi-metric, and suppose that $J$ is uniformly distributed over the index set $[M]$, and $(Z \mid J = j) \sim \mathbb{P}_{\theta_j}$. Then for any increasing function $\Phi : [0, \infty) \to [0, \infty)$, the minimax risk is lower bounded as*

$$\inf_{\widehat{\theta}} \sup_{\mathbb{P} \in \mathcal{P}} \mathbb{E}_\mathbb{P} \, \Phi\left( \rho\left( \widehat{\theta}, \theta(\mathbb{P}) \right) \right) \ge \Phi(\delta) \left( 1 - \frac{I(Z; J) + \log 2}{\log M} \right),$$

*where the infimum is over all estimators $\widehat{\theta}$ and $I(Z; J)$ is the mutual information between $Z$ and $J$.*

The Yang-Barron method gives an upper bound for the mutual information $I(Z; J)$.

**Lemma 18** (Yang-Barron method [59]). *Let $\mathcal{N}_{\mathrm{KL}}(\varepsilon; \mathcal{P})$ denote the $\varepsilon$-covering number of $\mathcal{P}$ in the square-root KL-divergence. Then the mutual information is upper bounded as*

$$I(Z; J) \le \inf_{\varepsilon > 0} \log \mathcal{N}_{\mathrm{KL}}(\varepsilon; \mathcal{P}) + \varepsilon^2.$$

*Proof of Theorem 2.* To each $(\boldsymbol{\Lambda}^\star, \boldsymbol{U}^\star) \in \Omega(\lambda^\star, \mu, r)$, we associate a probability distribution $\mathbb{P}_{\boldsymbol{\Lambda}^\star, \boldsymbol{U}^\star}$ on $\mathbb{R}^{n \times n}$ with density given by

$$f_{\boldsymbol{\Lambda}^\star, \boldsymbol{U}^\star}(\boldsymbol{W}) = \prod_{1 \le i \le j \le n} g\left( \frac{(\boldsymbol{W} - \boldsymbol{U}^\star \boldsymbol{\Lambda}^\star \boldsymbol{U}^{\star\top})_{ij}}{\sigma} \right),$$

where $g(\cdot)$ denotes the density of a standard normal. We define the class of distributions

$$\mathcal{P} = \{ \mathbb{P}_{\boldsymbol{\Lambda}^\star, \boldsymbol{U}^\star} : (\boldsymbol{\Lambda}^\star, \boldsymbol{U}^\star) \in \Omega(\lambda^\star, \mu, r) \}.$$

For $\boldsymbol{\mu}_1, \boldsymbol{\mu}_2 \in \mathbb{R}^n$ and a pair of normal distributions $N(\boldsymbol{\mu}_1, \sigma^2 \boldsymbol{I}_n)$ and $N(\boldsymbol{\mu}_2, \sigma^2 \boldsymbol{I}_n)$, their KL-divergence is given by (see Example 15.13 in [56])

$$\mathrm{KL}\Big(N(\boldsymbol{\mu}_1, \sigma^2 \boldsymbol{I}_n) \big\| N(\boldsymbol{\mu}_2, \sigma^2 \boldsymbol{I}_n)\Big) = \frac{1}{2\sigma^2} \|\boldsymbol{\mu}_2 - \boldsymbol{\mu}_1\|_2^2.$$

It follows that for any $\mathbb{P}_{\boldsymbol{\Lambda}^\star, \boldsymbol{U}_1^\star} \mathbb{P}_{\boldsymbol{\Lambda}^\star, \boldsymbol{U}_2^\star} \in \mathcal{P}$,

$$\mathrm{KL}\left(\mathbb{P}_{\boldsymbol{\Lambda}^\star, \boldsymbol{U}_1^\star} \big\| \mathbb{P}_{\boldsymbol{\Lambda}^\star, \boldsymbol{U}_2^\star}\right) \leq \frac{\lambda^{\star 2}}{2\sigma^2} \left\| \boldsymbol{U}_1^\star \boldsymbol{U}_1^{\star\top} - \boldsymbol{U}_2^\star \boldsymbol{U}_2^{\star\top} \right\|_F^2$$

and thus, taking square roots,

$$\sqrt{\mathrm{KL}(\mathbb{P}_{\boldsymbol{\Lambda}^\star, \boldsymbol{U}_1^\star} \| \mathbb{P}_{\boldsymbol{\Lambda}^\star, \boldsymbol{U}_1^\star})} \leq \frac{\sqrt{2}\lambda^\star}{2\sigma} \left\| \boldsymbol{U}_1^\star \boldsymbol{U}_1^{\star\top} - \boldsymbol{U}_2^\star \boldsymbol{U}_2^{\star\top} \right\|_F \leq \frac{\lambda^\star}{\sigma} d_{\mathrm{F}}\left(\boldsymbol{U}_1^\star, \boldsymbol{U}_2^\star\right), \qquad (136)$$

where the second inequality holds from Lemma 2.6 in [27].

To apply the Yang-Barron method, we follow a two-step procedure (see Chapter 15 of [56]):

1. Pick the smallest $\varepsilon$ such that $\varepsilon^2 \geq \log \mathcal{N}_{\mathrm{KL}}(\mathcal{P}, \varepsilon)$.

2. Choose the largest $\delta$ that we can find a $\delta$-packing that satisfies the lower bound
$$\log \mathcal{M}(\mathbb{K}_{r,\mu}, d_{2,\infty}, \delta) \geq 4\varepsilon^2 + 2\log 2.$$

Then it follows from Lemma 18 and Proposition 2 that the minimax risk is lower bounded by $\delta/2$.

For the first step, noting that by Equation (136), a $(\sigma\varepsilon/\lambda^\star)$-covering set for $\mathbb{V}_{n,r}$ under $d_{\mathrm{F}}$ yields an $\varepsilon$-covering set for the $\sqrt{\mathrm{KL}}$- divergence, we have

$$\mathcal{N}_{\mathrm{KL}}(\mathcal{P}, \varepsilon) \leq \mathcal{N}(\mathbb{V}_{n,r}, d_{\mathrm{F}}, \varepsilon/\lambda^\star) \leq \left(\frac{C_0 \lambda^\star \sqrt{2r}}{\sigma\varepsilon}\right)^{r(n-r)},$$

where the last inequality follows from Lemma 13. Thus, in order to have $\varepsilon^2 \geq \log \mathcal{N}_{\mathrm{KL}}(\mathcal{P}, \varepsilon)$, it suffices to have

$$\varepsilon^2 \geq r(n-r) \log \left(\frac{C_0 \lambda^\star \sqrt{2r}}{\sigma\varepsilon}\right)$$

which holds if we set $\varepsilon = C_0 \lambda^\star \sqrt{2r}/\sigma$. With this choice of $\varepsilon$, we then follow the second step of the Yang-Barron method and pick a $\delta$ to satisfy

$$\log \mathcal{M}\left(\mathbb{K}_{\mu,r}, d_{2,\infty}, \delta\right) \geq \frac{8C_0 r \lambda^{\star 2}}{\sigma^2} + 2\log 2. \qquad (137)$$

By Equation (115) in the proof of Lemma 4, we have

$$\log \mathcal{M}\left(\mathbb{K}_{r,\mu}, d_{2,\infty}, \delta\right) \geq \frac{c_0^2 r^2}{32 e^2 \delta^2} - \log 2 \qquad (138)$$

for $\delta$ satisfying Eqation (21). Combining Equations (137) and (138), our goal is to find a $\delta$, such that

$$\frac{c_0^2 r^2}{32 e^2 \delta^2} \geq \frac{8C_0 r \lambda^{\star 2}}{\sigma^2} + 3\log 2. \qquad (139)$$

We pick $\delta$ to be

$$\delta = \frac{c_0 \sigma \sqrt{r}}{12 e \lambda^\star \sqrt{2C_0}} \wedge \frac{c_0}{4e} \sqrt{\frac{\mu r}{6n \log(12n/\mu)}}. \qquad (140)$$

One can verify that $\delta$ satisfies Equation (21) under the assumption $\lambda^\star \leq (6\sqrt{C_0})^{-1} \sigma \sqrt{n}$. To see that the $\delta$ given by Equation (140) satisfies Equation (139), we separate our discussion into two cases, one for large $\lambda^\star$, and the other for small $\lambda^\star$. When

$$\lambda^\star \geq \sigma \sqrt{\frac{n \log(12n/\mu)}{3C_0 \mu}} \qquad (141)$$

by Equation (140) we have

$$\delta = \frac{c_0 \sigma \sqrt{r}}{12 e \lambda^\star \sqrt{2 C_0}} \tag{142}$$

and therefore, Equation (139) follows from

$$\frac{c_0^2 r^2}{32 e^2 \delta^2} = \frac{9 C_0 r \lambda^{\star 2}}{\sigma^2} \geq \frac{8 C_0 r \lambda^{\star 2}}{\sigma^2} + 3 \log 2,$$

where the first equality holds from Equation (142) and the last inequality follows from Equation (141) for all $n$ sufficiently large.

On the other hand, when

$$\lambda^\star < \sigma \sqrt{\frac{n \log(12 n / \mu)}{3 C_0 \mu}}, \tag{143}$$

by Equation (140) we have

$$\delta = \frac{c_0}{4e} \sqrt{\frac{\mu r}{6 n \log(12 n / \mu)}}, \tag{144}$$

and therefore, Equation (139) follows from

$$\frac{c_0^2 r^2}{32 e^2 \delta^2} = \frac{3 r n \log(12 n / \mu)}{\mu} \geq \frac{8 r n \log(12 n / \mu)}{\mu} + 3 \log 2 \geq \frac{8 C_0 r \lambda^{\star 2}}{\sigma^2} + 3 \log 2,$$

where the first equality follows from Equation (144), the next inequality holds for all $n$ sufficiently large, and the last inequality holds from Equation (143). Thus, we conclude that the choice of $\delta$ given in Equation (140) satisfies Equation (139), completing the second step of the Yang-Barron method.

Finally, combining Proposition 2 and Lemma 18, we conclude that for a universal constant $c > 0$,

$$\inf_{\widehat{\boldsymbol{U}}} \sup_{(\boldsymbol{\Lambda}^\star, \boldsymbol{U}^\star) \in \Omega(\lambda^\star, \mu, r)} \mathbb{E}_{\boldsymbol{\Lambda}^\star, \boldsymbol{U}^\star} \, d_{2,\infty}\left(\widehat{\boldsymbol{U}}, \boldsymbol{U}^\star\right) \geq \frac{\delta}{2} \geq c \left( \frac{\sigma \sqrt{r}}{\lambda^\star} \wedge \sqrt{\frac{\mu r}{n \log(n / \mu)}} \right),$$

completing the proof. $\qquad \square$

# I  Additional experiments

Here we collect additional experiments to complement our simulations in Section 5.

## I.1  Additional numerical results on rank-one eigenvector estimation

We provide additional details for the experiments previously discussed in Section 5.1. Recall that we observe

$$\boldsymbol{Y} = \boldsymbol{M}^\star + \boldsymbol{W} = \lambda^\star \boldsymbol{u}^\star \boldsymbol{u}^{\star\top} + \boldsymbol{W},$$

and wish to recover $\boldsymbol{u}^\star \in \mathbb{S}^{n-1}$. We refer the reader to Section 5.1 for the details of how $\lambda^\star$, $\boldsymbol{u}^\star$ and $\boldsymbol{W}$ are generated and how we run Algorithm 1.

In Section 5.1 we mentioned that under Laplacian noise, as shown in the third column of Figure 1, the estimator $\widehat{\boldsymbol{u}}$ given by Algorithm 1 seems to have a slight dependence on the coherence parameter $\mu$. We give a close examination of the estimation error of the largest entry of $\boldsymbol{u}^\star$ in Figure 3, with shaded bands indicating $95\%$ bootstrap confidence intervals. Figure 3 shows that the estimation error of the largest entry of $\boldsymbol{u}^\star$ for $\widehat{\boldsymbol{u}}$ is seen to be much smaller than $10^{-2}$, while in the third column of Figures 1, the estimation error $d_\infty(\widehat{\boldsymbol{u}}, \boldsymbol{u}^\star)$ is well above $10^{-2}$. This indicates that the dependence on $\mu$ observed in the right-hand subplot of Figure 1 comes not from estimating the largest entry of $\boldsymbol{u}^\star$, but rather from estimating the other entries. In our experiment, the other entries are all nearly incoherent (that is, they have a magnitude at most $O(\sqrt{\log n / n})$), whence their estimation errors are expected to have no dependence on $\mu$ asymptotically. Thus, the slight dependence on $\mu$ exhibited Figure 1 is most likely to be a small order dependence compared to the error rate stated in Theorem 1, and should not affect the asymptotic error rate.

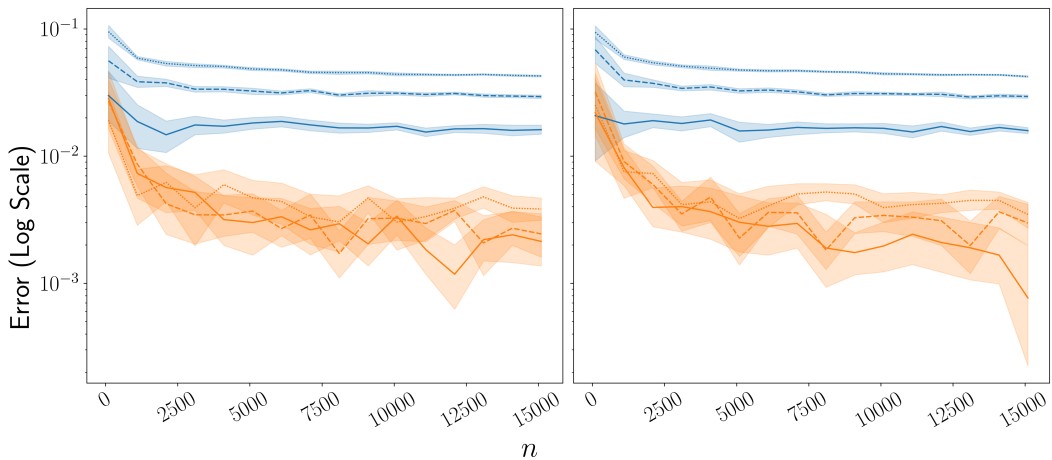

Figure 3: Numerical error in recovering the largest entry of $\boldsymbol{u}^\star$ as a function of matrix dimension $n$, by the leading eigenvector (blue line) or the estimator given in Algorithm 1 (orange line) for three different choices of $\|\boldsymbol{u}^\star\|_\infty$: 0.8 (dotted lines), 0.55 (dashed lines) and 0.3 (solid lines). The plot on the left corresponds to $\boldsymbol{u}^\star$ generated via the Bernoulli scheme, while the plot on the right corresponds to the Haar scheme.

## I.2 Additional numerical results on rank-$r$ eigenvector estimation

We provide additional details for the experiments previously discussed in Section 5.2. Recall that we observe

$$\boldsymbol{Y} = \boldsymbol{M}^\star + \boldsymbol{W} = \boldsymbol{U}^\star \boldsymbol{\Lambda}^\star \boldsymbol{U}^{\star\top} + \boldsymbol{W},$$

and wish to recover $\boldsymbol{U}^\star \in \mathbb{R}^{n \times r}$. In addition to the rank-2 setting discussed in Section 5.2, we include experimental results for the case when $\boldsymbol{M}^\star$ has rank 3. We also provide estimation error under $d_{2,\infty}$ for our estimate $\widehat{\boldsymbol{U}}$ in Algorithm 2 and the spectral estimate $\boldsymbol{U}$, to complement our results under $d_\infty$ reported in Section 5.2. The experiments follow the same setup outlined Section 5.2, and we refer the readers there for details regarding running Algorithm 2 the generation of $\boldsymbol{\Lambda}^\star$, $\boldsymbol{U}^\star$ and $\boldsymbol{W}$.

We first provide more details as to how we obtain the estimation error under $d_{2,\infty}$. Normally, since

$$d_{2,\infty}(\widehat{\boldsymbol{U}}, \boldsymbol{U}^\star) = \min_{\boldsymbol{\Gamma} \in \mathbb{O}_r} \left\| \widehat{\boldsymbol{U}}\boldsymbol{\Gamma} - \boldsymbol{U}^\star \right\|_{2,\infty},$$

obtaining the $d_{2,\infty}$ estimation error requires finding an orthogonal matrix that minimizes a nonsmooth function, which is hard to achieve. In our simulation, however, since we have sufficiently large eigengaps between the eigenvalues, we know how each column of $\widehat{\boldsymbol{U}}$ corresponds to the columns of $\boldsymbol{U}^\star$. Thus, we merely need to resolve the ambiguity in the sign of each column of $\widehat{\boldsymbol{U}}$, rather than searching over all $\boldsymbol{\Gamma} \in \mathbb{O}_r$. By assigning each column of $\widehat{\boldsymbol{U}}$ the sign of the corresponding column in $\boldsymbol{U}^\star$, we resolve this ambiguity and can obtain the $d_{2,\infty}$ estimation error.

Figure 4 compares the empirical accuracy, measured by $d_{2,\infty}$, of estimating $\boldsymbol{U}^\star$ via the $r$ leading eigenvectors $\boldsymbol{U}$ of $\boldsymbol{Y}$ (blue lines) and via $\widehat{\boldsymbol{U}}$ produced by Algorithm 2 (orange lines). Shaded bands are generated from point-wise 95% confidence intervals using bootstrap approximation. Similar to the rank-one setting, Algorithm 2 recovers $\boldsymbol{U}^\star$ with a much smaller estimation error under $d_{2,\infty}$ compared to the naïve spectral estimate, especially when the coherence $\mu$ is large. The figure also shows that Algorithm 2 performs well under different noise distributions.

Figure 5 corresponds to $\boldsymbol{M}^\star$ having rank-3. It compares the empirical accuracy, measured under $d_\infty$, of estimating each $\boldsymbol{u}_k^\star$, for $k = 1, 2, 3$ via the leading eigenvectors $\boldsymbol{u}_k$ of $\boldsymbol{Y}$ (blue/purple lines) and via $\widehat{\boldsymbol{u}}_k$ given by Algorithm 2 (orange/red lines). Shaded bands are generated from point-wise 95% confidence intervals using bootstrap approximation. As in the first plot of Figure 2, under Gaussian noise, the estimation error of $\widehat{\boldsymbol{u}}_k$ shows little to no visible dependence on $\mu$, and is much smaller compared to the leading eigenvectors of $\boldsymbol{Y}$. Under Rademacher noise, as in the second plot

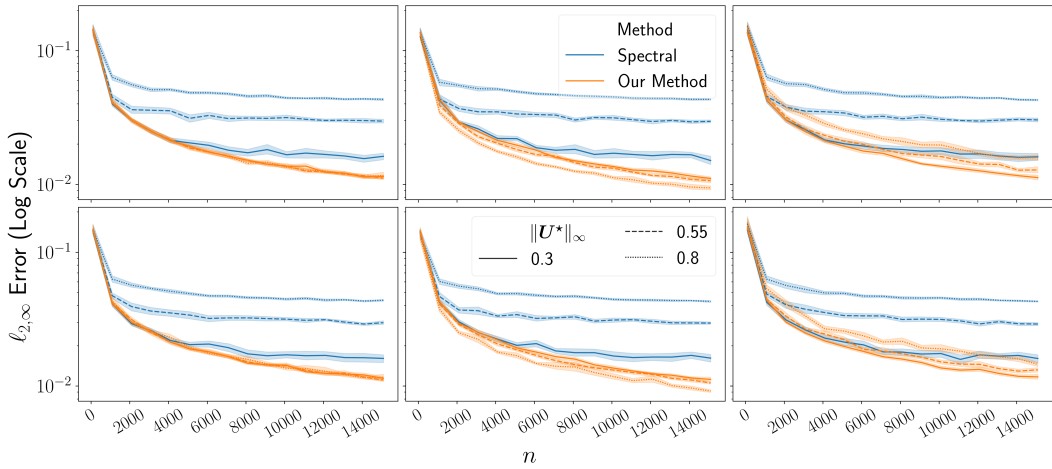

Figure 4: Error as measured in $d_{2,\infty}$ as a function of matrix dimension $n$, by spectral estimate (blue) and the estimator in Algorithm 2 (orange) for three different choices of $\|\boldsymbol{U}^\star\|_\infty$: $0.8$ (dotted lines), $0.55$ (dashed lines) and $0.3$ (solid lines). Columns correspond to $\boldsymbol{W}$ being Gaussian (left), Rademacher (center) and Laplacian (right). The rows correspond to the signal matrix having rank-2 (top) and rank-3 (bottom).

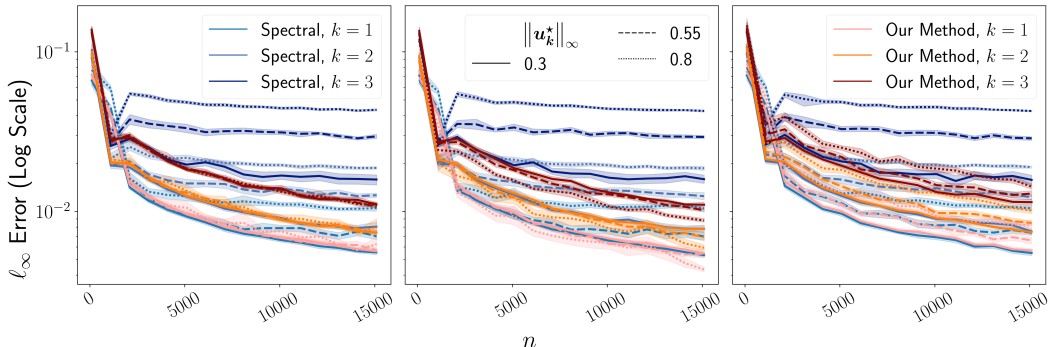

Figure 5: Error measured in $d_\infty$ as a function of matrix dimension $n$, for the three leading signal eigenvectors $\boldsymbol{u}_k^\star, k = 1, 2, 3$ (line width) by the spectral estimate (blue/purple) or the estimator given in Algorithm 2 (orange/red) for three different choices of $\|\boldsymbol{u}^\star\|_\infty$: $0.8$ (dotted lines), $0.55$ (dashed lines) and $0.3$ (solid lines). The plots correspond to $\boldsymbol{W}$ being Gaussian (left), Rademacher (center) and Laplacian (right).

of Figure 2, the dependence on $\mu$ again appears slightly reversed from that of the spectral estimator. For Laplacian noise, as in the third plot of Figure 2, there again seems to be a slight dependence on $\mu$. For the same reason discussed in Sections 5.1 and 5.2, we expect such dependence to be of smaller order than the rate in Theorem 1 and should not affect the asymptotic error rate.

## I.3 Comparison with AMP-based eigenvector estimation

As mentioned in the main text, to the best of our knowledge, we are the first paper to consider the problem of non-spectral entrywise eigenvector estimation. The nearest obvious method to serve as a comparison point, if we were to insist upon one, would likely be one based on approximate message passing (AMP; see [34] for an overview). AMP methods make no explicit coherence assumptions, but the underlying mechanism essentially requires incoherence: inherent to AMP-based eigenvector recovery methods is that the eigenvector is modeled as having its entries drawn i.i.d. according to a common distribution. Specifically, AMP methods typically make a mean field assumption whereby the empirical distribution of the entries of $\boldsymbol{u}^\star$ converges in $\ell_2$ to some distribution $\pi$. Since

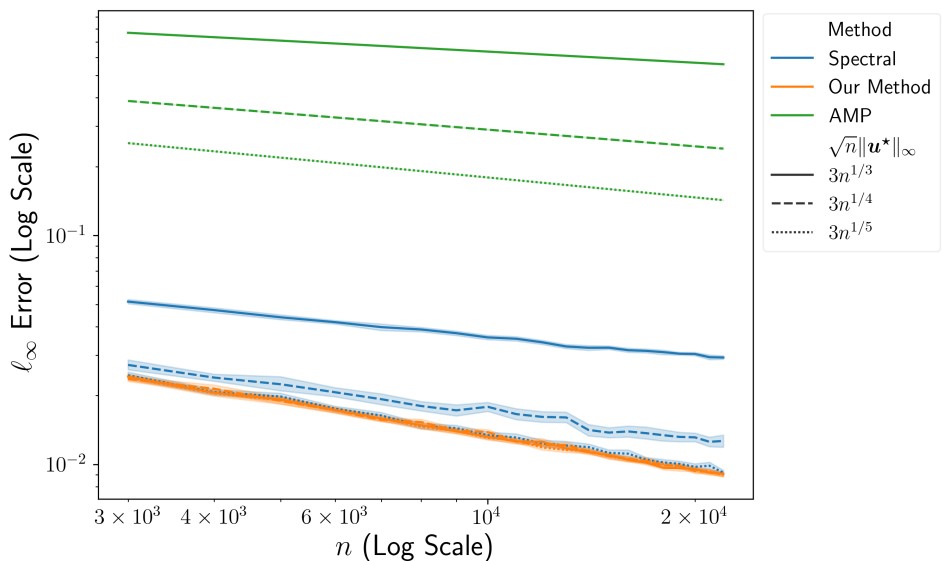

Figure 6: Estimation error under $d_\infty$ as a function of size $n$, by approximate message passing (AMP; green), the leading eigenvector (blue) and Algorithm 1 (orange) for $\sqrt{n}\|\boldsymbol{u}^\star\|_\infty$ equal to $3n^{1/3}$ (solid lines), $3n^{1/4}$ (dashed lines) or $3n^{1/5}$ (dotted lines) under Gaussian noise. Each data point is the mean of 30 independent trials.

AMP-based methods are tailored to $\ell_2$-recovery, they are suboptimal for entrywise recovery problems: small $\ell_2$ error does not necessary imply small entrywise error.

The unsuitability of typical AMP methods notwithstanding, we include here a comparison against our Algorithm 1 for the sake of completeness. The experimental setup mirrors that of Figure 1, but now includes estimation error for an AMP-based method, as well. We generate $\boldsymbol{u}^\star$ according to the following procedure: set a random entry of $\boldsymbol{u}^\star$ to be $a \in \{3n^{1/3}/\sqrt{n}, 3n^{1/4}/\sqrt{n}, 3n^{1/5}/\sqrt{n}\}$, then generate the remaining entries by drawing uniformly from $\{\pm 1\}^{n-1}$ and normalizing these to have $\ell_2$ norm $\sqrt{1-a^2}$. This way, $\|\boldsymbol{u}^\star\|_\infty = a$ and the coherence is $\mu = a^2 n$. We choose this setting to ensure that the limiting prior distribution satisfies the assumptions required by AMP, and thus we can apply the update procedures using Equations (22) (34) and the example after Equation (35) in [34].

Having generated $\boldsymbol{Y} = \boldsymbol{M}^\star + \boldsymbol{W}$, we estimate $\boldsymbol{u}^\star$ using the spectral estimate $\boldsymbol{u}$, the AMP method and our method as described in Section 5 and measure the estimation error under $d_\infty$. We report the mean of 30 independent trials for each combination of conditions (i.e., each combination of problem size $n$ and magnitude $a$). We vary the matrix size $n$ from 3000 to 22000 in increments of 1000.

Figure 6 compares the entrywise estimation error of the AMP method (green), the leading eigenvector of $\boldsymbol{Y}$ (blue) and our Algorithm 1 (orange). Across settings, Algorithm 1 recovers $\boldsymbol{u}^\star$ with a much smaller error under $d_\infty$ compared to the other two estimates, and shows an error rate with no visible dependence on the coherence. Both the spectral method and the AMP-based method exhibit different estimation error rates as the coherence increases. In particular, the AMP method performs much worse due to the fact that it is a mean field approximation method and not well-suited for our setting. Adapting AMP-based methods to target entrywise recovery rather than $\ell_2$ recovery, in hopes of rectifying the poor performance exhibited in Figure 6, is a promising direction for future work.

