# OpenReview forum: "Coherence-free Entrywise Estimation of Eigenvectors in Low-rank Signal-plus-noise Matrix Models"
_NeurIPS.cc/2024/Conference — NeurIPS 2024 poster_

### Official Review · Reviewer_pryd · 2024-07-09

**Soundness:** 4
**Presentation:** 3
**Contribution:** 3
**Rating:** 7
**Confidence:** 3

**Summary:**

The authors propose a new method for coherence free entrywise estimation of eigenvectors in signal plus noise model. Namely, entrywise estimation error usually depends on incoherence of the underlying matrix and can significantly increase error bounds for coherent matrix estimation. In this work, authors show that in suitable regime, entrywise error for recovery of rank-1 matrices scales provably as $\\tilde{O}(\\sigma/\\vert \\lambda^\\star \\vert)$ w.h.p. This is achieved by reestimating eigenvector entries with high amplitude. Moreover, authors propose general rank-$r$ algorithm that they empirically validate. Finally, authors prove a new lower bound on minimax eigenvectors estimation in $\Vert \cdot \Vert_{2\to\infty}$.

**Strengths:**

1) Theorem 1 showing coherence free entrywise estimation of eigenvectors is very interesting. It is a very practical result that can improve any experiments requiring good entrywise estimates. Also, the algorithm itself and its guarantee are interesting for their own sake.

2) Even though authors do not prove guarantee in general rank-$r$ setting, it is praiseworthy that they propose and empirically evaluate a generalization of rank-$1$ algorithm.

3) As authors nicely describe in Section 1.2, lower bounds for matrix (or eigenvector) estimation in $\\Vert \cdot \\Vert_{2\to\\infty}$ are usually derived based on lower bounds for Frobenius norm estimation, and are generally not tight. I am not aware of any previous results that are as tight as the one claimed in Theorem 2.

4) Lastly, experimental results complement really well theoretical results, and show that proposed algorithms look very promising even in practice.

**Weaknesses:**

1) The main theorem is proven only in rank-$1$ setting, and rank-$r$ setting is only empirically tested.

2) Gaussianity assumption is restricting. If your results hold under less restrictive assumptions (you mention Assumption 1 in Chen et al. 2021), I would prefer having at least a statement of an analogue of Theorem 1 in the most general setting you can have.

3) Bounds might be improvable in log terms.

**Questions:**

1) You consider only symmetric matrices in the paper. Are all results easily transferable to asymmetric case (for example, by symmetrization trick)?

2) In rank-$r$ case when you split matrix $Y$ into $\\lambda_k^{\\star} u_k^{\\star} {u_k^\\star}^\\top$ and the remaining terms that you consider as noise, how would you mitigate the fact that this new noise containing all non-$k$ eigenvectors is dependent on signal i.e. on $k$-th eigenvector? Is this an issue at all?

3) Could you please give some more precise hints why is rank-$r$ case more difficult than rank-$1$ case?

4) Are there any other entrywise lower bounds in the literature that are not simple corollaries of Frobenius lower bounds?

5) How does your method compare with other coherence free methods? For example, using leverage scores for sampling more the entries with high coherence (effectively reducing noise on those entries)? I agree that your model is not the same, but if you could comment on high level differences between the two methods.

**Limitations:**

The authors have addressed limitations adequately.

---

> ### Author Rebuttal · Authors · 2024-08-06
>
> We thank the reviewer for their praise and for their thoughtful suggestions.
> Specific responses to their concerns and questions are given below.
>
> 1) The main theorem is proven only in rank-$1$ setting, and rank-$r$ setting is only empirically tested.
>
> Please see our discussion in the global rebuttal.
>
> 2) Gaussianity assumption is restricting. If your results hold under less restrictive assumptions (you mention Assumption 1 in Chen et al. 2021), I would prefer having at least a statement of an analogue of Theorem 1 in the most general setting you can have.
>
> We note that in all cases where we invoke Gaussianity, we do so only to ensure an appropriate tail bound (i.e., concentration inequality).
> As such, our Gaussian assumption can be relaxed to a Gaussian tail decay.
> We will update the theorem statements and proofs accordingly to encompass this more general case.
>
> 3) Bounds might be improvable in log terms.
>
> We agree with this conjecture.
> As with many results in the low-rank estimation literature, the logarithmic terms in our bounds are incurred typically due to union bounds and/or the extra factors needed to ensure that bounds hold with probability polynomially small in $n$.
> As the referee is surely aware, removing these terms typically requires a great deal of highly technical work for a gain that is interesting primarily as a technical challenge.
> As such, we would suggest leaving this further analysis for future work in a probability or mathematics journal.
>
> ===Questions===
>
> Q1) You consider only symmetric matrices in the paper. Are all results easily transferable to asymmetric case (for example, by symmetrization trick)?
>
> In short, the answer is yes, using standard matrix dilation arguments (i.e., symmetrization).
> We will add a sketch of this argument to the text.
>
> Q2) In rank-$r$ case when you split matrix $Y$ into $\lambda^*_k u_k^* u_k^{* T}$ and [noise terms], how would you mitigate the fact that this new noise [...] is dependent on signal i.e. on $k$-th eigenvector? Is this an issue at all?
>
> Recall that we conjugate $Y$ by a Haar matrix $H$.
> Let $v$ be the leading eigenvector of $H Y H^T$ and let $Q$ be diagonal with entries given by the signs of the entries of $v$.
> Let $\xi$ be the indicator of whether or not the norm of $v$ is at least $|v_{(\lfloor n/2 \rfloor)}|$.
> Consider the case where $r = 2$.
> Our estimate is of $u^*_1$ is given by
>
> $\hat{u}_i = \frac{\lambda_1^* u_1^{* \top} H Q \xi}{\hat{S} \sqrt{\hat{\lambda}_1}} u_1^* + \frac{\lambda_2^* u_2^{* T} H Q \xi}{\hat{S} \sqrt{\hat{\lambda}_1}} u_2^* +\frac{W H Q \xi}{\hat{S} \sqrt{\hat{\lambda}_1}}.$
>
> Compared to the $r=1$ case, the middle term is new.
> Let us sketch why this term does not much matter.
> With high probability, the diagonal of $Q$ is the sign of some entries of $u_1^{* T} H$, which is almost independent of $u_2^{* T} H$ by properties of random vectors uniformly distributed on $\mathbb{S}^{n-1}$.
> Thus, one expects that $u_2^{* T} H Q \xi = O(\sqrt{\log n})$ with high probability, and the error introduced by $u_2^{* T}$ is at most $O(\sqrt{n^{-1} \log n})$ when $|\lambda_2^*| = \Theta(\lambda_1^*)$.
> This implies that when $|\lambda_1^*| = \tilde{\Theta}(\sigma \sqrt{n})$, we would expect an almost coherence free rate.
> Whether we can extend the proof to $|\lambda_1^*| \gg \sigma \sqrt{n})$ remains uncertain, but at least we would expect our method to work in the most interesting regime $|\lambda_1^*| = \tilde{\Theta}(\sigma \sqrt{n})$.
> Alternatively, we can possibly be more careful in selecting $\xi$, so that $u_1^{* T} H Q \xi = \Theta(n)$ while $u_2^{* T} H Q \xi \ll O(\sqrt{\log n})$ and the coherence-free rate holds in a wider range of $\lambda_1^*$.
>
> Q3) Could you please give some more precise hints why is rank-$r$ case more difficult than rank-$1$ case?
>
> In the rank-$r$ case, when we sum over the large entries of one spike, we need to also consider whether the other spikes affects the estimation.
> We choose to use a random orthogonal transformation to get the spikes into general positions, so that when summing over the large entries of one spike, the effect of other spikes is ignorable.
> As mentioned in the intuition of Alg. 1, our method basically has two steps: selecting a suitable subset, and fitting on this subset.
> In the rank-one case, we make Assumption 2 so that we can perform selection and fitting on the same data.
> The random orthogonal transformation we use for the rank-$r$ case breaks Assumption 2 and introduces some complicated dependence, which we do not yet know how to resolve technically.
>
> Q4) Are there any other entrywise lower bounds in the literature that are not simple corollaries of Frobenius lower bounds?
>
> To the best of our knowledge, this is the first such entrywise result that does not simply fall back on Frobenius lower bounds.
>
> Q5) How does your method compare with other coherence free methods? For example, using leverage scores for sampling more the entries with high coherence (effectively reducing noise on those entries)? I agree that your model is not the same, but if you could comment on high level differences between the two methods.
>
> We are not aware of any other coherence-free methods against which to compare.
> The referee's suggestion to use leverage scores to identify high-coherence entries is a good one, except that this is not easily done without already having a good estimate of the coherence.
> Indeed, at a high level, the intuition behind our Algorithm 1 is precisely that of the referee: we want to focus our efforts on the high-coherence entries, but we must identify them first.
> We will update the manuscript to discuss this point in more detail.
>
> If one must find a method to compare against aside from the ``purely spectral'' method in our experiments, the best we are aware of would be an approximate message passing (AMP) method for eigenvector recovery.
> See our discussion in the global rebuttal.

---

> ### Comment · Reviewer_pryd · 2024-08-12
>
> Thank you for your reply. I acknowledge reading the rebuttal and will maintain my initial score.

---

### Official Review · Reviewer_nQFd · 2024-07-11

**Soundness:** 2
**Presentation:** 2
**Contribution:** 2
**Rating:** 4
**Confidence:** 4

**Summary:**

The authors consider the spiked Gaussian Wigner matrices, where the main goal is to estimate the (low-rank) spike. Since the known performance of the spectral method for the estimation (of the spike) deteriorates as the maximal entry of the spike (more precisely, the incoherence parameter) increases, the authors propose a new algorithm that does not depend on the incoherence parameter. Roughly, the main idea of the proposed algorithm is that under several assumptions the entries of the noisy data corresponding to the large entries of the spike are dominated by the spike, and thus those entries themselves can be used to approximate the spike instead of the eigenvectors of the data matrix. Mathematical analysis and numerical experiments for the algorithm are presented.

**Strengths:**

- The proposed algorithm is new, and the error bound indeed does not depend on the incoherence parameter.
- The error bound of the algorithm is mathematically analyzed and also tested by numerical experiments.

**Weaknesses:**

- Non-spectral methods are not discussed. Since the proposed algorithm is not entirely spectral, I think its performance should be compared with other non-spectral methods as well.
- Assumption 2 is strange and cannot hold in many important cases. For example, if the spike $u^*$ contains many entries of the size $n^{-\alpha}$, then Assumption 2 may not hold due to a similar reason as it does not hold when $u^*$ is drawn uniformly from $S^{n-1}$.
- Several claims are not rigorous in the sense that they are cited from references in which assumptions are different from those in the current manuscript.  (See Questions.)

**Questions:**

Below, I collected several previous results used in the current paper that are not directly applicable since the assumptions in the original papers are different from those in the current paper.
- In line 44, the results in [6] assume that all entries of the spike $u^*$ are $O(1/\sqrt{N})$.
- In line 46, the original BBP transition in [8] is not for the signal-plus-noise matrix models. (It was for a Gaussian matrix where the spike is contained in the covariance matrix.)
- In line 52, the case $|\lambda^*| \gg \sqrt{n}$ is not considered in [8] and thus it is unclear whether the results in Lemma 1 can be applied to this case. Moreover, strictly speaking, when $|\lambda^*| = \Theta(\sqrt{n \log n})$, Lemma 1 only says that $\liminf_{n \to \infty} d_{\infty} (u, u^*) \geq 0$, not about the asymptotic bound for $d_{\infty} (u, u^*)$.
- In line 144, the result in [22] is under the assumption that $|\lambda^*| = \Theta(\sqrt{n})$ and the result in [45] is under the assumption that the noise matrix is GOE. (The noise matrix $W$ in the current paper is not a GOE matrix since the variance of the diagonal entries is the same as that of the off-diagonal entries.)
- In the inequality below line 491, since the probability estimate on $\max |W_{ii}|$ is basically a union bound, with the coefficient $4$, it seems to hold only with probability $1-O(n^{-7})$.

**Limitations:**

The work does not seem to have potential negative societal impact.

---

> ### Author Rebuttal · Authors · 2024-08-06
>
> We thank the referee for their careful reading.
> We must politely disagree with their correctness concerns, which mostly relate to citations in the literature review.
> These provide context and background to our paper and are unrelated to our proofs.
> We have clarified these points below and will edit the manuscript accordingly.
> We trust this will assuage the referee's concerns.
>
> 1) [...] performance should be compared with other non-spectral methods [...].
>
> The most obvious competing method is approximate message passing (AMP; see the global rebuttal).
> AMP methods typically require a limiting prior distribution $\pi$ for the entries of $\sqrt{n} u^*$, which limits performance when $u^*$ is coherent.
> To see this, let $\nu_n$ be the empirical distribution of the entries of $\sqrt{n} u^*$.
> $\pi$ must be the limit of $\nu_n$ and obey the conditions of Prop. 3.4 in Feng, et al (see also "Information-theoretically optimal sparse PCA" by Deshpande and Montanari, Section 2.1).
> If one "hides" a few moderate entries in $u^*$, increasing coherence without changing the limiting distribution, entrywise estimation fails.
> On the other hand, one can show that a highly coherent $u^*$ violates the conditions needed by AMP.
>
> Setting this aside, the global rebuttal PDF shows an experiment comparing AMP against our Alg. 1, where our method consistently outperforms AMP.
> We will add a detailed description of this experiment and why $\ell_2$ methods are unsuitable for entrywise estimation.
>
> 2) Assumption 2 is strange and cannot hold in many important cases. [...] if [$u^*$] contains many entries of the size $n^{-\alpha}$, then Assumption 2 may not hold [...].
>
> We agree Assumption 2 is strange. See our discussion in the global rebuttal.
>
> The referee is incorrect in their counter-example.
> If their $\alpha$ is not $1/2$, then absent a very careful choice of spacing, there will be a gap between the larger and smaller entries of $u^*$ and Assumption 2 will hold.
> The important condition is the existence of *some* $\alpha \in \mathcal{A}$.
> Choosing one "bad" $\alpha$ as the referee seems to suggest need not violate Assumption 2.
>
> 3) [...] [6] assumes all entries of the spike are $O(1/\sqrt{n})$.
>
> We cite [6] illustrate a widespread belief that when $\lambda^*$ is too small, estimation is difficult or impossible (see, e.g., the discussion after Lemma 1 in [6]).
> The citation is not toward a proof.
>
> 4) [...] the original BBP transition in [8] is not [in this paper's setting]
>
> Yes, [8] is for Gaussian covariance.
> Similar results for many other models are well known to researchers in this community.
> For the model in our paper, see "The eigenvalues and eigenvectors of finite, low rank perturbations of large random matrices" by Benaych-Georges and Nadakuditi, or Section 3 of the AMP survey above.
> We will add these to the manuscript.
>
> 5) [...] the case $\lambda^* \gg\sqrt{n}$ is not considered in [8] and thus it is unclear whether [...] Lemma 1 [applies]. When $|\lambda^*| = \Theta( \sqrt{n \log n} )$, Lemma 1 only says that $\lim\inf_n d_\infty(u,u^* )\ge 0$, not about the asymptotic bound for $d_\infty(u,u^* )$.
>
> The setting $\lambda^* \gg \sqrt{n}$ is indeed not in [8], but our aim is not to apply [8] in relation to Lemma 1 (or vice versa).
> Lemma 1 suggests that the upper bound in Eq. (4) is not generally improvable, so the error in the leading sample eigenvector must depend on coherence.
> Our paper removes this dependence by considering methods that are not based on the leading eigenstructure alone.
> In other words, Lemma 1 supports the idea that to remove dependence on coherence, we need a method that is not purely spectral.
> Lemma 1 indeed does not directly apply to $|\lambda^*| = \Theta( \sqrt{n \log n} )$, but it suggests a similar lower bound on the entrywise error of the spectral estimator, since there is no phase transition in this regime.
> A finite-sample version of Lemma 1 in this regime is also possible.
>
> 6) [...] [22] is under the assumption that $|\lambda^*| = \Theta( \sqrt{n} )$ and [...] [45] is under the assumption that [$W$] is GOE. [$W$ is not GOE due to the diagonal.]
>
> As stated at the bottom of page 4, Alg. 1 yields a new estimator for $\lambda^*$, investigated in Appendix E.
> The Equation (14) estimator is a known quantity to most researchers in this subfield.
> We use it in our experiments instead of our new estimator so we can investigate Alg. 1 and 2 without accounting for using a new eigenvalue estimator at the same time.
> We omit lengthy discussion of the estimator in Equation (14), as these details have been established elsewhere.
>
> The citations disputed by the referee point out other works that have studied this estimator, providing context to our use of it.
> Citations [22,25,49,16] all show versions of the decomposition $\lambda=\lambda^* +n\sigma^2 /\lambda^* +O(\sigma\sqrt{\log n})$.
> These are to highlight papers where similar ideas have been used, not because they all hold in our setting.
> The necessary proof ideas for our paper are found in [25].
> We will edit the text to make this more clear.
>
> The referee is correct that $W$ is not precisely GOE.
> The diagonal entries influence important quantities in random matrix theory (e.g., the leading eigenvalue).
> In our problem, though the $n$ on-diagonal entries are swamped by the $O(n^2)$ off-diagonal entries.
> Our proofs account for this.
> One could also use Weyl's inequality after uniformly bounding the diagonal entries as $O(\sqrt{\log n})$.
> This factor is swamped by the $\sqrt{n}$ factors elsewhere in the relevant bounds.
> See also Theorem 1.3 in "The largest eigenvalue of rank one deformation of large Wigner matrices" by Feral and Peche.
> We cite [45] to show where this estimator has been studied before, not necessarily under the exact same setting as us.
> We will clarify this in the text.
>
> 7) [The probability $n^{-7}$ bound below line 491 has the wrong exponent]
>
> We have increased the constant on the union bound and corrected Lemma 3 accordingly.

---

> > ### Comment · Reviewer_nQFd · 2024-08-13
> >
> > Thank you for the answers. I have checked the responses.

---

### Official Review · Reviewer_85jE · 2024-07-13

**Soundness:** 3
**Presentation:** 4
**Contribution:** 3
**Rating:** 7
**Confidence:** 2

**Summary:**

The paper studies the low rank matrix estimation problem. It aims to find an estimator that is good with respect to the $\ell_{2,\infty}$ norm. In general, such errors depend on incoherence parameters. The authors propose and prove a spectral algorithm that does not depend on the coherence parameters when the top eigenvalue of the signal is of order $\sqrt{n \log n}$. Furthermore, the paper proves estimation lower bounds with respect to the $\ell_{2,\infty}$ distance when the operator norm of the signal is on the same scale as the noise.

**Strengths:**

1) The authors introduce an efficient spectral algorithm to compute an estimator that out performs the spectral estimator (in terms of the $\ell_{2,\infty}$ distance) and tackles the case when the incoherence parameter $\mu$ is large with respect to $n$. The rates of convergence of the estimator do not depend on $\mu$. The algorithm is new and it appears to be a strict improvement over the naive estimator.
2) The main theorems in the paper are supported by detailed proofs of all results. The proofs are nicely written and the presentation of the results are clear and easy to follow. Furthermore, numerical experiments further support the claims and possible generalizations and weakening of the assumptions of the main results.
3) Although the Gaussian noise is required in Assumption 1, it appears that it can be removed quite easily. For instance, it appears that Lemma 1 does not use the Gaussian nature of $W$ at all.

**Weaknesses:**

1) The authors are able to prove a nice rate of convergence for the algorithm. Unfortunately, the proof relies on some technical assumptions to simplify the proof. For instance, the application of Lemma 9 relies crucially on the fact that $s$ and $I_\alpha$ are independent of $W$. This technical obstruction is dealt with quite creatively by introducing non-random sets $I_\alpha$, albeit at the cost of additional assumptions on the model.
2) An algorithm for finite rank spikes are proposed, but the generalization of Theorem 1 to the finite rank case has not been proven.
3) Theorem 2 is stated when $\lambda$ is a constant multiple of the identity, so it is slightly more restrictive than in equation 9.
4) Assumption 2 seems slightly limiting. It appears like a difficult condition to verify in practice.

**Questions:**

1) Assumption 2 is slightly hard to parse. It seems like it is quite easy to violate Assumption 2 by introducing some randomness in the generation of $u^\star$. Is it true that if $u^\star$ was generated by normalizing a vector with i.i.d entries that assumption 2 will be violated?
2) The subscripts of the expected value in Theorem 2 is mysterious. It appears that the $\Lambda_\star$ and $U_\star$ are non random, and the only randomness is in $W$. Perhaps some notational clarification is needed here?
3) Is it possible to extend Theorem 2 to general $\Lambda$ which are not necessarily constant multiples of the identity?
4) Perhaps proving an uniform bound in Lemma 9, will allow us to do a proof without assumption 2, since we can handle cases when $\hat I$ and $W$ are dependent. However, we will likely lose the $\sqrt{\log n}$ bound if we wanted something uniform.

Typos:
1) Line 449: A $\sum_{i}$ is missing
2) Line 673: It should be $\mathcal{E}_{5,\alpha}^c$ and the complement outside of the bracket should not be there.
3) Line 954: an extra $\leq$ appears
4) Line: 1046: it should be $\mathbb{K}_{r,\mu}$

**Limitations:**

The limitations of the assumptions are clearly stated in remarks.

---

> ### Author Rebuttal · Authors · 2024-08-06
>
> We thank the reviewer for their positive assessment and thoughtful suggestions. We address their concerns and questions below.
>
> 1) [The authors] prove a nice rate of convergence for [Alg. 1]. Unfortunately, the proof relies on some technical assumptions to simplify the proof. [remainder elided for space]
>
> Please see our discussion of Assumption 2 in the global rebuttal.
>
> 2) [...] the generalization of Theorem 1 to [$r > 1$] has not been proven.
>
> We agree that a proof for general $r \ge 1$ would be ideal. Please see our discussion of this in the global rebuttal.
>
> 3) Theorem 2 is stated when $\Lambda$ is a constant multiple of the identity, so it is slightly more restrictive than in equation 9.
>
> The referee is correct that Theorem 2 takes $\Lambda^*=\lambda^* I$, which is more restrictive than Eq. (9).
> To clarify, $\Lambda^*$ is a scalar multiple of $I$, but that scalar need not be constant: we require only $ 0<\lambda^*\le C\sigma\sqrt{n}.$
> If $\lambda^*$ grows faster than this, we are in the regime from Section 2.
>
> More importantly, Theorem 2 establishes a limit to estimation.
> The result implies eigenvector recovery is hard when $\Lambda^*$ is a scalar multiple of the identity, so the problem is no easier for a larger class of matrices.
>
> We believe Theorem 2 can be adapted to allow structure in $\Lambda^*$.
> In particular, we believe that the problem is hardest when the smallest eigengap in $\Lambda^*$ is small.
> This would suggest that $\Lambda^* = \lambda^* I$ is the hardest setting.
> We will add a brief discussion of this after Theorem 2, leaving a precise treatment to future work.
>
> 4) Assumption 2 [appears difficult to check in practice].
>
> Please see our discussion of Assumption 2 in the global rebuttal.
>
> As for verifying Assumption 2 in practice, this is perhaps a case where assumptions are part and parcel of the method (e.g., $t$-tests require independence which is usually hard or impossible to verify).
> That said, as mentioned around line 253, a fundamental step in our implementation of Alg. 1 (and in our stated Alg. 2) is to conjugate $Y$ by a Haar-distributed orthogonal matrix.
> This essentially obliterates any structure in $u^*$: $H u^*$ is Haar-distributed.
> The success of Alg. 1 despite this suggests practitioners should not be too concerned about Assumption 2 when applying this method.
>
> Questions
>
> Q1) [it seems easy to violate Assumption 2.] Is it true that if $u^*$ was generated by normalizing a vector with i.i.d entries that assumption 2 will be violated?
>
> Whether or not $u^*$ obeys Assumption 2 is somewhat technical.
> If $u^*$ is generated by renormalizing iid Gaussians, then Assumption 2 fails: the resulting vector is Haar-distributed.
> This is the focus of Fig. 1, where we explore two settings that violate Assumption 2: $u^*$ (before normalization) has iid Gaussian (top row) and iid Bernoulli (bottom row) entries.
> Per Fig. 1, our method outperforms "pure" spectral methods even in this "bad" setting.
>
> The referee is incorrect that renormalizing a vector of iid entries must violate Assumption 2.
> If the distribution has suitable structure, $u^*$ may still obey Assumption 2.
> Suppose $u^*$ is obtained by renormalizing a vector $g=(g_1,g_2,\dots,g_n)$ with iid entries from a distribution with variance $1$, so $u^*\approx g/\sqrt{n}$.
> If the $g_i$ are drawn by taking $g_i=a$ with probability $p$ and $g_i=b$ with probability $1-p$, then each entry of $u^*$ is either $\approx ap/\sqrt{n}$ or $\approx b(1-p)/\sqrt{n}$.
> The result is a $O(n^{-1/2})$ gap between the entries of $u^*$, and we can take $\alpha_0=(\log n)^{-L}\approx n^{-1/2}$.
> This is a "random" analogue of our example after Assumption 2, where we take $u^*=1_n/\sqrt{n}$.
>
> A complete account of which distributions do and do not violate Assumption 2 is interesting, but perhaps beside the point, since Assumption 2 seems removable.
> We will add examples after Assumption 2 to further elucidate this point.
>
> Q2) [... it appears that] $\Lambda^*$ and $U^*$ are non random, and the only randomness is in $W$. [...]
>
> As mentioned above, Theorem 2 establishes a fundamental limit to how well we can estimate $U^*$.
> The standard tool for these bounds, which we use here, requires finding a collection of parameters (in our case, choices of $U^*$ and $\Lambda^*$) that are hard to distinguish based on observed data (i.e., $Y$).
> This should clarify why $U^*$ and $\Lambda^*$ are not random: they are model parameters, not random variables.
>
> Perhaps the referee has in mind a Bayesian paradigm, where $U^*$ and $\Lambda^*$ have priors.
> This can be done, but the lower bound in Theorem 2 holds as a statement about estimation in general.
>
> Q3) Is it possible to extend Theorem 2 to general $\Lambda^*$ [that are not constant multiples of the identity]?
>
> See our response to Concern 3 above.
>
> Q4) [perhaps Assumption 2 can be removed via a union bound]
>
> We considered this when working on our proofs.
> Indeed, we apply Lemma 9 in a union bound over $\mathcal{A}$.
> Unfortunately, the "bigger" union bound the referee seems to be suggesting (i.e., over all $n$-dimensional binary vectors) incurs a factor of $2^n$ in the probability bound, which incurs an extra $\sqrt{n}$ in the error bound, which is too loose for our purposes.

---

> > ### Comment · Reviewer_85jE · 2024-08-12
> >
> > Thank you for the detailed response. I have no further questions, and will maintain my original score.

---

### Official Review · Reviewer_qfDx · 2024-07-13

**Soundness:** 3
**Presentation:** 4
**Contribution:** 4
**Rating:** 7
**Confidence:** 3

**Summary:**

This paper proposes an algorithm to estimate the eigenvector of a low-rank matrix under Gaussian noise. The algorithm provides a $\ell_{\infty}$ guarantee that is coherence free for rank-one matrices, at the cost of worsening the dependence on $\log n$ and some technical assumptions. The main idea is to utilize the low-rank structure and relies more on the "stronger" entries that are much larger than the noise rather than "weaker" entries. Empirical evidence shows that the algorithm continues to work for general low-rank matrices.

**Strengths:**

The result is a welcomed addition to the literature of low-rank estimation. Coherence-free estimation is an important step to get closer to minimax optimal estimation. Due to time constraints, I cannot check all the details of the proof, but the overall approach appears reasonable.

**Weaknesses:**

The main weakness is Assumption 2 and Assumption 3, which are a bit weird and could significantly worsen the bound in some cases. Also, the upper bound is only proved for rank-one matrices.

**Questions:**

I have no particular questions that may change my evaluation.

---

> ### Author Rebuttal · Authors · 2024-08-06
>
> We thank the reviewer for their kind words.
> A brief response to their concerns is below.
>
> 1) The main weakness is Assumption 2 and Assumption 3, which are a bit weird and could significantly worsen the bound in some cases.
>
> We agree that Assumption 2 is ungainly. Please see our discussion of this point in the global rebuttal.
>
> As for Assumption 3, we assume that the reviewer is referencing the $\epsilon_0$-dependence.
> As mentioned above, our experiments in Section 5 strongly suggest that this technical condition is not necessary for our results to hold.
> As discussed in Remarks 1 and 2 and sporadically throughout Sections 2 and 6, we believe this should be removable, but a formal proof is difficult and is a current focus of our ongoing research.
> Should a breakthrough be achieved before relevant deadlines, we will update the paper accordingly.
>
> In the event that the referee's concern is with the growth assumption in Assumption 3, we note that this assumption is related to our discussion elsewhere in the paper about the two different growth regimes.
> In short, Sections 2 and 3 concern the setting where $|\lambda^*| = \Omega( \sqrt{n} )$, up to logarithmic factors, hence the growth rate in Assumption 3.
> Section 4 concerns the ``small signal'' setting, where $|\lambda^*| = O( \sqrt{n} )$.
> There, our Theorem 2 provides a lower bound on the estimation rate, which improves previous known lower bounds.
> We perhaps did not adequately highlight the fact that the two main theorems of the paper concern different growth regimes for $|\lambda^*|$.
> We will update the manuscript to make this distinction more clear.
>
> 2) Also, the upper bound is only proved for rank-one matrices.
>
> Please see our discussion of this point in the global rebuttal.

---

> > ### Comment · Reviewer_qfDx · 2024-08-12
> >
> > Thank you for your response. I think the paper contains some interesting new ideas but awaits future work to provide a more complete analysis, e.g. relaxing the assumptions. Therefore, I elect to maintain my score.

---

### Official Review · Reviewer_dN7g · 2024-07-14

**Soundness:** 2
**Presentation:** 3
**Contribution:** 2
**Rating:** 6
**Confidence:** 4

**Summary:**

This paper mainly studies the problem of eigenvector estimation in low-rank signal-plus-noise matrix models and some new lower bounds for estimation rates in such models are derived. Specifically, the entrywise estimation error of the proposed procedure has no dependence on the coherence $\mu$ for the rank-one signal matrices, and could achieve the optimal estimation rate up to log-factors.

**Strengths:**

1. The classical spectral estimator has an intrinsic dependence on the coherence $\mu$. That is, when $\mu$ is large, the low-rank signal exhibits additional structures (e.g., sparsity) beyond low-rankness, and the spectral estimator performs particularly poorly due to its failure to fully utilize these additional structures. This paper proposes a new estimator designed to eliminate this dependence on $\mu$.
2. This paper carefully designs a series of simulations to further validate its theoretical findings (as shown in Figure 1), demonstrating that the proposed estimation procedure has little dependence on the coherence $\mu$.

**Weaknesses:**

1. The theoretical  results presented in this paper only fit for the scenarios where the low-rank signal matrix is symmetric, thereby limiting its practical use.
2. Assumption 2 seems to be confusing, according to the following comments.

a)  Firstly, in the first example given by the authors, and $c_1$ and $c_2$ that satisfy condition $\|u^*\|_2=1$ are related to $n$, while the authors state that they are constants (line 125).

b) Secondly, $\alpha_0$ is related to $n$, meaning $u^*$ is related to $n$. Therefore, under Assumption 2, considering the influence of $\alpha_0$, what will happen to the coherence? For example, will it no longer be related to $n$? In other words, Assumption 2 and the coherence condition are coupled together, making it difficult to determine whether the disappearance of coherence in the proposed method is due to the careful design of the algorithm or the existence of Assumption 2.

3. The lower bound derived in this paper does not seem to align with the environmental conditions. In other words, the upper bound is given under the assumption that Assumptions 1 through 4 are satisfied. When constructing a bad instance to prove the lower bound, this bad instance should also satisfy such assumptions. Additionally, the selection of $\lambda^*$ does not conform to Assumption 3.

4. The work lacks experimental validation on real-world datasets.

**Questions:**

1. The description of Algorithm 1 is too brief. Could you provide a more in-depth discussion?
2. The paper mentions that the computation of the simulation requires 3425 hours, which seems to be very very time-consuming. Do the classical spectral algorithms or other related algorithms also need  this high kind of computational cost?

**Limitations:**

See points 1 and 4 in the weaknesses part above.

---

> ### Author Rebuttal · Authors · 2024-08-06
>
> 1) The theoretical results [...] only fit for the scenarios where the low-rank signal matrix is symmetric, thereby limiting its practical use.
>
> We note that there are many applications (see, e.g., network analysis, neuroimaging, covariance estimation) where the target low-rank matrix to be estimated is symmetric.
> We believe the result can be extended to the asymmetric case (i.e., the signal matrix is now $\lambda^* u^* v^{* T}$) via standard matrix dilation arguments.
> We will add a brief discussion of this point to the paper.
>
> 2) a) [...] in the first example given by the authors, $c_1$ and $c_1$ that satisfy condition $\|u^*\|_2=1$ are related to  $n$, while the authors state that they are constants (line 125).
>
> We apologize for the miscommunication.
> In this example, in which $c_1$ and $c_2$ are chosen to ensure that $u^*$ has norm $1$, both are asymptotically constant with respect to $n$.
> That is, they are both $\Theta( 1 )$.
> We will clarify the example in the manuscript to avoid this misunderstanding.
>
> 2) b) [...] $\alpha_0$ is related to $n$, meaning $u^*$ is related to $n$. Therefore, under Assumption 2, considering the influence of $\alpha_0$, what will happen to the coherence? [...] Assumption 2 and the coherence condition are coupled together, making it difficult to determine whether the disappearance of coherence in the proposed method is due to the careful design of the algorithm or the existence of Assumption 2.
>
> $u^*$ is related to $n$, but not necessarily via $\alpha_0$.
> Inherently, $u^*$ is of dimension $n$, so given a fixed coherence, $n$ partially informs the behavior of $u^*$.
> For a particular choice of coherence, certain values of $\alpha_0$ are compatible with this coherence and others are not.
> For example, if $\alpha_0$ is large, $u^*$ must be coherent.
>
> The coupling of coherence and Assumption 2 is perhaps a red herring: the important aspect of our result is that spectral methods will depend inherently on the coherence whether Assumption 2 holds or not.
> Our method does not depend on the coherence, provided that Assumption 2 holds.
>
> As discussed in the paper, it seems likely that Assumption 2 can be removed, suggesting that the answer to the referee's question "whether the disappearance of coherence in the proposed method is due to the careful design of the algorithm or the existence of Assumption 2." is that the algorithm is the reason for the disappearance of the coherence.
>
> Please see the global rebuttal for further discussion of Assumption 2.
>
> 3) The lower bound derived in this paper does not seem to align with the environmental conditions. In other words, the upper bound is given under the assumption that Assumptions 1 through 4 are satisfied. When constructing a bad instance to prove the lower bound, this bad instance should also satisfy such assumptions. Additionally, the selection of $\lambda^*$ does not conform to Assumption 3.
>
> To clarify, the upper bound in Theorem 1 is under Assumptions 1 through 4 and should be compared to the lower bound in Equation (9).
> The lower-bound in Section 4 concerns a different regime, and establishes an impossibility result.
> In the beginning of Section 4, we explain that the lower bound in Equation (9) may be suboptimal in certain regimes (i.e., when $\lambda^*$ is small).
> Theorem 2 in Section 4 improves the lower bound in Equation (9) in this small-$\lambda^*$ regime.
>
> We will edit the introduction and Section 4 to clarify this point.
>
> 4) The work lacks experimental validation on real-world datasets.
>
> We agree that validation on real world data sets is always nice to have.
> This is a theoretical paper, concerned with understanding the fundamental limits of estimation in a particular problem.
> We would suggest that experiments on real data are best left for follow-up work dedicated to the engineering problems subsequent to this theoretical work.
>
> Questions:
>
> Q1) [...] Could you provide a more in-depth discussion [of Alg. 1]?
>
> Consider an entry of $Y$ given by
> $ \lambda^* u_i^* u_j^* + W_{ij} $.
> If $i,j \in [n]$ correspond to large entries of $u^*$, then $u_i^* u_j^*$ is large, and $Y_{ij}$ has a large SNR.
> If we knew the locations of these large entries, we could obtain a more accurate estimate of $u^*$ by concentrating on these locations.
> Essentially, Alg. 1 and Alg. 2 have two steps: finding large locations, then using them to improve the spectral estimate of $u^*$.
> We will expand the discussion of Alg. 1 accordingly.
>
> Q2) The paper mentions that the computation of the simulation requires 3425 hours, which seems to be very very time-consuming. Do the classical spectral algorithms or other related algorithms also need this high kind of computational cost?
>
> The 3425 number is the total computation time expended on all experiments reported in the paper.
> That is, this is the total number of compute hours  to produce the experimental results (both spectral and our method).
> This was included in accordance with the NeurIPS "Experiments Compute Resources" checklist requirement, which reads "For each experiment, does the paper provide sufficient information on the computer resources (type of compute workers, memory, time of execution) needed to reproduce the experiments?"
> Note that this figure comprised some 6000 or so experimental runs (i.e., individual problem instances) in total, including a few very large-scale problem instances, which account for the vast majority of the total compute time.
>
> The runtimes of our method and spectral methods are quite close, as discussed at the bottom of page 4: our method is a refinement of the spectral estimate, and this refinement can be done quickly.
> We will add more detailed discussion of this point and an explicit timing comparison.

---

> > ### Comment · Reviewer_dN7g · 2024-08-13
> >
> > I think the authors have well addressed my comments. I shall raise my rating to 6.

---

### Author Rebuttal · Authors · 2024-08-06

We thank the referees and area chairs for their time, efforts and for their helpful comments, which have greatly improved the paper.

Two common themes among the reviewers' reports were Assumption 2 and the extension of Theorem 1 to the general $r \ge 1$ case.
In addition to these two points, two reviewers asked questions about other non-spectral methods.
We address these three themes below.

**Concerning Assumption 2**

We agree with the opinion expressed by several reviewers that Assumption 2 is not especiall natural.
Per Remarks 1 and 2 in the paper, we believe that Assumption 2 is entirely technical, and can likely be removed.
Our Section 5 experiments support this belief: our methods succeed even when Assumption 2 fails (e.g., Fig. 1).
For example, Alg. 1 outperforms "pure" spectral methods when $u^*$ is Haar-distributed, though this violates Assumption 2.

We note that to the best of our knowledge, this is the first paper to remove dependence on coherence, so there are not yet many tools for this setting.
The main difficulty arises from the fact that without Assumption 2, the selected entries $\hat{I}$ depend on $W$ in a complicated way.
We will add a more detailed discussion of these technical difficulties to the manuscript.
We aim to develop the tools for analyzing this challenging setting, but removing Assumption 2 is fundamentally hard.
Nonetheless, this is the focus of ongoing work, and we will update the manuscript accordingly should a breakthrough be achieved before the deadline.

**Extension of Theorem 1 to $r > 1$**

We agree with the sentiment, expressed by several reviewers, that a proof for the general case of $r \ge 1$ would be ideal.
As discussed in the text, our experiments in Section 5 support our claim that Algorithm 2 succeeds in the general rank-$r$ case, though of course this does not constitute a proof.
A sketch of why the rank-$r$ case should follow similarly to our rank-$1$ case is provided just before Algorithm 2.
We have given a more detailed sketch in our response to reviewer "pryd", and we will incorporate this sketch into the manuscript to provide additional intuition to future readers.
Conditional on acceptance and if time permits, we will add detailed proofs for this general case to the appendix.

**Comparison with other non-spectral methods**

To the best of our knowledge, we are the first paper to consider the problem of non-spectral entrywise eigenvector estimation.
The nearest obvious method for comparison is one based on approximate message passing (AMP; see "A Unifying Tutorial on Approximate Message Passing" by Feng, Venkataramanan, Rush, and Samworth for an overview).
We include in our attached PDF a figure summarizing additional experiments comparing our method to another non-spectral method: one based on approximate message passing (AMP).
These are analogous to our experiments in Figure 1 of the manuscript, now includ
ing an additional method.
These experiments demonstrate that our method in Algorithm 1 outperforms the only other non-spectral eigenvector recovery method of which we are aware, namely one based on AMP.

AMP methods make no explicit coherence assumptions, but the underlying mechanism essentially requires incoherence: inherent to AMP methods for eigenvector recovery is that the eigenvector is modeled as having all its entries drawn i.i.d.~according to a common distribution.
Specifically, AMP methods make a mean field assumption whereby the empirical distribution of $\{ u^*_i : i \in [n] \}$ converges in $\ell_2$ to a distribution $\pi$.
AMP-based methods fail in entrywise recovery problems because they are tailored to $\ell_2$-recovery: small $\ell_2$ error does not necessary imly small entrywise recovery.
Adapting AMP-based methods to target entrywise recovery is an interesting direction for future work, but well beyond the scope of this paper.

---

### Decision · Program_Chairs · 2024-09-25

**Decision:**

Accept (poster)

**Comment:**

The overall evaluation by the reviewers concludes this to be a technically solid theory paper making a novel contribution in terms of a spectral algorithm for eigenvector estimation for a low-rank matrix estimation and its analysis. The paper's interest is theoretical, its implications and interest for a broader NeurIPS audience are limited. I recommend acceptance as a poster.